# Precipitation, Moisture Sources and Transport Pathways associated with Summertime North Atlantic Deep Cyclones

Rikke Stoffels[1,2,3], Imme Benedict[3], Lukas Papritz[4], Frank Selten[2], Chris Weijenborg[3]

[1]Institute for Environmental Studies, Vrije Universiteit Amsterdam, Amsterdam, the Netherlands
[2]Royal Netherlands Meteorological Institute (KNMI), de Bilt, the Netherlands
[3]Meteorology and Air Quality Group, Wageningen University and Research, Wageningen, the Netherlands
[4]Institute for Atmospheric and Climate Science, ETH Zurich, Zurich, Switzerland

*Correspondence to*: Rikke Stoffels (r.stoffels@vu.nl)

**Abstract.** Extratropical cyclones are essential for redistributing moisture from lower latitudes to the poles, and are known for their ability to produce extreme precipitation. While wintertime extratropical cyclones have been studied in great detail, little is known about these systems in summer. Nevertheless, studying summer cyclones is particularly relevant in the context of climate change, as future warming is expected to increase atmospheric moisture while reducing baroclinicity. This makes present-day summer conditions an analogue for future winter cyclones and critical for understanding how summertime cyclones themselves may evolve in a warmer climate. Hence, the objective of this study is to improve our understanding of how summertime extratropical cyclones shape the characteristics of the water cycle, focusing on their moisture sources and the transport of moisture to cyclone centers. For this purpose, 8-day backward trajectories are calculated for all air parcels in the vicinity of cyclone centers with ERA5 reanalysis data, for a subset of the most intense summertime cyclones over the North Atlantic. Subsequently, moisture uptakes along the trajectories of precipitating air parcels are identified using the moisture source diagnostic WaterSip. Using this approach, it is found that the bulk of the precipitation associated with summertime cyclones falls close to the cyclone center beneath the warm conveyor belt (WCB) and along the fronts, mainly during the cyclone's intensification phase. This moisture originates from areas of high ocean evaporation, with significant hotspots on the warm side of the Gulf Stream Front. In addition, some continental sources are found, especially for cyclones in the Labrador Sea. Moisture uptake occurs primarily in regions where the strong SST gradient induces intense ocean evaporation and during cold-air advection within the cyclone's cold sector, where oceanic evaporation is enhanced due to the strong air-sea temperature contrast. The moisture accumulated in the cold sector of the cyclone does not necessarily contribute to precipitation in its own center, but it can act as a source of moisture for a subsequent cyclone. As cyclones mature, distances between the moisture source and the location where the moisture rains out decrease, but the atmospheric residence time of moisture of about four days remains approximately the same throughout the cyclone life cycle. This is because the decrease in source distance is compensated by weaker winds and less strong convergence. Overall, these results are fairly similar to those found in a previous study for winter cyclones, although in winter there is more moisture exchange between primary and secondary cyclones, and stronger vertical ascent in the WCB. Summer cyclones, on the other hand, are distinguished by their

greater moisture supply from continental sources, and the significant influence from cyclones of tropical origin undergoing extratropical transition.

**Short summary.** Summertime North Atlantic storms bring heavy rainfall, especially near their centers and along their fronts. By tracking precipitating air parcels back in time we find that the moisture originates from areas of strong ocean evaporation, with hotspots in the Gulf Stream region. We also find that sometimes evaporation in a previous storm can contribute to rainfall in the next. Unlike in winter, summer storms also draw moisture from land, and their properties are partly shaped by former tropical storms.

**1 Introduction**

Extratropical cyclones make up an important component of the global atmospheric circulation by redistributing energy, moisture and momentum from lower latitudes to the polar regions (Chang et al., 2002; Hartmann, 1994; Peixoto and Oort,1992). Cyclones enhance atmospheric moisture by promoting evaporation from various surfaces such as the land, oceans, lakes and rivers. Once the moisture enters the boundary layer, the cyclones facilitate transport over long distances during which

the moisture is subject to mixing and convection, as well as synoptic and global-scale motions (Boutle et al., 2011). At the same time, they also act as an important sink for atmospheric moisture by returning it to the surface through associated heavy precipitation, which can have severe impacts on society. For instance, the extreme rainfall associated with cyclones can result in flood events, and occasionally they have been associated with other natural disasters such as storm surges or wind gust-related damage (Dacre et al., 2019; Laurila et al., 2021; Messmer and Simmonds, 2021).

Understanding the processes that govern precipitation in extratropical cyclones is essential, as several studies (e.g. Konstali et al., 2024; Pfahl and Wernli, 2012) have found that cyclones within the midlatitude storm track region account for more than 80% of the precipitation extremes in the Northern Hemisphere, if precipitation in the accompanying fronts is included. Given their significant role, accurately forecasting these precipitation extremes and understanding how they are projected to change under global warming has received increasing scientific attention. Improving the representation of extratropical cyclones in

climate models therefore remains a key component of the work of the Intergovernmental Panel on Climate Change (IPCC) (Catto, 2016). Recent analyses of the latest generation of CMIP6 models, for example, provide new insights into the projected changes, including a decrease in cyclone frequency but more intense precipitation associated with them (Priestley and Catto, 2022). However, to fully understand these processes, a comprehensive understanding of the role of extratropical cyclones in the atmospheric water cycle — throughout their life cycle — is first required. Despite their influence on midlatitude weather

and climate, there is still much to learn about the role of extratropical cyclones in the atmospheric water cycle. This is especially true for summertime cyclones, which have received less scientific attention than their winter equivalents (Mesquita et al., 2008). This lack of attention is particularly striking given the anticipated increase in humidity and decrease in baroclinicity resulting from future warming, conditions under which future winter cyclones may resemble present-day summer cyclones. A

better understanding of these systems is therefore important for assessing both how they themselves may evolve in a warmer climate and how they can inform our understanding of future winter cyclones.

Previous research on extratropical cyclones highlights the fundamental differences between these systems and their tropical counterparts. Tropical cyclones owe their existence to high sea surface temperatures (SSTs) and deep convection, which give rise to intense precipitation concentrated near the storm's core. Conversely, extratropical cyclones are driven by baroclinicity and have characteristic frontal structures (Ahrens, 2009; Chang and Song, 2006; Eckhardt et al., 2004). Within these systems, there is low-level convergence of moist and warm air within the warm sector – the region of warm air between the cold and warm front that extends from the cyclone center in a northeasterly direction (Ahrens, 2009; Chang and Song, 2006; Papritz et al., 2021). This convergence is driven by a cross-front ageostrophic circulation on the warm side of the cold front, which strengthens the cold front and enhances upward motions (Dacre and Clark, 2025). The rising airflow is often termed the warm conveyor belt (WCB), and it is intrinsically linked to the extratropical cyclone (Eckhardt et al., 2004; Pfahl et al., 2014). As air ascends in the WCB, moisture condenses and rains out, releasing latent heat and thereby enhancing the ascent (Browing, 1990; Heitmann et al., 2024; Madonna et al., 2014). Consequently, the WCB is capable of influencing the dynamics of the cyclones – a role that has been underscored in the context of cyclone intensification by Binder et al. (2016).

Over time, the WCB can overrun the warm front and reach the upper troposphere, after which it splits into two branches: one wrapping cyclonically northward around the low-pressure center and the other wrapping anticyclonically in an easterly direction (Boutle et al., 2010; Browning, 1990). As a result, the spatial extent of the WCB not only consists of a diagonal air flow, but also stretches horizontally in the northern outflow region, creating the characteristic comma-shaped structure described in the Norwegian model (Bjerknes and Solberg, 1922) and the T-bone structure of the Shapiro-Keyser model (Shapiro and Keyser, 1990).

In the WCB, moisture can be transported from the boundary layer to the upper troposphere in roughly two days (Eckhardt et al., 2004). This level of efficiency, in combination with the extent of the uplift, results in a substantial amount of precipitation being generated in the WCB. However, the actual amount of precipitation that is generated within a cyclone depends not only on the strength of the vertical motions in the WCB, but also on the availability of moisture near the cyclone center.

The moisture that precipitates in the cyclone centers may have originated locally and remained close to the center as it propagated, or it may have been sourced from more distant areas. For example, Sodemann and Stohl (2013) found that cyclones that induced intense precipitation over western Scandinavia were supplied with water vapour by means of atmospheric rivers (ARs) originating from remote southerly regions. ARs are long and narrow filaments of high vertically integrated moisture (Pfahl et al., 2014). Although WCBs and ARs are typically defined differently – WCBs as strongly ascending cyclone-relative airflows and ARs as Earth-relative moisture plumes (Dacre et al., 2019) – they are closely linked to cyclone development and spatial overlap often exists (Dacre and Clark, 2025; Knippertz et al., 2018). In the study by Papritz et al. (2021) on the moisture sources and transport pathways of precipitating water of wintertime North Atlantic deep cyclones, subtropical moisture plumes in the form of ARs are also recognized as potential source areas, especially in the early phase of cyclone development. Additionally, Papritz et al. (2021) suggest that evaporation from the ocean surface, triggered when cold polar air passes over

relatively warmer ocean currents, supplies the cyclone with moisture. Ultimately, they identify the cold sector of a preceding low-pressure system as the predominant moisture source for wintertime North Atlantic deep cyclones. From there, moisture is transported over a relatively short distance to the center of another developing cyclone following the same track, resulting in a short moisture residence time. This phenomenon, in which a primary cyclone is succeeded by several secondary cyclones forming in succession, is known as a cyclone family or cluster (Bjerknes and Solberg, 1922; Priestley et al., 2020) and is particularly common in winter (Weijenborg and Spengler, 2024). Geographically, Papritz et al. (2021) found that moisture uptakes are mainly restricted to the western North Atlantic and are concentrated on the warm side of the Gulf Stream Front and its extension.

The pathways and sources of the precipitating water in extratropical cyclones during the summer months remain largely unexplored. In winter, a stronger equator-to-pole temperature gradient enhances the jet stream, resulting in higher wind speeds and enhanced vertical motions within the WCB. In summer, weaker temperature gradients may reduce WCB strength, but stronger latent heat fluxes from the warmer ocean surface could provide additional moisture. This raises critical questions about whether moisture pathways, such as those from the warm side of the Gulf Stream Front or even from the (sub)tropics, play a greater role in generating precipitation within summertime extratropical cyclones.

Understanding the behavior of summertime cyclones is essential, as it will allow us to discern what makes these systems unique compared to their winter counterparts. However, before we can explore the differences, we must first establish a solid understanding of what drives precipitation and moisture uptake in the summer months. This study, therefore, aims to address this knowledge gap by studying the precipitation characteristics, moisture sources, and transport pathways of summertime North Atlantic deep cyclones. We specifically focus on how these components of the cyclone-related water cycle change throughout the cyclone life cycle, and across different subregions within the North Atlantic. The following research questions will be addressed.

Q1:     What is the spatial distribution of precipitation within and around the cyclone center?

Q2:     What are the geographical moisture sources of the precipitating air parcels?

Q3:     In which dynamical environment do moisture uptakes take place?

Q4:     How do the uptake and moisture transport characteristics change throughout the life cycle of a cyclone?

Answers to these questions will be sought by identifying the precipitating air parcels for a set of 688 cyclones and tracing their origins backwards using a Lagrangian approach. Lagrangian methods follow individual air parcels as they move in space and time, while Eulerian methods analyse moisture budgets at fixed locations (Gimeno et al., 2012). Although an Eulerian approach is also suitable for constructing a moisture budget (e.g. van der Ent and Tuinenburg, 2017), the advantage of the Lagrangian approach is that it allows for the quantification of moisture uptakes along air parcel trajectories and provides a high spatial resolution of moisture source diagnostics (Gimeno et al., 2012; Pérez-Alarcón et al., 2022). Furthermore, it enables the distinction between different air streams, allowing for the analysis of their associated moisture sources separately (e.g. Pfahl

et al., 2014). The North Atlantic has been chosen as a study region for the following reasons: (1) Moisture uptakes from the ocean surface appear to have a significant contribution to the precipitation that is generated, (2) the results can be compared with results from previous research, as the North Atlantic has been studied several times before (e.g. Chang and Song, 2006; Coll-Hidalgo et al., 2025; Pérez-Alarcón et al., 2022), and (3) difficult features such as orography and roughness, both of which are induced by the transition from land to sea (Field and Wood, 2007), can be disregarded. In addition, we focus on deep cyclones, since they generate the most heavy precipitation.

The study is structured as follows. In section 2, the dataset is introduced, the methodology for selecting a subset of the most intense North Atlantic cyclones is explained, and the trajectory calculation tool and the moisture source identification process are described. Section 3 presents a detailed case study of a selected cyclone over the East Atlantic, providing insight into the underlying processes. Next, a climatological analysis of the cyclone-related water cycle associated with all 688 cyclones is presented. To that end, the cyclone precipitation is first discussed, followed by an overview of the moisture sources and transport pathways of these precipitating air parcels. The paper is concluded by a discussion and final remarks in section 5, focusing on what distinguishes extratropical cyclones that occur during summer from their winter counterparts.

## 2 Methodology

### 2.1 ERA5 reanalysis dataset

The analyses presented in this study are based on the ERA5 reanalysis dataset, which is the most recent reanalysis released by the European Centre for Medium-Range Weather Forecasts (ECMWF; Hersbach et al., 2020). For this study, we obtain data at a horizontal resolution of 0.5° x 0.5° for all extended summers (May-September; MJJAS) between 1980 and 2021, on both single levels and model levels. The variables we obtain on single levels are sea surface temperature (SST), surface evaporation, convective and large-scale precipitation, surface pressure, mean sea level pressure (MSLP) and boundary layer height. Additionally, we retrieve the variables (potential) temperature, specific humidity, and three-dimensional wind fields on a selection of model levels extending from the surface (level 137, lower boundary) to 29 hPa (level 40, upper boundary). By using precipitation data from ERA5 reanalysis, it is possible to study precipitation during different phases of the cyclone life cycle, as reanalysis provides measurements of global coverage with high temporal resolution, whereas most satellite products do not (Pfahl and Sprenger, 2016). Moreover, ERA5 is good at resolving the large-scale atmospheric processes, although convective precipitation and the associated ascent are parameterized and therefore might be less well represented (Lavers et al., 2022). While these limitations, along with its coarse spatial resolution and reliance on sparse oceanic observations, can lead to underestimation of localized intense precipitation, ERA5 remains the most suitable dataset available for studying precipitation in extratropical cyclones.

## 2.2 Identification and selection of North Atlantic deep cyclones

We identify individual cyclone tracks that developed during MJJAS using the cyclone identification and tracking scheme by Sprenger et al. (2017), which is an improved version of the one initially developed by Wernli and Schwierz (2006). It is one of many methods used to detect and track flow features such as extratropical cyclones, which typically use MSLP or lower-tropospheric vorticity as their base metric but vary in their specific methodology. A method comparison study by Neu et al. (2013) revealed that the scheme used in this study detects a number of cyclones that is similar to the average of all methods and is not an outlier.

The tracking scheme automatically identifies cyclone centers by finding local minima in the gridded hourly MSLP fields, while the cyclone edges are determined as the outermost closed MSLP contour that encloses only the minimum under consideration using a contour search algorithm with a sampling interval of 0.5 hPa (for more details, see Wernli and Schwierz (2006)). Once identified, the cyclones are tracked over time and their coordinates and pressure are recorded.

Subsequently, a subset of the global cyclone tracks is created, comprising exclusively North Atlantic deep cyclones. To this end, the cyclone depth is calculated at hourly intervals along each cyclone track, which is defined as the pressure difference between the cyclone center and the cyclone edge and is a measure of the cyclone's intensity. Thereafter, a new time of reference relative to the cyclone life cycle is defined. The time axis is centered around the time of the cyclone's maximum depth, denoted by t = 0 h. The time period where t < 0 h then represents the intensification phase, while t > 0 h, represents the decay phase. Using the cyclone depth as a measure of intensity and the relative time, two selection criteria are imposed: (1) a cyclone must reach its maximum intensity within the North Atlantic region (defined by the black line in Fig. 1), and (2) its track length must extend from at least t = -24 h to t = 24 h. In this way, we select a subset of the cyclones located in the geographical region of interest, characterized by a well-defined intensification and decay phase, thereby delineating a distinct life cycle. Single cyclones that split into separate features, or systems that merge with another cyclone are excluded by the second criterion, as such features have an intensification or decay phase that is less than 24 hours (Neu et al., 2013; Papritz et al., 2021). In addition, we exclude cases where the track length in October exceeds that in September or where the cyclone reaches its maximum intensity in October, as these cyclones already begin to exhibit wintertime characteristics due to increased baroclinicity.

For all North Atlantic cyclones with a sufficient track length (3810 tracks), the maximum depth ranges from 2 hPa to nearly 70 hPa, with a mean of 16.8 hPa. The distribution is strongly skewed, with a tail towards higher maximum intensities (Fig. 1e). Since this study focuses on the cyclone-related water cycle, the analysis is further limited to the most intense cyclones that produce the largest precipitation (Pfahl and Sprenger, 2016). To this end, a third and final criterion is used. Only those tracks with a maximum cyclone depth exceeding the 80[th] percentile threshold value are selected. As a result, 688 cyclone tracks are obtained and are divided into four groups based on the location where they reach their maximum depth: the Gulf Stream region (86 tracks), the Labrador Sea (105 tracks), the East Atlantic (231 tracks), and the Nordic Seas (266 tracks) (Fig. 1a-d). It should be noted that cyclone tracks may extend beyond the respective region where they reach their maximum depth, at any point in their lifetime. For instance, many cyclones originate over the North American continent or originate in the tropics before

extending eastward along the North Atlantic storm track, with some eventually making landfall in Europe. Consequently, the
selection of summer extratropical cyclones exhibit strong movement. To quantify this, we apply the definition of a moving
cyclone proposed by Eckhardt et al. (2004), which defines a moving cyclone as one that during its life cycle travels at least
1000 km and intensifies by more than 10 hPa. Based on this criterion, the selection is limited to three stationary cyclones,
while the remaining 685 cyclones all travel more than 1000 km and meet the intensification threshold.

In our subset of cyclones, 46 have tropical origins, which we define as originating at or below 23° N, and then undergo
extratropical transition (28 in the Gulf Stream region, 3 in the Labrador Sea, 15 in the East Atlantic). Given the tendency of
tropical cyclones to exhibit greater intensity compared to extratropical cyclones, they shift the overall distribution of maximum
cyclone depth towards higher intensity values (Fig. A1). This is particularly true in the Gulf Stream region, which also has the
highest percentage of cyclones of tropical origin (~33%). Cyclones that originate in the tropics and are among the 20%
strongest cyclones in the extratropics are not observed in winter, making them unique. Therefore, including these cyclones in
our study is both necessary and particularly interesting.

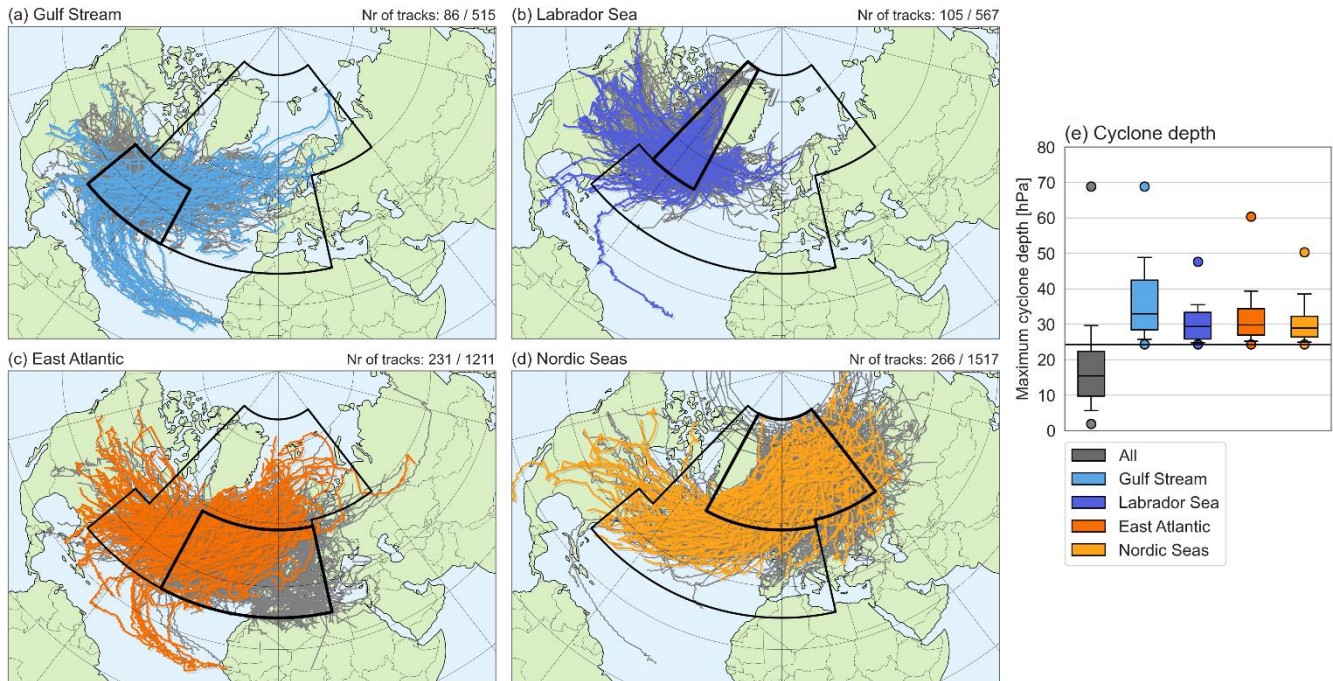

**Figure 1: Selected tracks in (a) the Gulf Stream region, (b) the Labrador Sea, (c) the East Atlantic Ocean, and (d) the Nordic Seas. The selected tracks are highlighted in colour and meet the three selection criteria for duration, intensity and location. At the top right of each panel, the number of cyclone tracks that have been selected are given as the proportion relative to the total of tracks within each of the four subregions. (e) The distribution of the maximum cyclone depth of all North Atlantic cyclones with a distinct life cycle (grey) and of the selected cyclones in the four subregions (other colours). The black horizontal line represents the threshold of the 80th percentile of all tracks. In addition, the whiskers indicate the 10th-90th percentile range, and the dots show maxima and minima.**

## 2.3 Calculation of the backward trajectories

The precipitating air parcels of each summertime cyclone are traced back in time to identify the moisture sources. This is achieved by calculating kinematic backward trajectories of air parcels surrounding the cyclone center every three hours within the time interval of t = -24 h to t = 24 h. The trajectories are computed using the Lagrangian Analysis Tool (LAGRANTO; Sprenger and Wernli, 2015; Wernli and Davies 1997), which employs three-dimensional gridded wind fields on model levels and surface pressure data. A detailed description of the data structure expected by LAGRANTO is given in Sprenger and Wernli (2015).

The LAGRANTO methodology involves a three-step procedure. In the first step, initial grid points are defined from where the air parcels are released. The number of possible grid points is constrained by the requirement that they be initialized on an equidistant grid having a mesh size of 50 km in the horizontal direction and 25 hPa in the vertical direction, extending from the surface up to 400 hPa. Furthermore, the starting positions have to be located within a circle with a radius of 500 km around the center of the cyclone (Papritz et al., 2021). The size of the radius around the cyclone center was chosen and justified by Papritz et al. (2021), and is based on the attempt to include most of the precipitation in the cyclone core that is driven by dynamical ascent, and to exclude precipitation located further away from the cyclone center that might be triggered by orography and frontal circulation. In the second step, eight-day backward trajectories are calculated from these positions (illustrated by grey dots in Fig. 2). The number of days is chosen based on three factors: (1) computational cost and computer power, (2) numerical accuracy, and (3) the ability to allocate a significant proportion of precipitation to specific moisture sources. The latter is dependent on the residence time of moisture in the atmosphere, which in the North Atlantic region is estimated to be approximately four days on average, with values up to one day longer in summer than in winter (Läderach and Sodemann, 2016). Finally, various atmospheric variables, including (potential) temperature, specific humidity, relative humidity, the boundary layer height and the latent heating rate are traced along these trajectories. The latent heating rate is estimated by LAGRANTO from changes in specific humidity, while the other variables are obtained from the ERA5 data set. Using these variables, the moisture budget of the air parcel can be constructed by applying the moisture source diagnostic WaterSip (Sodemann, 2025).

## 2.4 Diagnosis of the moisture uptakes

In this study, we employ WaterSip (version 3), a software tool that implements the widely used Lagrangian moisture source and transport diagnostic developed by Sodemann et al. (2008), to identify the moisture sources (Sodemann, 2025). The tool needs backward trajectories calculated by a trajectory calculation tool such as LAGRANTO as input, and provides source, transport and arrival properties of the moisture. We initiate simulations for each cyclone following the methodology described by Sodemann et al. (2008), as outlined below.

Since we are interested in the moisture sources of precipitation, we first select the precipitating trajectories within the cyclone center, defined as trajectories where the relative humidity (RH) exceeds 80% at the trajectory start (t = 0 h), and where a

significant amount of moisture is lost during the first three hours back in time ($\Delta q < -0.1$ g kg$^{-1}$ 3h$^{-1}$). The RH threshold aligns with the parameterisation of the ECMWF model, which assumes the presence of clouds from which precipitation falls at a RH above 80% (Sodemann et al., 2008). The two criteria have also been employed by Papritz et al. (2021), thereby facilitating a comparison of our results for the summer and theirs for winter.

In order to analyse the evaporation and precipitation events experienced by an air parcel, the changes in specific humidity are evaluated along each of the remaining trajectories. Increases in specific humidity indicate moisture uptakes, while decreases (prior to reaching the cyclone center) suggest intermediate precipitation. Assuming that (1) the integrity of these air parcels is maintained over a period of several days, (2) interactions with neighbouring parcels can be neglected, and (3) either evaporation or precipitation dominates during the 3-hour interval, the sign of $\Delta q$ indicates which process occurs during a given part of the trajectory (Sodemann et al., 2008). To avoid including small fluctuations in the specific humidity that do not correspond to actual evaporation or precipitation events, a threshold value of 0.075 g kg$^{-1}$ 3h$^{-1}$ is applied. Once all the moisture uptake locations are identified, the absolute uptake amount is translated into a moisture source footprint for every three hours within the 48-hour window around maximum cyclone depth (from -24 h to 24 h). Additionally, the fractional contribution of each uptake event along the trajectory to the precipitation at the arrival location is determined. The sum of these fractional contributions then yields the fraction of the estimated precipitation at the arrival location that can be attributed to the identified origins (Sodemann et al., 2008). Besides this accounted fraction, WaterSip also provides source, transport, and arrival properties for each trajectory. Source properties are determined as the precipitation-weighted mean of all moisture uptake locations. A similar weighting approach is applied to obtain the mean source distance and the mean accounted fraction over all trajectories of a given cyclone at a given time step, ensuring that trajectories with higher precipitation rates have a greater influence on the mean values.

WaterSip makes a distinction between moisture uptakes within the near-surface boundary layer and higher up in the free troposphere. Uptakes in the boundary layer are attributed to surface evaporation, whereas uptakes in the free troposphere may result from subgrid-scale convective ventilation or the evaporation of precipitation (Papritz et al., 2021). By summing these fractions, we obtain a more complete picture of the total moisture uptake during transport.

Figure 2 shows an example of a LAGRANTO trajectory for a particular cyclone and illustrates the WaterSip methodology. Along the trajectory there are two moisture uptakes of almost similar amounts at t = -168 h and t = -48 h, as well as an intermittent precipitation event at t = -96 h. The uptake event at t = -48 h fully contributes to the precipitation in the cyclone center since there is no precipitation event between t = -48 h and t = 0 h. In contrast, the moisture that the air parcel has gained at t = -168 h partly precipitates during the intermittent precipitation event, and will have a smaller contribution to the precipitation at the cyclone center.

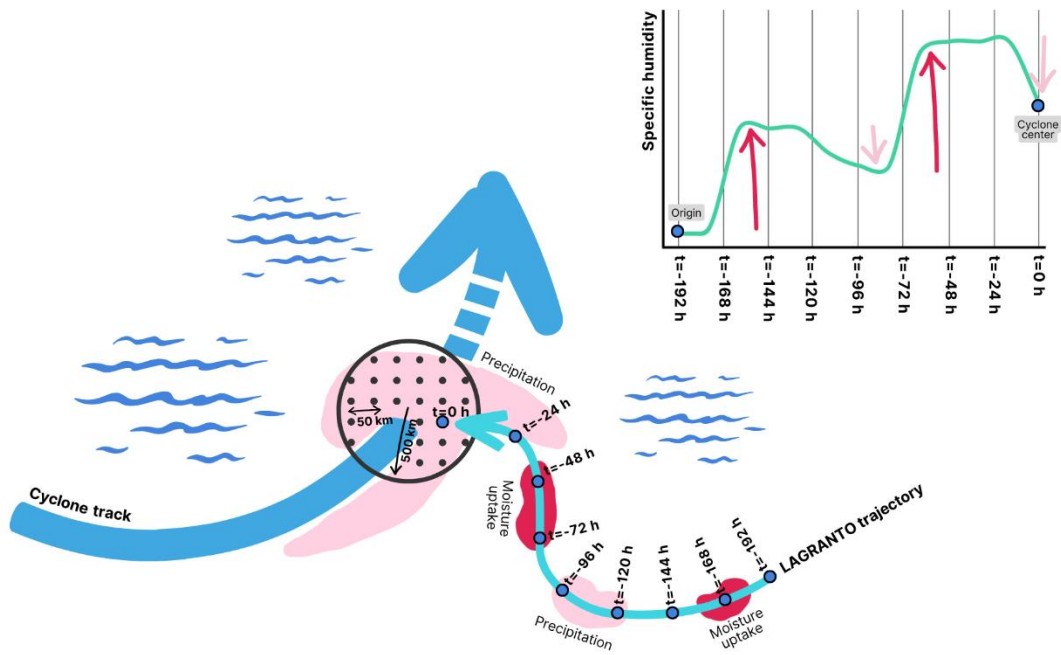

**Figure 2: Illustration clarifying the principle of the moisture source diagnostics applied to the LAGRANTO backward trajectories.** The blue arrow shows the cyclone track, from which, at a given time, precipitating air parcels that are within a 500 km radius of the cyclone center (grey circle) are tracked backward in time. The light blue arrow shows a possible trajectory that has been calculated by LAGRANTO and will be evaluated by WaterSip. The light pink patches indicate (intermittent) precipitation and the dark pink patches indicate evaporation along the trajectory. The top diagram shows the change in specific humidity over time along the trajectory shown by the light blue arrow, due to the precipitation events (light pink arrows) and evaporation events (dark pink arrows). Figure is adapted from Papritz et al. (2021).

## 3. Case study of an East Atlantic cyclone

We begin this study with a detailed analysis of a randomly selected case study over the East Atlantic. Studying this case helps clarifying the underlying processes that shape the cyclone-related water cycle and the mechanisms by which moisture is transported into the center of the cyclone, which will ultimately help in the interpretation of the climatological analyses. Although this cyclone was chosen arbitrarily, its landfall in Europe makes it particularly relevant, as such cyclones can have significant regional impacts.

### 3.1 Synoptic flow evolution

On 4 May 2019, at 23:00 UTC, a cyclone developed in the Gulf Stream region as a secondary cyclone that is part of a cyclone cluster (white cross labelled C2 in Fig. 3 and 4). Over the next two and a half days, the cyclone intensified and travelled to the East Atlantic, where it attained its maximum depth of 35.2 hPa. The decay phase lasted just over two days, during which the cyclone made landfall in Europe. The maximum depth of this cyclone falls within the interquartile range of the selected intense

East Atlantic cyclones (cf. Fig. 1e), and it has a representative amount and timing of precipitation, with most precipitation falling during the intensification phase (not shown). Therefore, this cyclone can be considered a typical deep East Atlantic cyclone.

Figures 3 and 4 show the synoptic flow evolution prior to and during the development of the East Atlantic cyclone. In the three days prior to the cyclone attaining its maximum depth, the presence of another preceding cyclone was observed (C1 in Fig. 3 and 4). An examination of the SLP contours reveals that C1 is embedded within a larger wave train, with downstream alternating cyclones and anticyclones. The rapid formation of multiple cyclones in succession, so-called cyclone clustering, is a common feature observed over the North Atlantic. According to Priestley et al. (2020) and Weijenborg and Spengler (2024),

more than 50% of cyclones along the main North Atlantic storm track are part of a cyclone cluster. While this behaviour has primarily been studied in the context of winter cyclones, the present case illustrates that such cyclone clusters also occur in summer. The dynamical flow pattern in the days before C2 developed was predominantly influenced by C1, as shown in Fig. 3. This figure shows 12-hourly maps of 900 hPa sea–air potential temperature difference for t = -72 h to t = 24 h. The passage of C1 is followed by the presence of a region of strongly positive sea–air potential temperature difference, i.e., cold-air

advection over a warm ocean surface, particularly in the hours leading up to the development of C2 (Fig. 3a-c). This region is surrounded by strongly negative sea–air potential temperature difference, i.e., warm-air advection. The region where colder air is advected over a warmer underlying sea surface represents the cyclone's cold sector, and is located to the southwest of the cyclone center. Here, we observe strong upward fluxes of sensible and latent heat (Fig. 4a-c), which will result in a moistening and deepening of the boundary layer (Demirdjian et al., 2023; Papritz et al. 2015). To the west of this region of

intense surface evaporation, we observe the genesis of C2 (Fig. 3c). Given that C2 does not develop upstream as part of the prevailing wave train and the cyclone is considerably smaller in scale compared to C1, it seems likely that the cyclogenesis of C2 is the result of a small-scale disturbance along the baroclinic jet. Therefore, this particular case appears to deviate from the typical cyclone cluster as described in the literature (e.g., Bjerknes and Solberg, 1992; Demirdjian et al., 2023) in that the two cyclones do not follow a similar track, but instead, C1 becomes stagnant and C2 propagates to the south of it and eventually

overtakes it.

Another region of high evaporation is located to the south the C2, where a positive sea–air potential temperature difference coincides with the subtropical high. Here, relatively cold, dry air descends over a warmer ocean surface, enhancing evaporation. The positioning of the cyclone and anticyclone induces easterly winds along the Gulf Stream Front, a region characterized by a sharp transition from negative to positive sea-air temperature differences due to the strong SST gradient.

As a result, intense evaporation occurs on the warm side of the Gulf Stream Front, particularly between t = −12 h and t = 12 h (Fig. 4f-h).

Additional regions of elevated surface evaporation seen in Fig. 4 include the land surface and cold-air advection within the cyclone's own cold sector. However, the latter becomes prominent only later in the cyclone's life cycle, from t = 0 h onward. To determine whether these evaporation hotspots indeed contribute to precipitation within the cyclone center, we will in the

following section consider the moisture sources for this case.

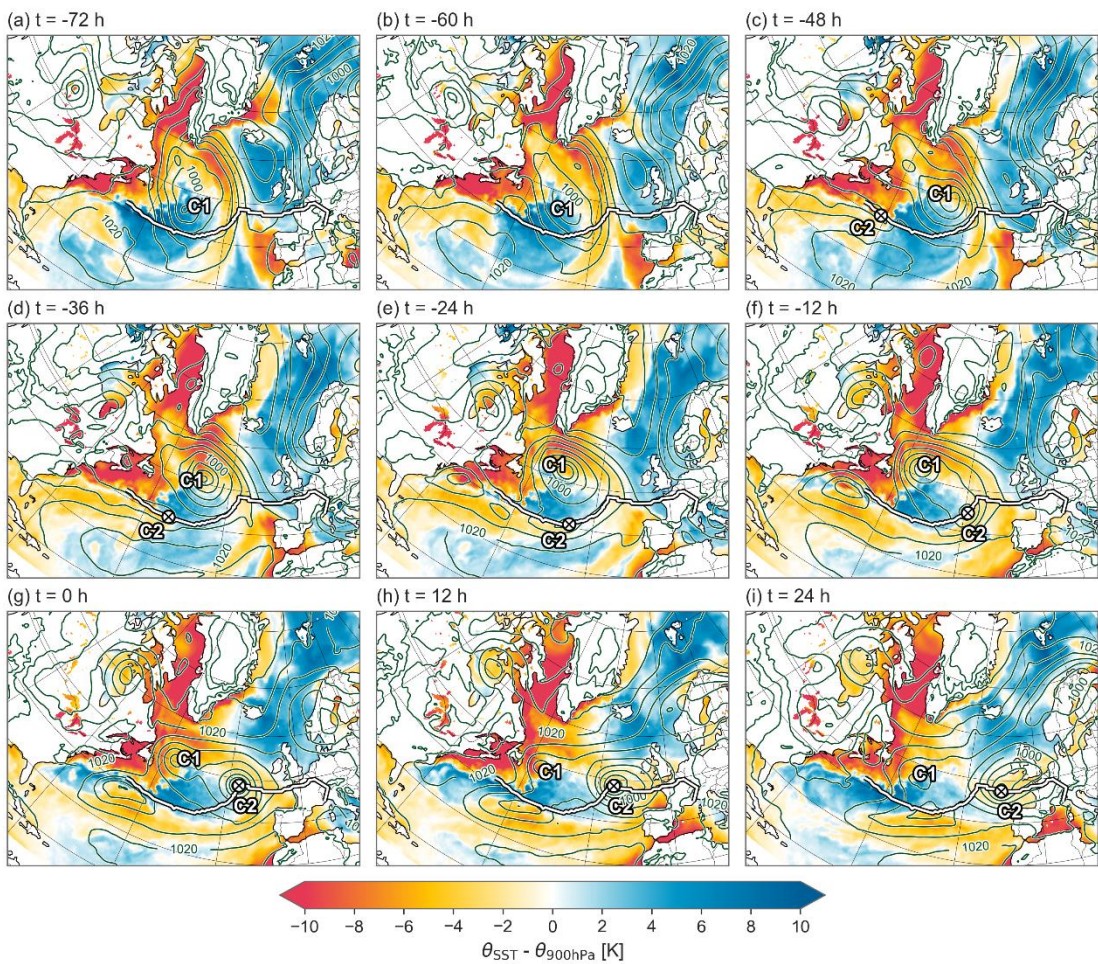

**Figure 3: Sea-air potential temperature difference ($\theta_{SST}-\theta_{900hPa}$, in K) associated with an East Atlantic cyclone, for t = -72 h (4 May 2019, 10:00 UTC) to t = 24 h (8 May 2019, 10:00 UTC) in 12-hourly intervals. Additionally shown are SLP contours (in intervals of 5 hPa; dark green lines), the cyclone track (white line), and the cyclone center location (white cross labelled C2; visible only from -48 hours onward, as cyclogenesis did not occur until -59 hours). Label C1 represents the location of the preceding primary cyclone.**


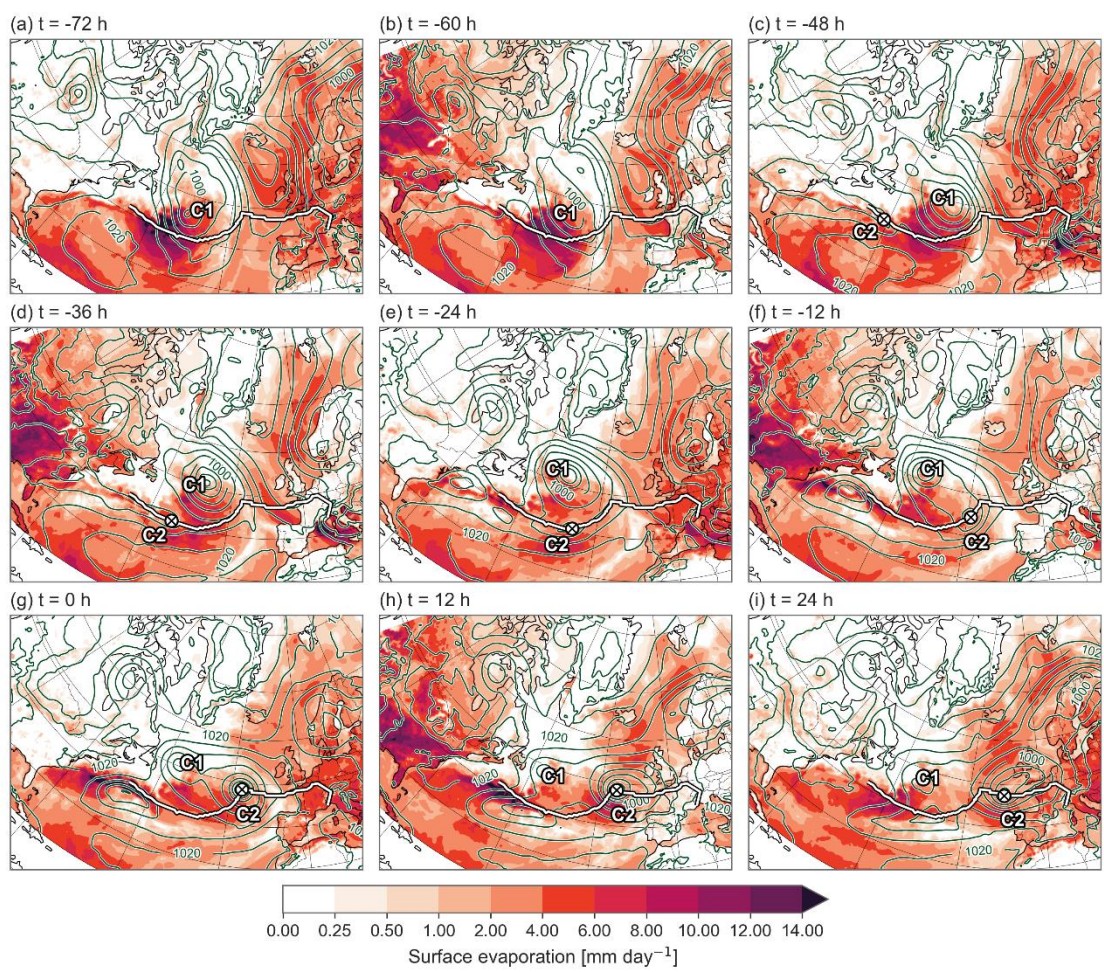

**Figure 4: Surface evaporation (in mm day⁻¹) associated with an East Atlantic cyclone, for t = -72 h (4 May 2019, 10:00 UTC) to t = 24 h (8 May 2019, 10:00 UTC) in 12-hourly intervals. Additionally shown are SLP contours (in intervals of 5 hPa; dark green lines), the cyclone track (white line), and the cyclone center location (white cross labelled C2; visible only from -48 hours onward, as cyclogenesis did not occur until -59 hours). Label C1 represents the location of the preceding primary cyclone.**

### 3.2 Moisture sources and transport pathways

For this East Atlantic cyclone, precipitation peaked 12 hours before the cyclone reached its maximum depth, after which it notably decreased (not shown). The corresponding moisture sources of the precipitation falling at three distinct times during the cyclone life cycle – the intensification phase (t = -24 h), the mature stage (t = 0 h), and the decay phase (t = 24 h) – are shown in Fig. 5. For this particular cyclone, the largest amount of moisture is taken up by the later precipitating air parcels during the intensification phase (Fig. 5a), on the warm side of the Gulf Stream Front, and only very few uptakes take place over land. During this phase, the weighted mean source distance exceeds 2000 km. As the cyclone moves eastward throughout its life cycle, the sources of moisture undergo a similar shift, moving in an eastward direction and being closer to the cyclone center (Fig. 5b, c), while most of the uptakes remain aligned with the north-eastward extension of the Gulf Stream Front.

During the time of maximum depth and during the decay, the weighted mean source distance remains above 1500 km, indicating the existence of remote moisture sources throughout the cyclone's life cycle. The overall moisture source footprint of this cyclone is also reduced over its life cycle, but this is because there is simply less precipitation falling during the later part of its life cycle that can be attributed to a specific source.

Throughout most of its life cycle, more than 85% of the moisture content of the precipitating trajectories for this cyclone have,
on average, been assigned to a source region, either within the boundary layer or in the free troposphere (Fig. 5). The majority of the moisture is taken up within the boundary layer, implying that most of the moisture is gained by surface evaporation. Roughly 10% of the precipitating air parcels even reach more than 90% attribution, while only 5% have 10% or less attribution, meaning that WaterSip effectively attributes the majority of precipitation for this particular case to its source region. Moisture can also enter air parcels through subgrid-scale convective ventilation or evaporation of precipitation in the free troposphere,
but the fraction of moisture uptake identified in the free troposphere is less than 30% in more than 70% of the trajectories.

Figure 5 presents the moisture sources for this case, but does not provide insight into the movement of air parcels before they gained the moisture or their subsequent transport while carrying it. Specifically, if the moisture uptake is already accounted for in the first day(s), Fig. 5 does not reveal where the air parcels were located in the days before uptake, nor whether the moisture was acquired in a single event or multiple stages. To address this, we consider the trajectories of the precipitating air
parcels and show for the 20% of trajectories exhibiting the most intense precipitation their evolution 8-days backward in time on a latitude-longitude grid (Fig. 6a-c) and in the vertical direction (Fig. 6d-f). While the moisture source footprint showed a rather coherent source region, the trajectory figures illustrate that there are actually several different pathways how the precipitating air parcels reach the cyclone center.

During the intensification phase, the precipitating air parcels originate from a range of vertical levels (Fig. 6d) and appear
relatively unorganized, also in a geographical frame of reference (Fig. 6a). Some trajectories flow along the Gulf Stream front, where the strong SST gradient drives moisture uptake far from where the eventual precipitation falls. Other air parcels arrived from the north of the cyclone, having been embedded in the cyclonic flow of C1. Some of these parcels experienced intermittent precipitation, remained aloft and subsequently joined cold, dry air in the mid-troposphere. Together, they descended behind the cold front of C1 resembling the dry intrusion airstream described in Browning's conveyor belt model (1990, 1997). Upon
descending into the boundary layer, they gained substantial moisture through mixing. Moreover, surface winds are relatively weak prior to the arrival of C2, facilitating a build-up of moisture. As the developing cyclone propagates into the region, the accumulated moisture is likely entrained into the warm sector and connected into the WCB and the AR through a feeder airstream, according to the mechanism proposed by Dacre et al. (2019). Lastly, there are also some trajectories that travelled over land, where there were several alternating cycles of precipitation and evaporation.

At the time of maximum depth, two distinct vertical pathways emerge: one characterized by descending air parcels from higher up in the atmosphere, and another remaining close to the surface (Fig. 6e). Both airstreams originated from C1 or further north along the Greenland coast before being swept up by C2. During the observed pathways, minimal or negligible changes in

specific humidity are evident, with the exception of moisture uptakes closer to the cyclone center, suggesting the possibility of the airmasses being initially cold and dry.

Finally, as the cyclone matures and the velocity field becomes more organized, almost all precipitating trajectories originate from within the boundary layer, near the surface (Fig. 6f). There are still air parcels that trace back to C1 (Fig. 6c), but several are now embedded in the cyclonic flow of the cyclone itself, which becomes more stationary over time. This is associated with several cycles of precipitation and re-evaporation, implying that local moisture recycling is becoming important later on in the cyclone life cycle.

This case study demonstrates that moisture sources correspond to regions of high ocean evaporation, but the mechanisms driving moisture uptakes contributing to precipitation in the cyclone center vary throughout the cyclone's life cycle. Flow along the Gulf Stream Front facilitates long-range moisture transport from warm ocean surfaces, particularly during the intensification phase. Meanwhile, cold-air advection in the cold sector of a primary cyclone is essential throughout the entire life cycle. Two distinct transport pathways emerged for precipitating air parcels originating from the primary cyclone: dry air
descending from higher up in the atmosphere and cold-air advection along the surface. Regardless of the origin, all parcels are eventually drawn into the warm sector near the cyclone center by low-level convergence, where they ascend rapidly just before the onset of precipitation.

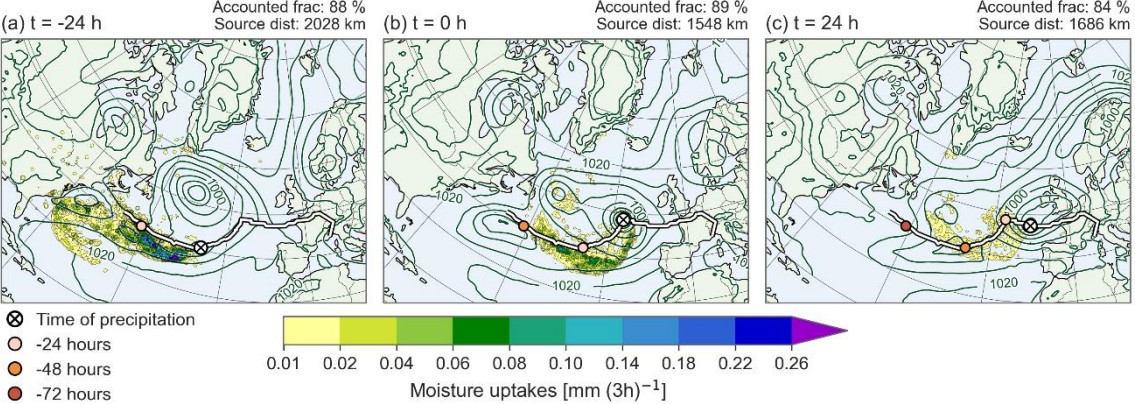

**Figure 5: Moisture source footprint of uptakes contributing to precipitation in the cyclone center of an East Atlantic cyclone, for**
**(a) t = -24 h (6 May 2019, 10:00 UTC), (b) t = 0 h (7 May 2019, 10:00 UTC), and (c) t = 24 h (8 May 2019, 10:00 UTC). Additionally shown are MSLP (in intervals of 5 hPa; dark green contours), the cyclone track (white line), and the cyclone center location at the time of the precipitation (white cross). Cyclone center positions at 24-hour intervals prior to the precipitation are indicated by coloured dots. The weighted mean accounted fraction and source distance are given in the top right of each panel.**

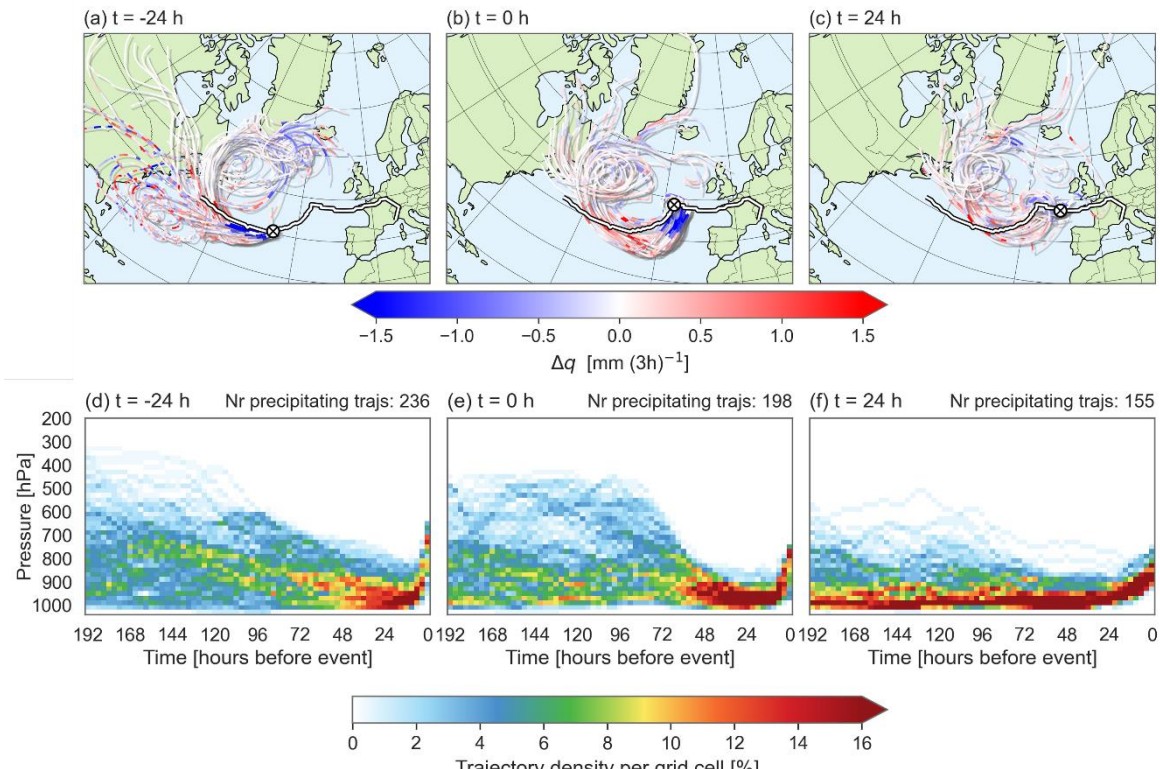

**Figure 6: 8-day backward trajectories of the precipitating air parcels of an East Atlantic cyclone, for (a) t = -24 h (6 May 2019, 10:00 UTC), (b) t = 0 h (7 May 2019, 10:00 UTC), and (c) t = 24 h (8 May 2019, 10:00 UTC). The trajectories are coloured by their change in specific humidity over a 3-hour period (red means moisture uptake events and blue intermittent precipitation). Additionally shown are the cyclone track (white line) and the cyclone center location (white cross). The three bottom panels show the vertical density, and the number of trajectories shown here is indicated in the top right of each panel. For all panels, only the 20% of trajectories exhibiting the most intense precipitation are shown.**

## 4. Climatology

In this section, we consider all 688 summertime North Atlantic deep cyclones. We first examine the spatial distribution of the cyclone precipitation, followed by the geographical moisture uptake locations, and a general characterization of the moisture uptake conditions and transport. Finally, we also provide an analysis of the cyclone-relative perspective on the uptake locations and transport pathways. Our goal is to identify common patterns and key moisture source regions, building on insights from the case study.

### 4.1 Cyclone precipitation

Figure 7 shows the mean spatial distribution of precipitation for the cyclones in each subregion during the intensification phase (t=- 24  h), time of maximum depth (t=0  h) and decay phase (t=24  h). In the figure, the total precipitation from ERA5 is averaged on a cyclone-relative grid, where the grid centre corresponds to the cyclone center, regardless of the geographical

location in the study area. For all subregions, precipitation within the cyclone center (black circle) is highest during the intensification phase (Fig. 7a, d, g, j), prior to the cyclones reaching their maximum depth. In contrast, the lowest precipitation rates are observed during the decay phase (Fig. 7c, f, i, l).

Cyclones in the Gulf Stream region exhibit the highest precipitation rates throughout their entire life cycle, while those in the Nordic Seas show the lowest. The relatively cold ocean surface in the Nordic Seas limits surface evaporation and moisture supply, leading to low precipitation rates. In contrast, the Gulf Stream region experiences higher precipitation rates due to enhanced surface evaporation over a warmer ocean surface, which significantly moistens the warmer boundary layer (Aemisegger and Papritz, 2018; Bui and Spengler, 2021; Pfahl et al., 2014). In addition, the presence of dry air masses associated with subsidence in the downward branch of the Hadley circulation further enhances evaporation by lowering near-surface relative humidity. Together, these processes increase the moisture content in the Gulf Stream region, leading to more convective precipitation and allowing for stronger latent heat release during large-scale ascent within the cyclones, which in turn serves to further intensify them. As a result, the strongest cyclones are observed in the Gulf Stream region (Fig. 1e). In the early phase of the cyclone life cycle, the higher precipitation observed in the Gulf Stream region relative to other subregions is due to stronger convective precipitation (Fig. A2). As the cyclone progresses towards its maximum intensity, there is also a significant increase in large-scale precipitation, peaking just before the time of maximum depth. Even during the decay phase, large-scale precipitation remains higher for Gulf Stream cyclones compared to those in other regions of the North Atlantic.

In addition to these differences in the magnitude, there are also important differences in the spatial distribution of precipitation around the cyclone center across the subregions. In the Gulf Stream region, the highest precipitation rates are found near the cyclone center, displaying a near-symmetrical structure (Fig. 7a-c). This symmetry reflects the influence of cyclones with tropical origins, which have undergone tropical-extratropical transitions and tend to be more intense. In contrast, cyclones in the other subregions exhibit a more typical midlatitude cyclone structure, characterised by precipitation concentrated along the warm and cold fronts – primarily generated by dynamical uplift – and in the warm sector, associated with thermodynamically driven convection (Fig. 7d-l). A similar cold frontal signature is also evident in Gulf Stream cyclones, though not until later in the decay stage when the cyclone adopts a more extratropical structure.

In the regions not affected by cyclones with tropical origins, a prominent diagonal band of precipitation that stretches from the southwest (-10°, -10°) to the northeast (+5°, +5°) is visible, which is mainly present during the intensification phase and at the time of maximum depth. Within this region, precipitation rates range from 8 to more than 32 mm day$^{-1}$ at the center of the cyclone, suggesting the presence of the WCB. As the cyclones evolve, occluded fronts develop, resulting in a narrowing of the warm sector and an increasing gap between the warm sector and the cyclone's low-pressure center. This transition marks the decay phase, which is accompanied by a significant drop in precipitation at $t = 24$ h for all cyclones except those in the Gulf Stream region, likely due to the sustained high moisture availability. Although exact occlusion timings are difficult to determine due to the absence of frontal detection and because Fig. 7 is a spatial composite, the observed patterns support this interpretation.

In summary, precipitation associated with North Atlantic deep cyclones depends on cyclone strength and moisture availability.

Precipitation rates therefore peak during the intensification phase when vertical motions are strongest, and in the Gulf Stream region, where there is abundant moisture from strong ocean evaporation over warm waters.

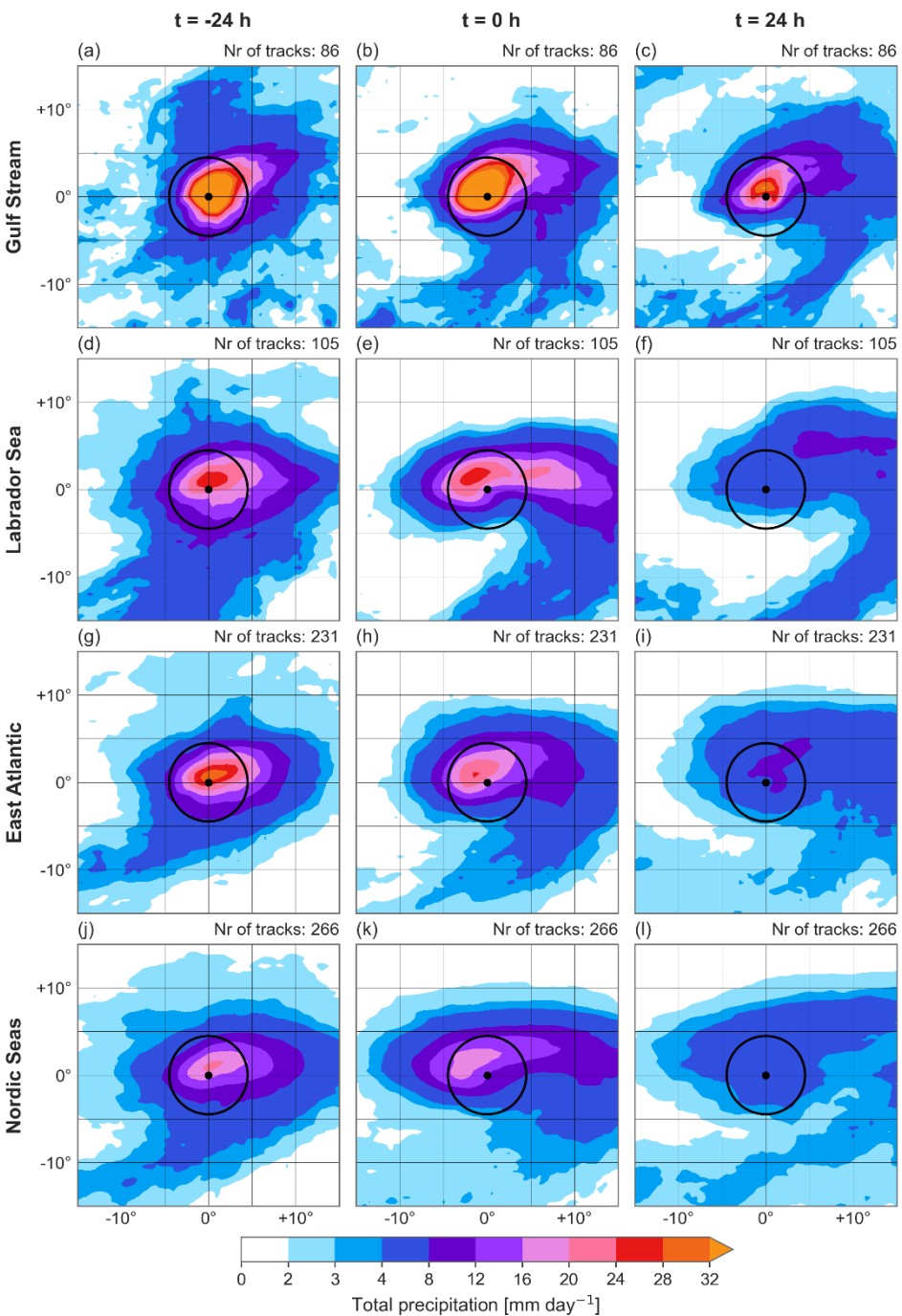

**Figure 7: Cyclone-relative composites of the mean total precipitation (in mm day$^{-1}$) of the cyclones in (a)-(c) the Gulf Stream, (d)-(f) the Labrador Sea, (g)-(i) the East Atlantic, and (j)-(l) in the Nordic Seas from t = -24 h to t = 24 h in intervals of 24 h. Precipitation is averaged over 12-hourly intervals centered on the given relative time t. Additionally shown are the cyclone center location (black dot) and surrounding circle with a radius of 500 km (black circle). The number of cyclones contributing to the composite is indicated in the top right of each panel.**

## 4.2 Geographical origin of moisture

The geographical locations of moisture uptake associated with precipitation falling during the three phases of the cyclone life cycle, for the four subregions, are shown in Fig. 8. Overall, WaterSip effectively attributes the majority of precipitation to its source region. For approximately 90% of all precipitating air parcels, over 50% of the water vapour arriving at the cyclone center could be accounted for within the 8-day period (Fig. A3). The remaining moisture either originates from an unidentified source region or was acquired by air parcels beyond the backward trajectory tracking period.

Figure 8 highlights a clear pattern: cyclones attaining their maximum depth in the Gulf Stream region take up the most moisture, while those in the Nordic Seas acquire the least. This finding is consistent with what we found for precipitation (Fig. 7). Moreover, moisture uptake associated with precipitation from cyclones in all subregions is pronounced in the western part of the North Atlantic, and concentrated on the warm side of the Gulf Stream Front. These areas provide exceptional moisture supply due to strong oceanic evaporation over elevated sea surface temperatures. This pattern is important because it applies to all subregions within the North Atlantic, making it a consistent and robust feature of moisture sources for cyclone precipitation. The findings are also in agreement with a previous study (Pfahl et al., 2014) of moisture sources for WCBs and underline the significance of these sources.

An intriguing detail of the moisture source footprint for cyclones in the Gulf Stream region and East Atlantic is its slight extension to the southeast, which likely reflects the influence of cyclones with tropical origins. These cyclones contribute to the moisture supply by transporting moisture from the subtropics to the extratropics. However, these southeastern moisture sources remains secondary to midlatitude sources, and we do not see any sources deep in the tropics.

Another important observation of the moisture source footprint is its large spatial extent, especially during the intensification phase. Consequently, the weighted mean source distance is also large during this phase. This suggests that North Atlantic cyclones can access moisture reservoirs distributed across large parts of the ocean, rather than depending solely on moisture that has evaporated in close proximity of the cyclone center. As the cyclones evolve, the spatial distribution of the moisture footprints becomes more compact, particularly for those that reach their maximum intensity in the eastern part of the North Atlantic. This is shown by a decreasing standard deviation of the sources' latitude and longitude positions (Fig. A4f). As the footprints become more compact, they also shift in space. Across all subregions, the moisture sources remain aligned with the warm side of the Gulf Stream Front, moving in a northeasterly direction, consistent with the movement of the mean cyclone centers (Figs. 8 and A4a-d). During the decay phase, however, the mean moisture footprints in the Labrador Sea and Nordic Seas tend to shift more northward relative to the mean cyclone track. This north-eastward shift of the moisture footprint exceeds the movement of the mean cyclone center in all subregions except in the Gulf Stream region (Fig. A4e), as the cyclones slow down once they mature. This implies that, over time, moisture sources move closer to the cyclone center. The weighted mean

source distance, which scales with the cyclones' propagation speeds, thus decreases throughout the cyclone life cycle (Fig. 8). At a later stage, local moisture recycling becomes more important, although overall moisture uptake is lower than during the intensification phase.

Lastly, it is observed that there are also some continental moisture sources for cyclone precipitation associated with cyclones in the Labrador Sea and, to a lesser extent, the East Atlantic during the summer months. This finding suggests that land evapotranspiration becomes a significant contributor to cyclone precipitation during the summer season (Pfahl et al., 2014; van der Ent and Tuinenburg, 2017). The extent to which surface evapotranspiration contributes to cyclone precipitation depends on soil moisture availability, but a discussion of this influence and variability is beyond the scope of this research. Nevertheless, this finding underscores the diverse nature of the origin of moisture precipitating in summertime cyclones, emphasizing the significance of taking into account seasonal variability and terrestrial impacts in the cyclone related water cycle.

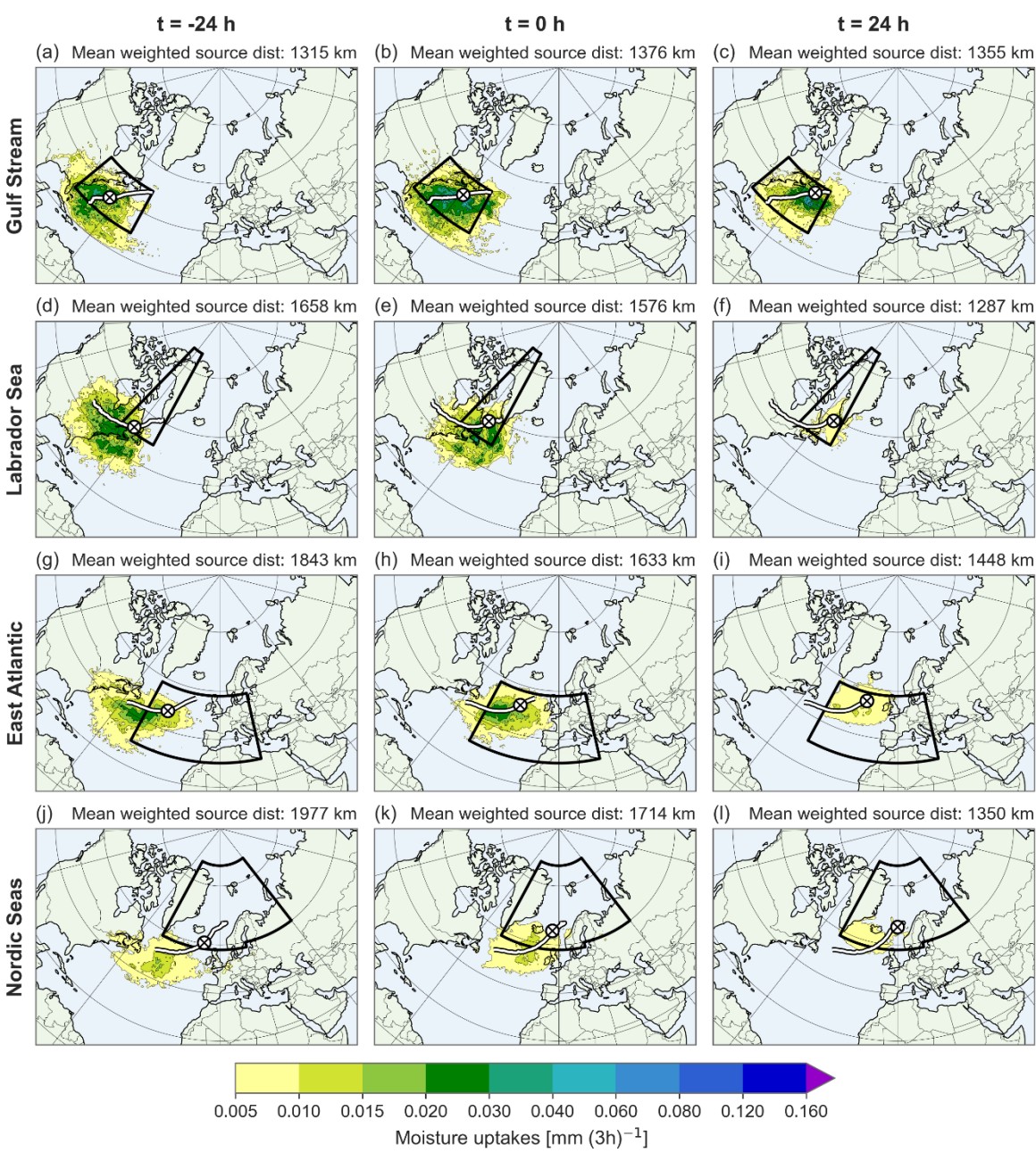

**Figure 8: Climatological mean moisture source footprint of moisture uptakes contributing to precipitation in the cyclone center of the cyclones in (a)-(c) the Gulf Stream, (d)-(f) the Labrador Sea, (g)-(i) the East Atlantic, and (j)-(l) in the Nordic Seas from t = -24 h to t = 24 h in intervals of 24 h. The regions used for the selection of the cyclones at their time of maximum depth are outlined by black boxes and the mean weighted source distance is given in the top right of each panel. Additionally shown are the mean cyclone track (white line), and the mean cyclone center location at the time of the precipitation (white cross).**

## 4.3 Moisture uptake characteristics

Figure 9 provides insight into the environment in which the moisture uptakes take place. Unlike Fig. 8, which focuses on three specific phases during the cyclone life cycle, Fig. 9 presents data at 3-hourly intervals, allowing us to analyse how these characteristics evolve over time.

Moisture uptake occurs over a wide range of distances from the cyclone center, from local sources to over 3000 km away from the location where the precipitation ultimately falls. Throughout the cyclone life cycle, about 20% of the moisture comes from sources within 1000 km from the cyclone center, a fraction that remains relatively constant. However, moisture uptake from sources located closer than 500 km from the precipitation location is remarkably rare. During the intensification phase of the cyclone, roughly 20% of the moisture sources are located more than 2500 km from the cyclone center, indicating enhanced long-range transport of warm, moist air masses that traverse the Gulf Stream Front or originate from the subtropics. This finding is consistent with Fig. 8, which highlighted that the moisture source footprints have a large spatial extent, especially during the intensification phase. While distant moisture sources are still observed, the contribution of moisture originating 1000-1500 km from the precipitation locations increases from just over 20% to nearly 40% during the decay phase.

Most of the moisture uptake occurs over the ocean surface (Figs. 8 and 9b), thereby indicating that contributions from land surfaces are of secondary importance. For at least 80% of the cyclones, less than 50% of the moisture comes from land. However, land sources play a dominant role for cyclones intensifying in the Labrador Sea, where over 60% of cyclones receive more than half of their moisture from land during this phase (Fig. A5). In fact, for 25% of cyclones in this subregion, moisture originating from land remains the primary contributor until they reach their maximum depth, with over 70% of the precipitating water linked to terrestrial sources.

The atmospheric residence time – the time between moisture uptake and precipitation – is shown in Fig. 9c. More than 80% of the moisture uptake occurs within four days of precipitation, suggesting that this is the typical residence time for cyclone-related moisture during the summer. For the remainder, nearly 15% of the moisture rains out within only two days, indicating that there is also a rapid turnover within the cyclone system. Despite the greater source distances observed during the intensification phase (Fig. 9a), the residence time remains relatively constant, suggesting that moisture must be transported more rapidly during this phase. This is consistent with the presence of stronger winds and a faster-moving cyclone, both of which facilitate faster long-range transport. As the cyclone matures, weaker winds tend to slow down the convergence of moisture at the surface, but at the same time the source distances are not as large.

The analysis reveals that summertime deep cyclones in the North Atlantic exhibit subtle shifts in their moisture uptake characteristics. During the intensification phase, stronger winds enhance long-range transport, allowing moisture to originate from distances exceeding 2500 km. These sources are typically linked to land evapotranspiration or evaporation from warm ocean surfaces into subtropical air masses. As the cyclone progresses, these contributions decrease, and moisture uptakes from sources 1000–1500 km away from the cyclone center become increasingly important.

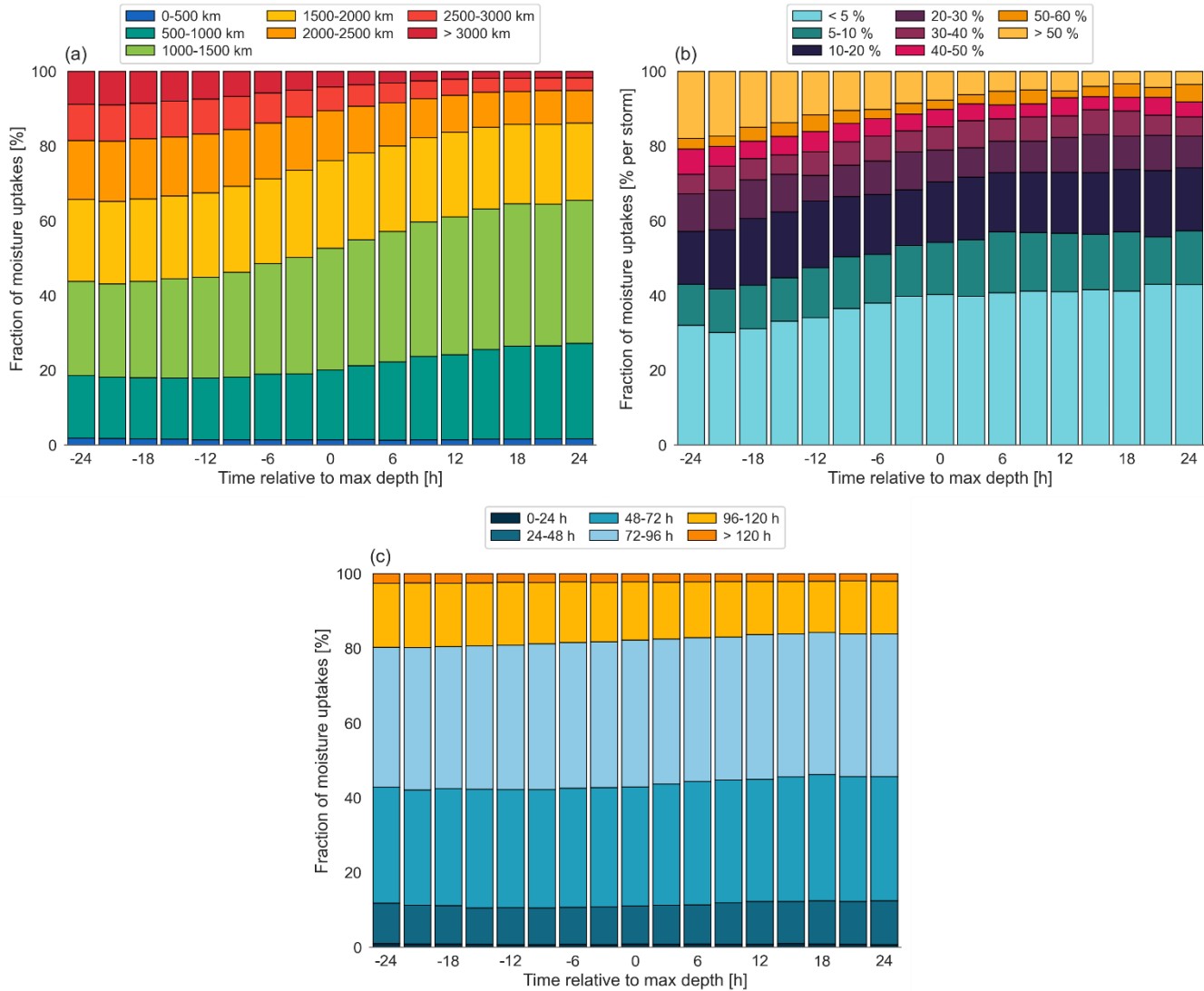

**Figure 9: Characteristics of moisture uptakes contributing to precipitation at t = -24 h to t = 24 h in 3-hourly intervals, associated with cyclones from all regions. Shown are uptake contributions according to (a) distance between uptake and precipitation locations, (b) fraction of uptakes that have occurred over land, and (c) moisture residence time.**

### 4.4 Vertical distribution of the precipitating air parcels

Further insight into how the moisture is transported to the cyclone center can be obtained by analysing the vertical distribution of the trajectories of precipitating air parcels. Figure 10 shows the vertical density of these trajectories over the full eight-day period preceding precipitation. Since the figure represents the mean of all precipitating trajectories across 688 cyclones, it is difficult to identify distinct vertical transport pathways. However, some common patterns emerge.

During all three phases in the cyclone life cycle, air parcels are concentrated in the lower troposphere (below 800 hPa) in the days leading up to precipitation, followed by a pronounced ascending motion within the last 12 hours before the precipitation.

This ascent occurs primarily in the warm sector of the cyclone and is associated with the WCB. As the air parcels are lifted in the WCB, they cool and reach saturation, leading to condensation and strong latent heat release that further enhances vertical motion.

Previous studies (e.g., Madonna et al., 2014) have identified WCBs by selecting trajectories that ascend at least 600 hPa within 48 hours. In our analysis, we did not explicitly isolate WCB trajectories using this criterion. However, a visual inspection of Fig. 10 reveals that the trajectories do not reach altitudes above 500 hPa, indicating that such strong ascent is less common in our subset of summertime North Atlantic deep cyclones. This observation supports that vertical motions in the warm sectors of summertime deep North Atlantic cyclones are weaker than in other seasons, likely due to reduced baroclinicity and overall weaker cyclones in summer. While the ascent may fall short of the standard WCB threshold, these airstreams still exhibit poleward transport of warm, moist air and some ascent, and we therefore refer to them more loosely as WCB-like flows throughout this study.

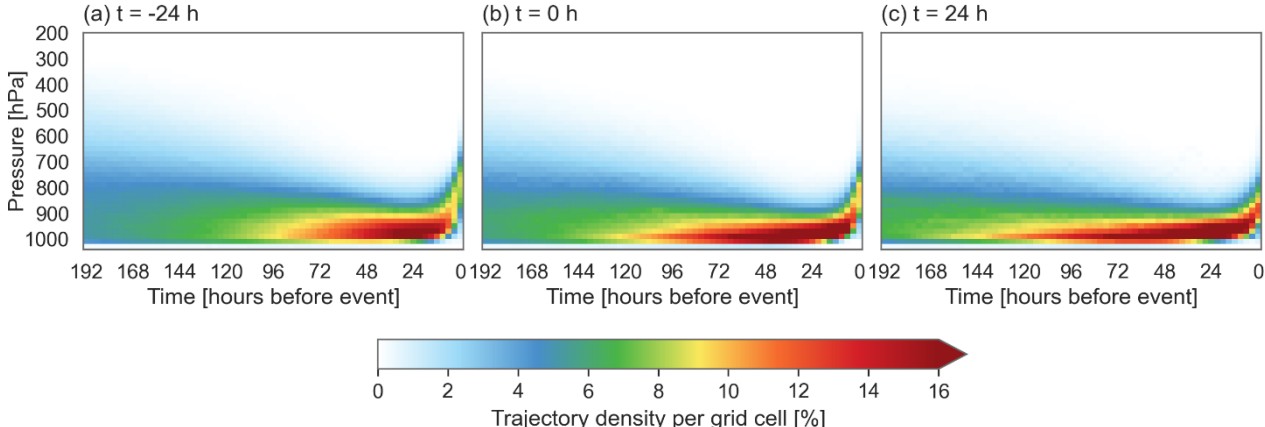

Figure 10: The mean vertical density of only the 20% of trajectories of each cyclone exhibiting the most intense precipitation, for (a) t = -24 h, (b) t = 0 h, and (c) t = 24 h.

## 4.5 Cyclone relative perspective on evaporation

Based on our previous findings, we have identified that moisture from the warm side of the Gulf Stream Front is supplied to the cyclone centers in the North Atlantic. From a cyclone-relative perspective, these moisture sources lie to the south of the cyclones. Additionally, we observed that cold-air advection can contribute to boundary-layer moistening. To quantify this for the entire cyclone subset and to better understand where it occurs relative to cyclone centers, we study the spatial distribution of surface evaporation within and around the cyclone center. The moisture source footprints produced by WaterSip are aggregated fields over the 8-day period, making it impossible to examine instantaneous moisture uptake locations and their positioning relative to the cyclone center. Nevertheless, composites of surface evaporation for the days prior and during cyclone development can be examined. Figure 11 shows the mean surface evaporation, along with cyclone-relative winds and sea level

pressure contours, for the cyclones in all subregions on a cyclone-relative grid in 12-hourly intervals. Please note that evaporation during this period does not necessarily contribute directly to precipitation in the developing cyclones.

In the early phase of cyclone development, the MSLP contours indicate a distinct anticyclone to the southeast of the cyclone

center, as was observed in the case study. Surface evaporation is particularly high within and to the south of this anticyclone, as warm, subsaturated air subsides over the warm ocean surface, promoting enhanced evaporation (Boutle et al., 2010). The cyclone-relative winds, however, indicate that the flow in this region is directed away from the cyclone center toward the south and southwest. The converging motion expected during the intensification phase is instead observed to the north of the cyclone center, and will promote the ascending motion. As the cyclones mature, the cyclone-relative winds become more cyclonic,

linked to the cyclones propagating more slowly and becoming deeper. At this stage, evaporation to the south of the cyclones still cannot contribute to the cyclones' precipitation, because the winds still have no northerly component.

Closer to the cyclone center, strong surface evaporation is also evident. One pronounced region of elevated evaporation appears southwest of the developing cyclone, co-located with the cyclone's cold sector. As shown by Dacre et al. (2020), the strong upward latent heat fluxes observed behind the cyclone's cold front are a result of sharp temperature and specific humidity

gradients between the ocean surface and the overlaying air, with notably cooler and drier air at 10 meter. This environment allows cold, dry air that is descending behind the cold front – commonly referred to as the dry intrusion airstream (Browning 1990; Browning 1997) – to efficiently take up moisture that has evaporated from the ocean surface (Aemisegger and Papritz, 2018; Ilotoviz et al., 2021). According to Boutle et al. (2010), horizontal divergence induced by boundary-layer drag transports the moisture away from this region, maintaining the saturation deficit in the cold sector. However, the moisture evaporating

behind the cold front is not transported to the warm sector or into the WCB, as the presence of strong horizontal divergence in the cyclone-relative winds, together with subsidence in the cold sector, acts as a barrier to vertical ascent and transport across the front. Instead, much of this moisture rains out further from the cyclone center or remains in the boundary layer and can be utilized by another cyclone, assuming that the subsequent system propagates over the moist boundary layer. This mechanism suggests a potential moisture preconditioning effect, wherein surface evaporation in one cyclone's cold sector influences

moisture availability for subsequent cyclones following a similar path.

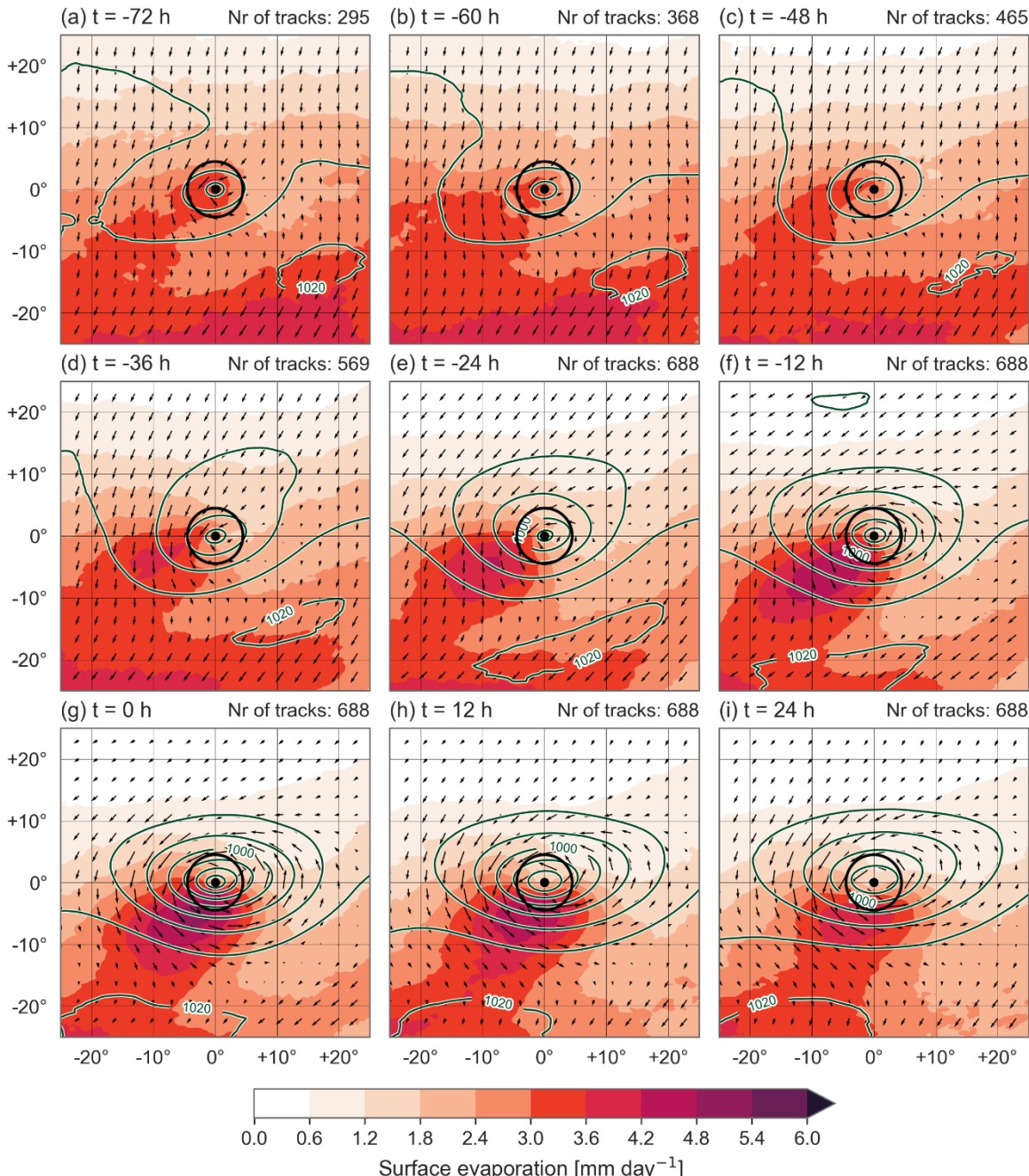

**Figure 11: Cyclone-relative composites of the surface evaporation (in mm day⁻¹) of all cyclones in the North Atlantic, for t = -72 h to t = 24 h in 12-hourly intervals. Surface evaporation is averaged over 12-hourly intervals centered on the given relative time t. Vectors of the cyclone-relative winds at 900 hPa are also shown, defined as the actual 900-hPa winds minus the 12-hour average cyclone propagation speed. Additionally shown are MSLP (in intervals of 5 hPa; dark green contours), the cyclone center location (black dot), and surrounding circle with a radius of 500 km (black circle). The number of cyclones contributing to the composite is indicated in the top right of each panel.**

# 5 Conclusions

## 5.1 Synthesis

In this study, we have analysed summertime extratropical cyclones, since they have received less attention than their winter counterparts, leaving key aspects of the cyclone-related water cycle poorly understood. To address this gap, we studied the precipitation distribution and identified precipitating air parcels within 500 km from the cyclone center for both a case study of a representative cyclone and a climatology of 688 deep summertime cyclones. By computing backward trajectories and analysing changes in specific humidity along these trajectories, we were able to pinpoint the moisture sources, characterize the uptake environment and understand moisture transport pathways. Below, answers per research question are given.

Q1:     What is the spatial distribution of precipitation within and around the cyclone center?

The bulk of the precipitation associated with summertime cyclones falls close to the cyclone center beneath the WCB, mainly during the intensification phase. This precipitation is driven by vertical motions, frontogenetic dynamics and thermodynamic (local) convection. Moreover, cyclones in the Gulf Stream region exhibit the highest precipitation rates throughout their entire life cycle. This can be attributed to higher surface evaporation over a relatively warmer ocean surface in this region, which significantly moistens the boundary layer and supplies the cyclones with more moisture than in the other subregions.

Q2:     What are the geographical moisture sources of the precipitating air parcels?

From a geographical perspective, three moisture source regions are identified: (1) the warm side of the Gulf Stream Front, where evaporation from the ocean surface is particularly strong; (2) continental sources, which are especially relevant for cyclones in the Labrador Sea; and (3) a south-eastward extension of the moisture source footprint into the subtropics, likely reflecting the influence of cyclones with tropical origins that are present in the East Atlantic and Gulf Stream region. Another region of high surface evaporation, though not necessarily contributing to the precipitation in the cyclone center itself, is within the cold sector of the cyclone. From a cyclone-relative perspective, this region is located to the southwest of the cyclone center. Here, significant moisture accumulates within the boundary and is transported away by horizontal divergence where it eventually rains out further from the cyclone center or remains in the boundary layer. In the latter case, the moisture could potentially supply another cyclone that propagates through the region. Our case study confirms this process, showing clear evidence of moisture being handed over from one cyclone to the next.

Q3:     In which dynamical environment do moisture uptakes take place?

Moisture uptake occurs through two main mechanisms over the ocean. First, there is flow across the Gulf Stream Front, where the strong SST gradient induces intense ocean evaporation. This occurs in regions where the sea-air potential temperature difference is positive but situated near a sharp transition to negative values. Second, moisture uptake occurs in regions of cold-air advection, such as the cold sector of a cyclone. In these environments, near-surface temperatures are lower, but the

underlying ocean remains relatively warm, creating a strong temperature contrast that drives intense upward latent heat fluxes. In addition to these oceanic sources, continental moisture uptake also plays an important role in summer, when land evapotranspiration contributes to cyclone-related precipitation.


Q4:      How do the uptake and moisture transport characteristics change throughout the life cycle of a cyclone?

During the early stages of cyclone development, moisture originates from more remote sources located on the warm side of the Gulf Stream Front. Meanwhile, moisture uptakes associated with cold-air advection are responsible for cyclone precipitation throughout the entire life cycle. As revealed by the case study, this occurs when cold air masses from the north
move equatorward along the surface and interact with the ocean, or when dry air masses descend behind the cold front of either the primary or secondary cyclone. These moisture sources, located closer to the cyclone center, become more dominant as cyclones generate less precipitation and start to decay. Consequently, the distances between the moisture sources and precipitation locations are highest during the intensification phase and decrease throughout the cyclone life cycle. Despite this shift in source regions, the atmospheric residence time of moisture remains relatively constant, at about four days.

**5.2 Discussion and final thoughts**

Previous studies, such as Papritz et al. (2021), have extensively analysed moisture sources and uptake characteristics for North Atlantic deep cyclones in winter. During this season, the stronger equator-to-pole temperature gradient intensifies the jet stream, resulting in cyclones with higher wind speeds and more intense vertical motions within the WCB. According to Papritz et al. (2021), cyclone-related precipitation in winter is mainly fuelled by strong ocean evaporation, especially when cold air is
advected over a relatively warm ocean surface. In summer, however, the atmosphere is warmer, reducing the temperature contrast that drives strong evaporation. In addition, cyclones develop in a weaker baroclinic environment (Chang and Song, 2006). Nevertheless, evaporation over very warm SSTs can still provide substantial moisture to the warmer summer atmosphere, resulting in high specific humidity values. These seasonal differences motivated us to study how the cyclone-related water cycle is shaped in summer.

Regarding the moisture sources of the precipitating water, we found that the Gulf Stream region and the Gulf Stream Front serve as robust moisture sources in both summer and winter, underscoring their importance. However, despite this and other similarities, there are also notable seasonal differences in moisture sources and transport pathways. In summer, continental moisture sources become more relevant, especially for cyclones developing in the Labrador Sea. In addition, subtropical moisture sources appear more frequently than in winter, especially during the intensification phase, and contribute to more
long-range moisture transport into the storm system. Yet they remain secondary to midlatitude sources, and we have not found any sources deep in the (sub)tropics. Winter cyclones, on the other hand, tend to rely more on local moisture sources, with a higher contribution from short-distance evaporation regions. The mean atmospheric residence time of moisture is therefore longer in summer, averaging about four days compared to three days in winter (Papritz et al., 2021; van der Ent and Tuinenburg, 2017). While the relatively greater contribution of remote sources during summer months does contribute to the observed

longer residence time, the difference cannot be explained by source distance alone. It is plausible that in summer weaker atmospheric moisture convergence, reduced cyclone strength, and the increased water vapour holding capacity of a warmer atmosphere, together delay saturation as well.

    Despite the larger role of remote sources in summer, (sub)tropical moisture contributes only marginally to cyclone precipitation. This contradicts the common assumption that (sub)tropical moisture is an important contributor to precipitation
associated with extratropical cyclones. The assumption likely arises from two factors: first, moisture availability is highest over subtropical oceans, where evaporation in summer exceeds precipitation by up to 4 mm day$^{-1}$ (Kållberg et al., 2005; Knippertz and Wernli, 2010). Secondly, ARs (or moisture transport axes in general) typically run parallel to the cold front, directed from the subtropics toward heavy precipitation regions in the extratropics (Dacre and Clark, 2025). Taken together, these factors would suggest a pathway for direct transport, that is especially evident in late summer when tropical cyclones
undergoing extratropical transition are expected to follow this pathway and enhance (sub)tropical moisture import. Yet our results indicate that midlatitude precipitation associated with deep cyclones is predominantly caused by moisture originating from the same latitude band. This finding aligns with results for winter by Papritz et al. (2021) and other studies (e.g. Coll-Hidalgo et al., 2025; Sodemann and Stohl, 2013). As explained by Dacre and Clark (2025), a potential explanation for the midlatitude sources is that AR features can move faster than the air within them, requiring continual local moisture
replenishment to maintain high water vapour at their leading edge. At the same time, it is important to acknowledge that the estimated contribution of (sub)tropical sources depends on how the precipitation target is defined. In this study, we use a radius approach. Coll-Hidalgo et al. (2025) found that using alternative precipitation-targeting methods the moisture source footprint is extended farther south into the subtropics. This finding underscores a broader point that different feature-identification frameworks emphasize different aspects of the same storm system, even though many features share similar generation
mechanisms. Consequently, this can result in different outcomes. Coll-Hidalgo et al. (2025) already explored this sensitivity; however, extending this research with systematic approaches that also capture more general weather features, such as ARs and fronts, using the methodology of Konstali et al. (2024), would further clarify each feature's contribution to precipitation in summer cyclones. A meaningful comparison can for example be made between precipitation attributed to fronts and that within the WCB footprint, as both are generated by the same underlying mechanisms.

Another seasonal distinction is the role of cyclone clusters in moisture transport. In winter, moisture sources are often associated with a preceding cyclone, with moisture uptake occurring within the cold sector of the primary cyclone or within the interaction zone between the cyclone and an anticyclone (Papritz et al., 2021). In our analysis for summer, we presented a case study that illustrates that such cyclone clusters also occur in summer. In the composites of all 688 cyclones, however, we could not identify a primary cyclone in the MSLP contours, nor a region of enhanced surface evaporation to the northeast. This
suggest that there are fewer cyclone clusters in our subset. This is in agreement with Weijenborg and Spengler (2024), who found that cyclone clusters are less common in summer compared to winter.

    Weijenborg and Spengler (2024) distinguish two types of cyclone clusters: a 'Bjerknes' type and a stagnant type. The former is in line with the original idea of Bjerknes and Solberg (1922), where several successive cyclones follow a similar path. These

systems are more common in winter and are more likely to appear in composite analyses. The latter, stagnant cyclone clusters, remain nearly stationary throughout their life cycle. Consequently, our inability to detect a preceding cyclone in the summer composite (Fig. 11) may be due to both the reduced frequency of 'Bjerknes' cyclone clusters and the difficulty of compositing stagnant systems due to inconsistent relative positioning between cyclones. Future research could address this by distinguishing between primary, secondary, or individual cyclones, and analysing the cyclone-related water cycle for each group separately. A methodology for this is proposed by Priestley et al. (2017, 2020), which applies a frontal identification algorithm and checks whether detected cyclones form along the trailing cold front of a previous cyclone.

Once moisture reaches the cyclone center, it contributes to precipitation through convergence at the surface and subsequent ascending motion in the warm sector and within the WCB. These processes are similar in both summer and winter cyclones, but the strength of vertical motions and the role of moisture availability differ. In winter, stronger baroclinicity enhances vertical ascent in the WCB, lifting the moisture all the way into the free troposphere. In contrast, our analysis shows that the air parcels responsible for precipitation in the 688 summer cyclones cover less vertical distance, as weaker baroclinicity results in reduced vertical uplift within the WCB. However, increased moisture inflow, driven by the Clausius-Clapeyron relationship, compensates for this by ensuring that even with less uplift, the higher moisture content in warmer air parcels results in similar precipitation to winter extratropical cyclones once saturation is reached.

Both our case study and climatological analysis suggest that moisture uptake occurs in two distinct dynamical environments. The first is flow across the Gulf Stream Front, associated with strong evaporation. The second involves cold-air advection, in which relatively colder air from the north acquires moisture within the cyclone's cold sector. In our case study, we observed that this mechanism can serve as a moisture source for precipitation in a subsequent cyclone. From a climatological perspective, however, this behaviour is more common in winter. As for precipitation, the WCB remains a key mechanism, although its intensity is weaker in summer. However, this reduced intensity is offset by an increase in atmospheric moisture transport. These findings highlight not only the seasonal differences in cyclone dynamics but also the necessity of studying summer cyclones to improve our understanding of the global water cycle. Moreover, the poleward shift of the storm track in summer, which increases the frequency of cyclones reaching western and northern Europe (Mesquita et al., 2008), is projected to intensify under climate change. Priestley and Catto (2022) suggest that this shift will extend further into Europe, affecting both summer and winter cyclones. This raises the possibility that future climate conditions may lead to summer-like cyclone behaviour becoming more prevalent throughout the year.

## Appendix A

Appendix A consists of additional figures that complement the methodology and results section of this study.

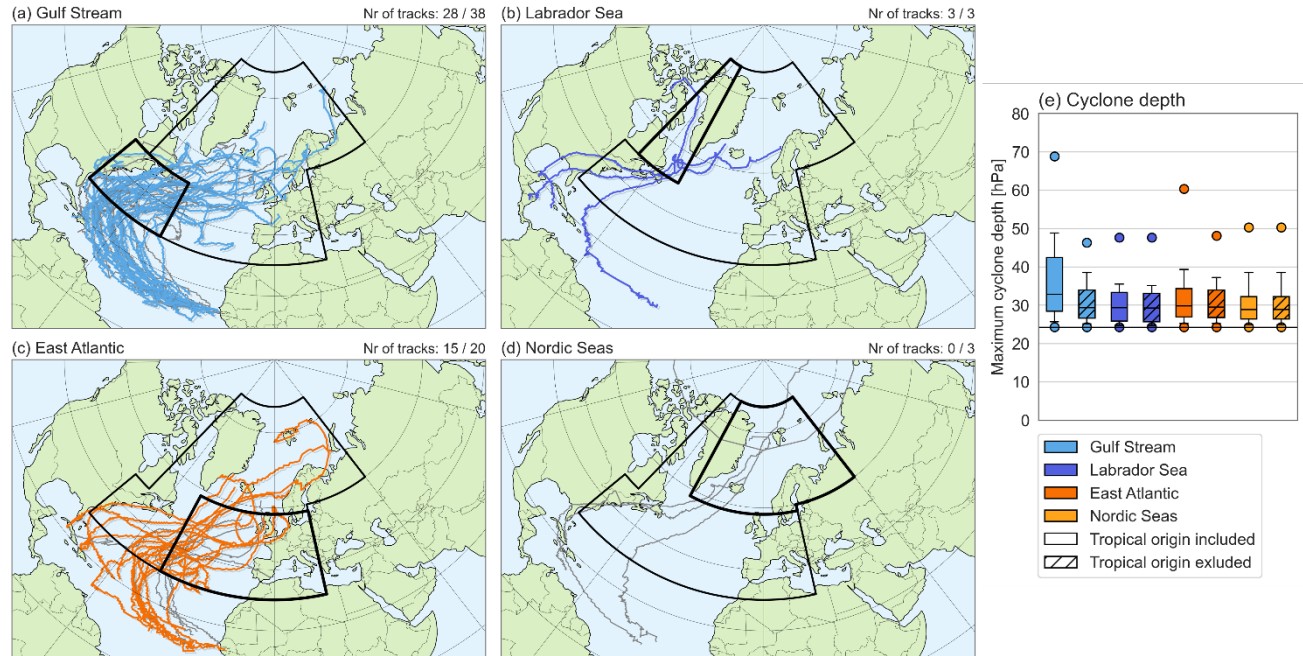

**Figure A1: Selected tracks with tropical origin in (a) the Gulf Stream region, (b) the Labrador Sea, (c) the East Atlantic Ocean, and (d) the Nordic Seas. The selected tracks are highlighted in colour and meet the three criteria of duration, intensity and location. At the top right of each panel, the number of cyclone tracks that have been selected are given as the proportion relative to the total of tracks with tropical origin within each of the four subregions. (e) The distribution of the maximum depth of the selected cyclones in the four subregions when the ones with tropical origin are included (final subset; solid boxes) and excluded (hatched boxes). The black horizontal line represents the threshold of the 80th percentile of all tracks. In addition, the whiskers indicate the 10th-90th percentile range, and the dots show maxima and minima.**

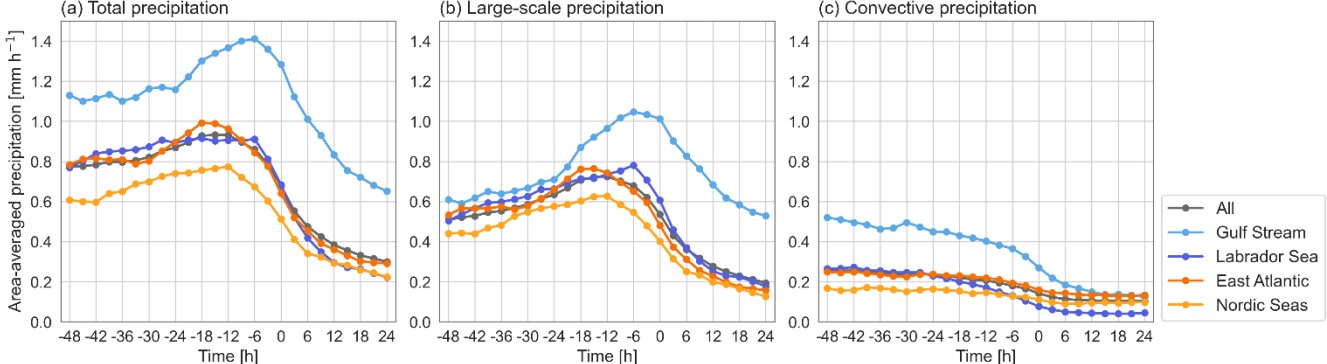

**Figure A2: Area-averaged (a) total precipitation, (b) large-scale precipitation, and (c) convective precipitation (all in mm h⁻¹) within the time interval t = −24 hours to t = 24 hours. The grey line is the mean for all tracks. The coloured lines are the means for each subregion.**

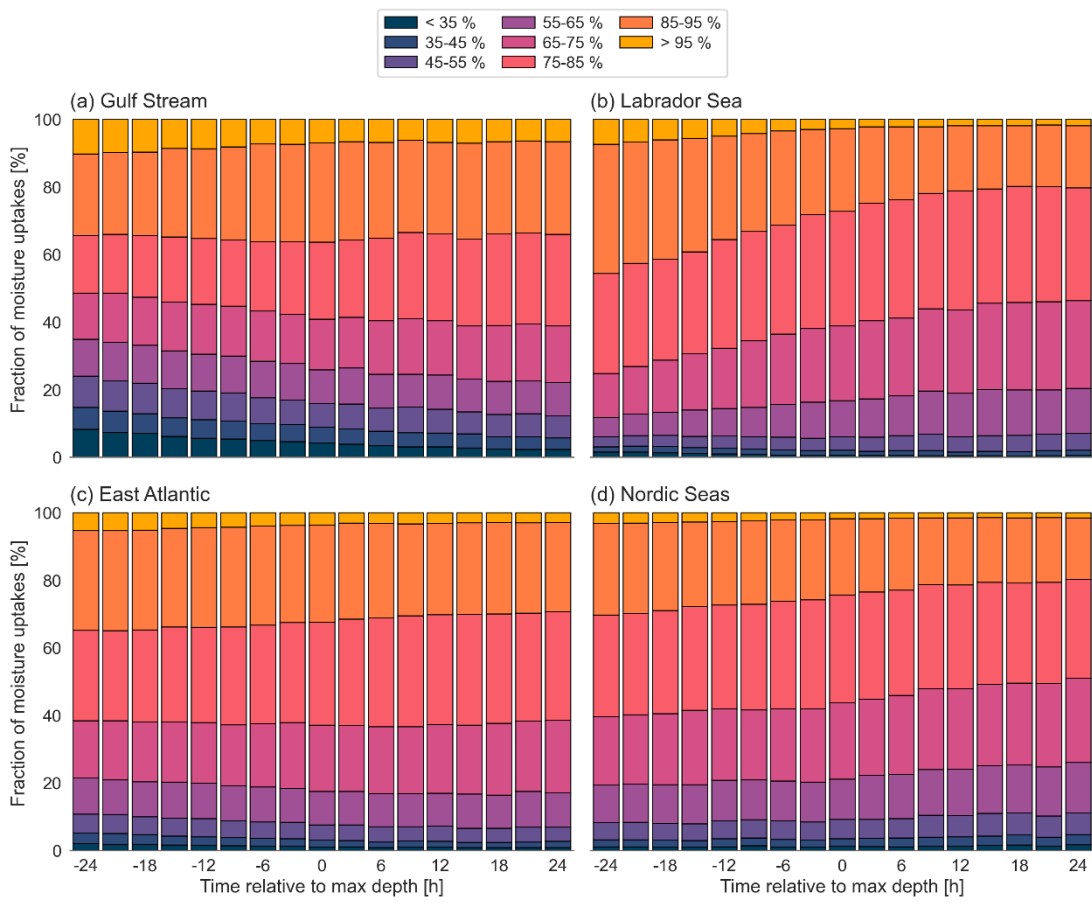

**Figure A3: Uptake contributions according to the fraction of precipitation that has been assigned to a source region, at t = -24 h to t = 24 h in 3-hourly intervals, associated with cyclones in (a) the Gulf Stream, (b) the Labrador Sea, (c) the East Atlantic, and (d) in the Nordic Seas.**

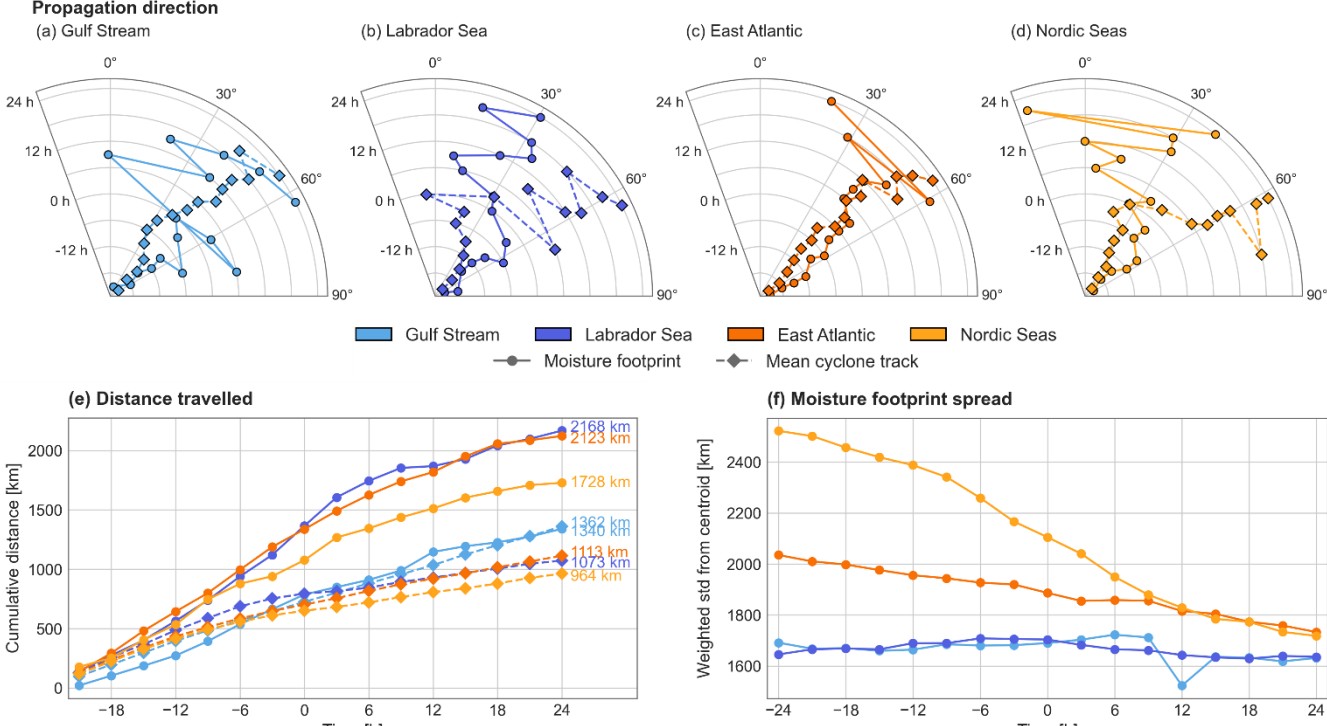

**Figure A4: Movement of the moisture footprint and the mean cyclone track, characterized in terms of (a)-(d) propagation direction, (e) distance travelled, and (f) moisture footprint spread, which is defined as the weighted mean standard deviation of the source latitude and longitude positions. The coordinates of the moisture footprint are found by calculating the spherical center of mass of the mean moisture footprints. We then used the great-circle distance to track its movement and compared it with that of the mean cyclone center.**

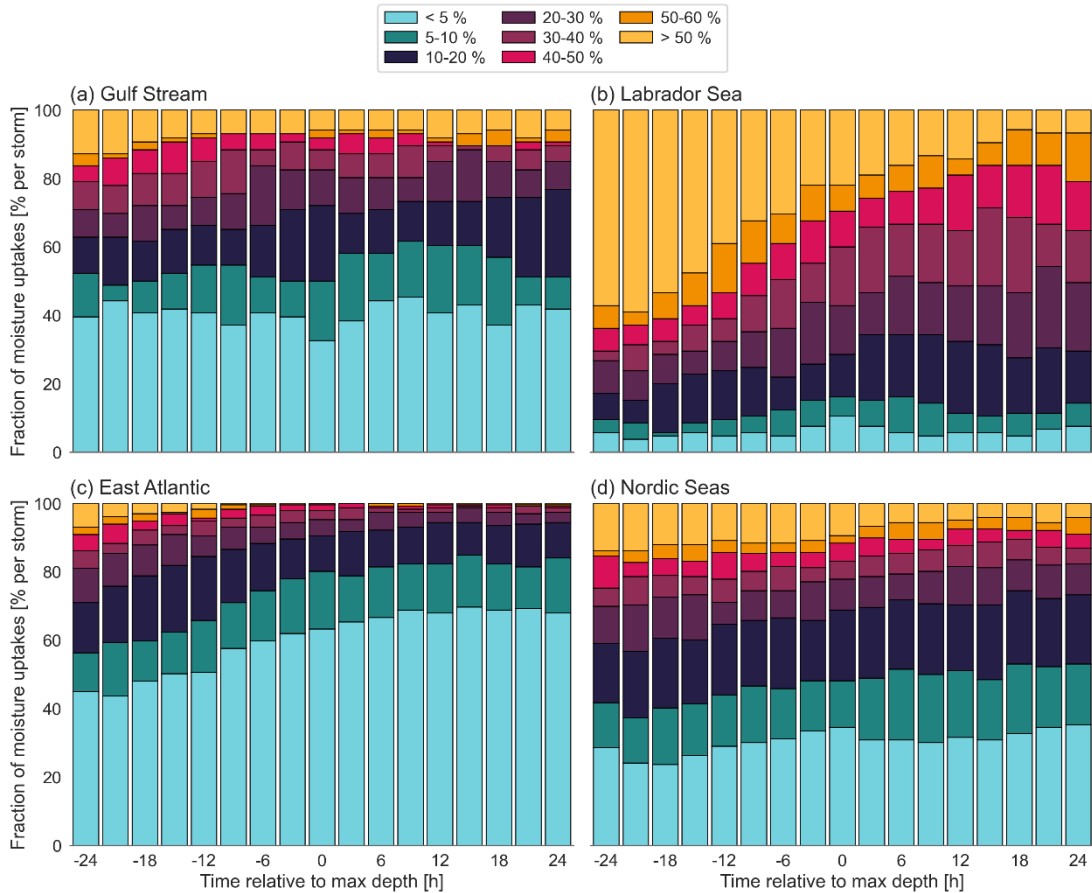

**Figure A5: Uptake contributions according to the fraction of uptakes that have occurred over land, at t = -24 h to t = 24 h in 3-hourly intervals, associated with cyclones in (a) the Gulf Stream, (b) the Labrador Sea, (c) the East Atlantic, and (d) in the Nordic Seas.**

*Data availability.* The ERA5 reanalysis data are publicly available and can be downloaded from the Copernicus Climate Change Service Data Store (https://climate.copernicus.eu/climate-reanalysis). The code used for the analysis will be made available on GitHub upon publishing [*link will be added*].

*Author contributions.* The main research question was conceptualized by LP and RS. The design and scope of the study were then designed by CW, IB and RS. RS performed the analysis and produced the figures, under the guidance of CW and IB. RS wrote the manuscript, with valuable feedback from FS, LP, CW and IB.

*Competing interests.* One of the co-authors is a member of the editorial board of Weather and Climate Dynamics. The authors
declare that they have no other conflict of interest.

*Acknowledgements.* The authors would like to thank Michael Sprenger for providing the data set of cyclone tracks. We also thank Dim Coumou, Vera Melinda Galfi and Hylke de Vries for some fruitful discussions, which helped to improve this work. This study was partly funded by the European Union Horizon Europe research and innovation programme under Grant Agreement 101137656 (EXPECT project).

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
