# Peer review of "Precipitation, Moisture Sources and Transport Pathways associated with Summertime North Atlantic Deep Cyclones"

_EGUsphere, 2025_

## Referee Comment (RC1)

Precipitation, Moisture Sources and Transport Pathways associated with Summertime North Atlantic
Deep Cyclones
egusphere-2025-1752

This paper aims to evaluate the moisture sources for summertime extratropical cyclones. It employs a Lagrangian back trajectory method to determine the sources and sinks of moisture for parcels that result in precipitation at the centre of cyclones. Overall, the paper is well written, the analysis interesting and conclusions a valuable contribution to the literature. I have made quite a lot of specific comments, but they are largely points of clarification rather than extensive requests for further analysis. I recommend that this paper would be suitable for publication in WCD if the points below are addressed.

General comments
1. Throughout the paper the authors refer to moisture being 'transported from remote regions to the centre of the cyclone' or the 'delivery' of moisture to the cyclone. These phrases imply that the cyclone is stationary, and that moisture is transported towards it. How good is the approximation of a stationary cyclone? If the cyclone is not stationary, then moisture may remain 'local' to the cyclone centre, but still undergo 'long-range transport' as it moves with the cyclone. Please could the authors be more specific with their phrasing and define what they mean by local and remote.
2. Q2 states that cyclone-relative moisture sources will be determined. While cyclone-relative evaporation is shown (figure 11) I could not find any cyclone relative moisture source analysis in the paper. In a cyclone-relative frame of reference, it would be necessary to subtract the cyclone motion from that of the Lagrangian trajectories. However, since many of the cyclones will not exist for 8-days I'm not sure how this would be performed. As noted in comment 56, simply showing cyclone-relative streamlines or wind vectors overlaid in figure 11 would go some way to answering the question, but as it is, I do not think the authors have answered part of Q2.

Specific comments
3. Line 14: The data source should be referred to here, i.e. ERA5
4. Line 17: Do the authors mean 'beneath the warm conveyor belt', rather than 'in the WCB' or are they suggesting that some precipitation associated with the WCB evaporates within the airflow?
5. Line 24: It is not clear what the authors mean by remote and localised sources. Are they referring to distance relative to the cyclone position at some fixed point along its track, or relative to a moving cyclone?
6. Line 32: The authors refer to moisture 'coming from' specific source regions. This suggests somehow that the atmosphere is stationary, and that the moisture is transported through that stationary atmosphere. Perhaps 'originate' would be more specific term to use?
7. Line 38: There are many earlier papers that describe the role of extratropical cyclones in redistributing energy, moisture and momentum. Please refer to original sources where possible.
8. Line 63: This sentence seems strangely worded to me since it implies that the convergence on the warm side of the cold front is coincidentally co-located with the WCB. In reality, the cross-front ageostrophic circulation causing convergence near the surface increases frontogenesis along the cold front strengthening the horizontal part of the WCB. The convergence also leads to ascent strengthening the vertical part of the WCB. Thus, there is no coincidence but direct cause and effect (see Dacre and Clark, 2025)
Dacre, H.F. and Clark, P.A., 2025. A kinematic analysis of extratropical cyclones, warm conveyor belts and atmospheric rivers. *npj Climate and Atmospheric Science*, *8*(1), p.97.
9. Line 65: Here it is stated that large-scale vertical motion lifts the WCB above the warm front. What is causing the large-scale vertical motion? Are the authors referring to divergence aloft downstream of a planetary-scale Rossby wave or some other mechanism?
10. Line 71: I do not see follow the logic here. The fact that WCB ascend from the boundary layer to the upper-troposphere in 2-days does not imply that they transport more moisture to the upper troposphere than shallow convective processes, or that they are the main sink of precipitable water in the atmosphere. Please add more details to explain this sentence.
11. Line 74: The WCB is generated by the same mechanisms that are responsible for the formation of extratropical cyclone fronts, therefore they are 'intrinsically linked' not 'often linked'.

12. Line 78: If you are saying that moisture is 'transported from remote regions to the centre of the cyclone' then are you assuming that the cyclone is stationary? If not, you need to consider that the moisture may have remained local to the centre of the cyclone, but that it moved with the cyclone as it travelled.

13. Line 81: ARs are generated by the same processes that create the WCB airflow, thus they do not exist without cyclone airflows. The apparent distinction is merely a matter of choice of threshold and the different methods used to identify them.

14. Line 142: Are there any limitations to using ERA5 precipitation in this study?

15. Line 183: Over what period is the intensification measured? 24 hours or the entire cyclone lifecycle?

16. Line 196: In the figure caption what does the maximum depth refer to? Is this the maximum deepening in 24 hours? Or the total deepening over the cyclone lifecycle?

17. Line 212: Frontal circulations occur within 500km of the cyclone centre, particularly in the developing stages before frontal fracture occurs.

18. Line 214: Without observations, how do you know that the precipitation is allocated to the 'right' moisture sources?

19. Line 227: Is the first timestep, the first 3 hours?

20. Line 233: Is the start point of the trajectory the cyclone centre, or the location of the trajectory 8-days earlier?

21. Line 237: Is there sensitivity to this choice of threshold?

22. Line 274: Does the maximum depth refer to a deepening of 35.2 hPa over the entire cyclone period in which case is it a maximum or should it be a total? Or is it the maximum deepening over a 24-hour time window, in which case the time period should be included.

23. Line 293: What is meant by 'keep the moisture in place'? Typically, the winds near cyclones are relatively strong.

24. Line 295: The authors might be interested in reading Demirdjian et al. (2023) which describes this process very nicely.
Demirdjian, R., Doyle, J.D., Finocchio, P.M. and Reynolds, C.A., 2023. Preconditioning and intensification of upstream extratropical cyclones through surface fluxes. *Journal of the Atmospheric Sciences*, 80(6), pp.1499-1517.

25. Line 300: Figure 3 uses a colour scale that makes the fluxes on the warm side of the Gulf Stream appear to be positive. I had to look very carefully to see the 'sharp transition' from negative to positive. I suggest using a divergent red-blue colour scale instead.

26. Line 303: What is the surface evaporation elevated with respect to? No climatology of evaporation is shown so it is not clear if this is a climatological response or related to the passage of the cyclone.

27. Line 305: Figure 4 uses a colour scale in which it is difficult to observe the 'evaporation hotspots'. Perhaps anomalies or normalised anomalies could be shown? I.e. subtract the climatological field and divide by the standard deviation of the climatology at each location. Then it would be possible to see how large anomalies are relative to the climatological mean, i.e. 1 std deviation larger/smaller than climatological mean?

28. Line 318: No evidence is provided to support the statement that the dynamical flow is setup by the primary cyclone, or that significant moisture accumulates ahead of the developing cyclone.

29. Line 326: Here the authors refer to the sources of moisture moving eastward. Presumably this is related to the fact that the cyclone is also moving eastward?

30. Line 332: How are long-range and local defined. Do they refer to a distance travelled by the trajectories over the 8-day period or are they the distance relative to a fixed feature, or the moving cyclone?

31. Figure 5: It is useful to show the cyclone track, but I wonder if the position of the cyclone at 24hr intervals preceding the trajectory start time should be shown instead?

32. Line 351: I don't think that the trajectory figures 6a-c are very legible. It's difficult to identify the different pathways. If 80% of the moisture uptake occurs within 4 days of precipitation (line 515), would it be better to show shorter trajectories in figure 6 to make it more legible?

33. Line 354: How do you know the parcels were lifted by convection?

34. Line 355: If the parcels are lifted by convection, how do they converge at the surface?

35. Line 359, 370, 570: Here and elsewhere, the descending trajectories behind the cyclone cold front are known as the dry intrusion. It might be helpful to refer to this term to enable links with previous work.
36. Line 360, 578-9: Likewise, the air behind the primary cyclone cold front which is then 'fed into the secondary cyclone' is known as the feeder-airstream. It might be helpful to refer to this term to enable links with previous work.
37. Line 366: Use of the phrase 'came from' suggests air travelling towards the cyclone from the primary cyclone. Would a better description be that this air was swept up by the secondary cyclone as it travels into the region of enhanced moisture?
38. Line 410: You have not shown evidence that maximum vertical motion occurs before minimum MSLP? I think maximum precipitation typically occurs before maximum dynamical intensity due to limited moisture availability as cyclones travel polewards.
39. Line 29: Can you concentrate precipitation. Perhaps extending would be a better word to describe the shape of the precipitation pattern?
40. Line 438: The precipitation does not significantly drop for Gulf Stream cyclones since the moisture availability remains high, even in the decay phase. By contrast, precipitation drops significantly for the other regions due to reduced dynamical forcing and reduced moisture availability.
41. Figure 7: Much of the precipitation pattern in the decaying phase is outside the domain shown. Could a larger domain be used to capture the full shape of the cyclone precipitation pattern in this phase?
42. Line 469: The authors describe the eastwards shift in the moisture sources as 'slight'. This is a subjective word. Could they calculate the magnitude of the eastward shift? This could then be compared with the eastward shift in the mean cyclone position to see if they are comparable.
43. Line 469: The eastward shift is also evident for the Gulf Stream cyclones.
44. Line 474: Local to what?
45. Figure 8: Would it be possible to show the mean cyclone position at various stages in the cyclone lifecycle.
46. Line 495: Within 1000km of what?
47. Line 513: Would these terrestrial sources be the Great Lakes, or are they more widespread than this?
48. Line 526: Sources 1000-1500km away from what?
49. Line 539: Ascending trajectories ahead of the cold front are part of the WCB airflow. It might be helpful to refer to them as the ascending part of the WCB to enable links with previous work.
50. Line 540: What is the evidence for moisture accumulation near the cyclone centre?
51. Figure 11: These figures correspond well with figures 5 and 6 in Dacre et al. (2020), showing maximum latent heat fluxes behind the cyclone cold front as a result of large vertical gradients in specific humidity.
Dacre, H.F., Josey, S.A. and Grant, A.L., 2020. Extratropical-cyclone-induced sea surface temperature anomalies in the 2013–2014 winter. *Weather and Climate Dynamics*, *1*(1), pp.27-44.
52. Line 615: What about moisture evaporated in the subtropical high as referred to earlier in the paper?
53. Line 623: Here the authors refer to 'delivery' of moisture to the cyclone. Again, this implies the cyclone is stationary and that moisture is transported towards it from 'remote sources'. Is this really what is happening?
54. Line 630: Source distances from what?
55. Line 650: Dacre et al. (2023) use a very different method to that used here and estimate a mean moisture residence time of 36 hours. Do you think this is due to methodological differences or because they focus on extreme winter cyclones where the ascent rates are typically higher than for summer cyclones?
56. Line 657: You have not included a figure of cyclone-relative low-level winds so it is not possible to determine if a feeder airstream is present in your cyclone composites. Would it be possible to overlay cyclone-relative streamlines to see if the feeder airstream can be identified?
57. Line 671: This sentence suggests there are multiple ascending motions. Are the authors referring to frontal line convection as an alternative to the WCB airflow?
58. Line 676: Similar to what, the precipitation in winter cyclones?

59. Line 680: How are you separating fronts and WCB airflow?  They are generated by the same mechanism, so it seems strange to separate the precipitation associated with them.  Perhaps you are referring to the horizontal and vertical parts of the WCB?  Or to frontal line convection and WCB ascent?

Typographical errors
60.  Line 10: Why is the word 'extreme' in brackets?
61. Line 42: Why is 'heavy' in brackets?
62. Line 97: Here and elsewhere, it is more standard to refer to the singular 'precipitating water' rather than the plural 'precipitating waters'.
63. Line 101: 'How' should be 'whether' since you haven't determined this yet.
64. Line 164: You do not need cf.
65. Line 171: The number 2 needs units.
66. Line 216: I think interpolated would be a better description than 'traced'.

---

## Referee Comment (RC2)

**Review of "Precipitation, moisture sources and transport pathways associated with summertime North Atlantic deep cyclones" Stoffels et al. submitted to WCD**

This paper adopts a cyclone-centred perspective to evaluate the moisture sources and transport pathways of North Atlantic deep cyclone precipitation in summer. The paper strongly builds scientifically and methodologically on a previous paper (Papritz et al. 2021) focussing on winter deep North Atlantic cyclone precipitation. The paper is well written and the main findings are interesting and related to

   i)     moisture residence times being relatively constant of about 4 days throughout the cyclone life cycle;
   ii)    the moisture sources of cyclones originating from the tropics and making extratropical transition being mainly located in the subtropics and midlatitudes, with very limited amounts coming from the tropics directly
   iii)   contributions from different key geographical and cyclone-relative regions, such as land, the warm side of the Gulf Stream as well as evaporation from the cold sector of preceding cyclones, although the discussion mainly relates to geographic regions and not quantitatively to cyclone-relative regions.

I have a few minor comments mainly related to the writing and presentation of the results.

1) Innovation: I think the paper could become a bit sharper in terms of its innovative contributions to science. In my reading, I got the impression that it was very closely following the preceding paper by Papritz et al. 2021 both in terms of scientific focus and methodological approach. The fact that summer deep cyclones are generally less studied has good reasons, they are rarer and less intense than winter cyclones. Therefore, the motivation for this study could be carved out a bit more convincingly. I do think there are good reasons to investigate summer cyclones separately, e.g. to investigate the dynamical impact of added moisture from the land sources, the role and moisture transport pathways related to cyclones from tropical origin, potential similarities with future warmer conditions with weaker baroclinicity also in winter, contrasts between cyclones over the ocean vs. over land… I encourage the authors to make a stronger case for their paper in the abstract and the introduction (e.g. at L. 51-55). The fact that summer cyclones are less studied does not make it a good reason to study them.
2) L. 62: WCBs here I think Madonna et al. 2014 and Heitmann et al. 2024 should be referenced. And at L. 73 about the link between WCBs and cyclones: Binder et al. 2016.
3) WCBs: throughout the paper the authors should be much more cautious with their definition of the WCB and clearly define what they mean. Usually, this airstream is defined as ascending by 600 hPa or more in 48 h. It is very likely that in summer the airstreams are ascending less (see also the substantially lower frequency of WCBs in summer over the North Atlantic). Also, the convective parts of the ascent are probably missed in the approach chosen by the authors (Oertel et al. 2021) calculating the trajectories based on 3 hourly 3D wind fields on the still relatively coarse ERA5 grid.
4) L. 76-87: I don't understand the use of discussing atmospheric rivers in such great detail since they are not identified or discussed further in the results of this paper. I would suggest shortening and shifting the discussion on the potential role and link with ARs to the conclusion.
5) Q2 is not addressed in a cyclone-relative way. I do think that the authors would have the necessary data and tools to address this question, with a bit more coding work and gridding the uptakes for different times relative to cyclone maximum depth.
6) L. L117: Here the Lagrangian method used should also be referenced (Sodemann et al. 2008).
7) L. 120: these are not adequate references for the use of Lagrangian moisture source identification to distinguish between different air streams. For example Pfahl et al. 2014 could be a good option.

8) L. 123: if the North Atlantic was studied several times before, then mention several studies. Here maybe e.g Gimeno et al. 2012, Perez-Alarcon et al. 2022 could be good options.
9) L. 160: "Thereafter a new time axis … is defined", I think it's not a new axis, just a new time of reference.
10) L. 168-170: Why did you exclude these cyclones? Some of them might also have made extratropical transition and actually be quite interesting to study in more detail.
11) L. 181: "exhibit strong movement" what does this mean exactly?
12) Fig. 1: add contours of track density to help interpret Fig. 8.
13) L. 202: add Wernli and Davies 1997 for LAGRANTO to reference the original publication as well as.
14) L. 212: justify the 8-days based on studies about the moisture residence time in this region and season.
15) L. 214: "LAGRANTO's ability to allocate a significant portion of precipitation to the right moisture sources" not sure I understand what you mean here. Do you mean Watersip instead of LAGRANTO? And how do you know what the "right sources" are?
16) L. 219: Did you use the official Watersip code? Or an own implementation in which case it would be clearer to simply reference the original publication of the algorithm with Sodemann et al. 2008 and not call the algorithm Watersip. For reproducibility it would be easiest if you provided a link to the code used.
17) L. 239: in the 48 h around maximum cyclone depth.
18) L. 255: at arrival in the cyclone (trajectory start is confusing).
19) L. 370: I would not call this the upper troposphere, there are very few trajectories coming from above 500 hPa.
20) L. 376: implying that local moisture recycling is becoming important: what does that mean exactly?
21) L. 415-416: here Aemisegger and Papritz, 2018 would be more fitting.
22) L. 453: It's not clear what the initial moisture is: the diagnosed precipitation?
23) L . 465: I think this is really an important point of this paper. It should be emphasised more. This contradicts the usual assumption that in cyclones from the tropics making extratropical transition, subtropical or even tropical moisture gets exported into the midlatitudes.
24) L. 469-472: this raises the question of cyclone-relative sources, which is not addressed quantitatively in this paper.
25) L. 474: Here "local" needs to be defined more quantitatively. Local relative to the cyclone?
26) Fig. 9b: I think panel b does not make much sense given the relatively large North-South temperature gradient.
27) Fig. 9d: make clear that panel d is based on the explained fraction.
28) L. 526: "…hinting at cold-air advection" I think this is speculative.
29) L. 545: then I would say they are not WCB trajectories. What is the role of convection and the relatively coarse spatial and temporal resolution of the ERA5 data?
30) Section 4.5: the discussion in this section is a bit speculative without cyclone-relative analysis of the moisture sources.
31) L. 570: "… facilitating strong upward latent heat fluxes": yes and what matters even more for your study, is that given that the subsiding air is dry, it's efficiency in taking up humidity is large (Aemisegger and Papritz 2018).
32) L. 571: here maybe Illotovitz et al. 2021 about the impact of the dry intrusion on boundary layer dynamics and surface fluxes would be a good reference.
33) L. 657: The fact that you do not find a feeder airstream is due to the fact that you do not look at the moisture sources in a cyclone-relative perspective. Without such a cyclone-relative analysis (which I do think is feasible and which I would strongly recommend), you cannot really make this statement. If you really do not find a "feeder airstream" in your summer cyclone based on such an analysis this would reveal an interesting contrast and rise new questions about the cyclone-relative moisture cycling in summer vs. winter cyclones.
34) L. 688: "… or from the developing cyclone's own cold sector (which appears to be more important in summer)", this sounds contradictory with the explanation at L. 573ff, where the authors discuss that this moisture recycling pathway is unlikely due to the necessary cross-cold front motion of the air parcels that would rain out in the same cyclone. I recommend to rephrase this according to the statements made earlier in the paper.

**References**:

Aemisegger, F., and L. Papritz, 2018: A Climatology of Strong Large-Scale Ocean Evaporation Events. Part I: Identification, Global Distribution, and Associated Climate Conditions. *J. Climate*, **31**, 7287–7312, https://doi.org/10.1175/JCLI-D-17-0591.1.

Binder, H., M. Boettcher, H. Joos, and H. Wernli, 2016: The Role of Warm Conveyor Belts for the Intensification of Extratropical Cyclones in Northern Hemisphere Winter. *J. Atmos. Sci.*, **73**, 3997–4020, https://doi.org/10.1175/JAS-D-15-0302.1.

Heitmann, K., Sprenger, M., Binder, H., Wernli, H., and Joos, H., 2024: Warm conveyor belt characteristics and impacts along the life cycle of extratropical cyclones: case studies and climatological analysis based on ERA5, Weather Clim. Dynam., 5, 537–557, https://doi.org/10.5194/wcd-5-537-2024.

Ilotoviz, E., Ghate,V. P., and Raveh-Rubin, S., 2021: The impact of slantwise descending dry intrusions on the marine boundary layer and air-sea interface over the ARM Eastern North Atlantic site, *J. Geophys. Res.*, 126, e2020JD033879, https://doi.org/10.1029/2020JD033879.

Madonna, E., H. Wernli, H. Joos, and O. Martius, 2014: Warm Conveyor Belts in the ERA-Interim Dataset (1979–2010). Part I: Climatology and Potential Vorticity Evolution. *J. Climate*, **27**, 3–26, https://doi.org/10.1175/JCLI-D-12-00720.1.

Oertel, A., Sprenger, M., Joos, H., Boettcher, M., Konow, H., Hagen, M., and Wernli, H., 2021: Observations and simulation of intense convection embedded in a warm conveyor belt – how ambient vertical wind shear determines the dynamical impact, Weather Clim. Dynam., 2, 89–110, https://doi.org/10.5194/wcd-2-89-2021.

Pfahl, S., E. Madonna, M. Boettcher, H. Joos, and H. Wernli, 2014: Warm Conveyor Belts in the ERA-Interim Dataset (1979–2010). Part II: Moisture Origin and Relevance for Precipitation. *J. Climate*, **27**, 27–40, https://doi.org/10.1175/JCLI-D-13-00223.1.

---

## Author Comment (AC1)

Precipitation, Moisture Sources and Transport Pathways associated with Summertime North Atlantic Deep Cyclones

**Final response**

We thank the two reviewers for their insightful comments. In the following we outline how we will address their comments and for some questions already provide detailed answers. (Planned) changes to the manuscript are italicized.

Review 1

This paper aims to evaluate the moisture sources for summertime extratropical cyclones. It employs a Lagrangian back trajectory method to determine the sources and sinks of moisture for parcels that result in precipitation at the centre of cyclones. Overall, the paper is well written, the analysis interesting and conclusions a valuable contribution to the literature. I have made quite a lot of specific comments, but they are largely points of clarification rather than extensive requests for further analysis. I recommend that this paper would be suitable for publication in WCD if the points below are addressed.

Thank you very much for reviewing our paper and all your suggestions on how to improve the writing and clarify certain parts. We are glad to hear that you find the analysis interesting and the conclusions a valuable contribution to the literature! A more detailed answer per comment is given below.

General comments

1) Throughout the paper the authors refer to moisture being 'transported from remote regions to the centre of the cyclone' or the 'delivery' of moisture to the cyclone. These phrases imply that the cyclone is stationary, and that moisture is transported towards it. How good is the approximation of a stationary cyclone? If the cyclone is not stationary, then moisture may remain 'local' to the cyclone centre, but still undergo 'long-range transport' as it moves with the cyclone. Please could the authors be more specific with their phrasing and define what they mean by local and remote.

Thank you for pointing this out. We are happy to clarify this aspect of our paper. We agree that our phrasing may have unintentionally implied a stationary cyclone, which is not the case in our analysis. In fact, we explicitly account for the movement of cyclones by following them throughout their life cycle. Every three hours, we identify moisture sources relative to the current cyclone location and timing of precipitation within the cyclone center, rather than a fixed geographic point that remains the same throughout the life cycle.

However, once the air parcels are released from their starting positions, we only follow them eight days back in time, without tracking the cyclone's position during this eight-day period. Therefore, the *source distance* we refer to in the manuscript is the great-circle distance between the location where precipitation occurs (within the cyclone center) and the origin of the moisture, rather than the distance between the moisture origin and the cyclone center at the time of the uptake. In this context, the moisture sources and the way the moisture is transported is defined in a **geographical frame of reference**. A **cyclone-relative framework** is adopted when we discuss the spatial distribution of precipitation and surface evaporation around the cyclone center.

We agree that we can be more specific about the definition and the frame of reference when we state the numbers. In addition, as you mentioned in one of your specific comments, a short source distance does not necessarily imply short-range transport, because the air parcels may have moved with the cyclone as it travelled. Therefore, we agree that the terms "local" and "remote" can be misleading in this context. To avoid this confusion we will make the following changes in the manuscript:

1. To avoid implying that the cyclones are stationary we revised 'coming from' to 'originates' and rephrase sentences like 'transported from remote regions to the cyclone center', or 'delivery of moisture to the cyclone' and emphasise source origin relative to precipitation location when we quantify these distances.
2. We replaced or better defined terms like "local" and "remote".
3. Clarify that the cyclone is moving and that air parcels can undergo long-range transport even when the source distance is small.

2) Q2 states that cyclone-relative moisture sources will be determined. While cyclone-relative evaporation is shown (figure 11) I could not find any cyclone relative moisture source analysis in the paper. In a cyclone-relative frame of reference, it would be necessary to subtract the cyclone motion from that of the Lagrangian trajectories. However, since many of the cyclones will not exist for 8-days I'm not sure how this would be performed. As noted in comment 56, simply showing cyclone-relative streamlines or wind vectors overlaid in figure 11 would go some way to answering the question, but as it is, I do not think the authors have answered part of Q2.

We thank the reviewer for pointing this out and agree that our current Q2 is not fully answered with our analysis. Our current performed analysis, which is based on output that provides moisture uptake as an accumulated footprint over the eight-day period rather than at individual time steps, limits us to determine the cyclone-relative moisture sources. An alternative method, such as subtracting the cyclone motion from the Lagrangian trajectories to estimate relative source locations, is indeed not straightforward since many of the cyclones will not exist for eight days, as is correctly pointed out by the reviewer.

We will explore the following two options to align the analysis and research question:
1) Rephrase the Q2 to clarify that moisture sources are analysed in a **geographical frame of reference** only, while noting that the surface evaporation is examined in a **cyclone-relative frame**.
2) Add cyclone relative winds as this gives a first-order estimate of the direction of moisture transport. This will help assess whether regions of high surface evaporation (which are analysed in a cyclone-relative frame) are likely to contribute moisture to precipitation in the cyclone center, making them moisture sources.

Specific comments
3) Line 14: The data source should be referred to here, i.e. ERA5

Thank you for this comment. Line 14 has been revised as follows:

*For this purpose, 8-day backward trajectories are calculated for all air parcels in the vicinity of cyclone centers with ERA5 reanalysis data, for a subset of the most intense summertime cyclones over the North Atlantic.*

4) Line 17: Do the authors mean 'beneath the warm conveyor belt', rather than 'in the WCB' or are they suggesting that some precipitation associated with the WCB evaporates within the airflow?

Thank you for pointing this out. Our intention was to express that the bulk of the precipitation occurs near the cyclone center and is co-located with the WCB. We agree that the phrasing could be more precise and have revised the sentence to say "beneath the warm conveyor belt". We have revised Line 17, and made the same change in Line 595.

5) Line 24: It is not clear what the authors mean by remote and localised sources. Are they referring to distance relative to the cyclone position at some fixed point along its track, or relative to a moving cyclone?

We have revised this line in the manuscript to clarify what we mean. For a more detailed discussion, please see our reply to the first general comment above.

6) Line 32: The authors refer to moisture 'coming from' specific source regions. This suggests somehow that the atmosphere is stationary, and that the moisture is transported through that stationary atmosphere. Perhaps 'originate' would be more specific term to use?

We have revised this line in the manuscript to clarify what we mean. For a more detailed discussion, please see our reply to the first general comment above.

7) Line 38: There are many earlier papers that describe the role of extratropical cyclones in redistributing energy, moisture and momentum. Please refer to original sources where possible.

Thank you for this comment. Instead of Field and Wood (2007), Neu et al. (2013) and Papritz et al. (2021), we have now referred to Chang et al. (2002), Hartmann (1994) and Peixoto and Oort (1992):

*Extratropical cyclones make up an important component of the global atmospheric circulation by redistributing energy, moisture and momentum from lower latitudes to the polar regions (Chang et al., 2002; Hartmann, 1994; Peixoto and Oort,1992).*

8) Line 63: This sentence seems strangely worded to me since it implies that the convergence on the warm side of the cold front is coincidentally co-located with the WCB. In reality, the crossfront ageostrophic circulation causing convergence near the surface increases frontogenesis along the cold front strengthening the horizontal part of the WCB. The convergence also leads to ascent strengthening the vertical part of the WCB. Thus, there is no coincidence but direct cause and effect (see Dacre and Clark, 2025)

Dacre, H.F. and Clark, P.A., 2025. A kinematic analysis of extratropical cyclones, warm conveyor belts and atmospheric rivers. npj Climate and Atmospheric Science, 8(1), p.97.

Thank you for pointing us to the Dacre and Clark (2025) paper, which we were not aware of. We agree that our original phrasing may have suggested a coincidental co-location between low-level convergence and the WCB, rather than explaining the underlying relationship. We have revised the text in the introduction to reflect this cause-and-effect relationship more accurately and have incorporated a reference to Dacre and Clark (2025) to support this explanation.

9) Line 65: Here it is stated that large-scale vertical motion lifts the WCB above the warm front. What is causing the large-scale vertical motion? Are the authors referring to divergence aloft downstream of a planetary-scale Rossby wave or some other mechanism?

With 'large-scale vertical motion' we are referring to the ascending motion in the warm sector, which is driven by the cross-front ageostrophic circulation. As you also point out in the previous comment, this circulation leads to convergence near the surface on the warm side of the cold front, which strengthens the front and supports ascent. This is the same mechanism that fuels the WCB, eventually lifting it above the warm front. In the manuscript, we have revised this part of the introduction, emphasizing the underlying cause of vertical motion in the WCB. This revision is intended to clarify Line 65, without needing to link it to divergence aloft downstream of a planetary-scale Rossby wave or some other mechanism, as we also do not provide any evidence that supports this hypothesis.

10) Line 71: I do not see follow the logic here. The fact that WCB ascend from the boundary layer to the upper-troposphere in 2-days does not imply that they transport more moisture to the upper troposphere than shallow convective processes, or that they are the main sink of precipitable water in the atmosphere. Please add more details to explain this sentence.

Thank you for pointing this out. We agree that the original phrasing implied a causal relationship. Our intention was not to suggest that the rapid ascent of WCBs alone makes them the main sink of precipitable water. Rather, we wanted to highlight the efficiency with which WCBs transport moisture vertically from the boundary layer to the upper troposphere over relatively short timescales. We have revised this part of the introduction to clarify this point and to better reflect the role of WCBs in moisture transport, particularly in relation to their similarities with atmospheric rivers (see also our response to Comment 13).

11) Line 74: The WCB is generated by the same mechanisms that are responsible for the formation of extratropical cyclone fronts, therefore they are 'intrinsically linked' not 'often linked'.

Thank you for this suggestion. Line 73 has been revised as follows (also including comment 2 of reviewer 2):

*According to Eckhardt et al. (2004) and Pfahl et al. (2014), the WCB and extratropical cyclones are intrinsically linked. Furthermore, the WCB is capable of influencing the dynamics of the cyclones – a role that has been underscored in the context of cyclone intensification by Binder et al. (2016).*

12) Line 78: If you are saying that moisture is 'transported from remote regions to the centre of the cyclone' then are you assuming that the cyclone is stationary? If not, you need to consider that the moisture may have remained local to the centre of the cyclone, but that it moved with the cyclone as it travelled.

We have revised this line in the manuscript to clarify what we mean. For a more detailed discussion, please see our reply to the first general comment above.

13) Line 81: ARs are generated by the same processes that create the WCB airflow, thus they do not exist without cyclone airflows. The apparent distinction is merely a matter of choice of threshold and the different methods used to identify them.

Thank you for raising this point. We agree that the apparent distinction between WCBs and ARs largely stems from differences in how they are defined and identified. Therefore, we have revised this part of the introduction to clarify that WCBs and ARs are indeed generated by the same underlying mechanisms and often have spatial overlap. We now explicitly note that WCBs are defined as strongly ascending cyclone-relative airflows, while ARs are identified as Earth-relative moisture plumes.

*ARs are narrow filaments of high vertically integrated moisture that can act as a moisture supply for extratropical cyclones (Pfahl et al., 2014). Although WCBs and ARs are typically defined differently – WCBs as strongly ascending cyclone-relative airflows and ARs as Earth-relative moisture plumes (Dacre et al., 2019) – they are generated by the same mechanisms and spatial overlap often exists (Dacre and Clark, 2025; Knippertz et al., 2018).*

14) Line 142: Are there any limitations to using ERA5 precipitation in this study?

There are indeed some limitations in using ERA5 reanalysis data, including its coarse spatial resolution, the parameterization of convective processes and associated vertical motion, and an underestimation of intense, localized precipitation. In our study, this particularly affects the representation of diabatic processes, which typically occur at scales smaller than the model can resolve and are therefore parameterized (Binder et al., 2020). For instance, WCBs over the North Atlantic are often linked to deep convection (Oertel et al., 2019), which may be underestimated in ERA5, potentially also reducing the ascending motion. However, ERA5 benefits from data assimilation, which integrates observational data into the model and helps compensate for some of these shortcoming. As shown by Binder et al. (2020), ERA5 can still capture the large-scale structure of WCB clouds, including their location and thermodynamic cloud phase, which is sufficient for the scope of our study.

Moreover, ERA5 offers a physically consistent dataset with global coverage and high temporal resolution, which is of great importance when studying cyclone life cycles in regions with sparse observational data, such as over the ocean. Given current data availability, it remains

the most suitable dataset for this type of analysis. Finally, the main advantage of ERA5 is that the meteorological fields remain physically consistent – that is, that temperature, moisture, and vertical motion evolve coherently with the precipitation field.

We have slightly revised the relevant paragraph (Line 142-146) in the manuscript to more clearly acknowledge these limitations.

> 15) Line 183: Over what period is the intensification measured? 24 hours or the entire cyclone lifecycle?

The intensification is measured from the time of cyclogenesis to the time of maximum intensity. We have updated the text (also incorporation comment 11 of reviewer 2) to reflect this definition more clearly.

*To quantify this, we apply the definition of a moving cyclone proposed by Eckhardt et al. (2004), which defines a moving cyclone as one that during its life cycle travels at least 1000 km and intensifies by more than 10 hPa. Based on this criterion, the selection is limited to three stationary cyclones, while the remaining 685 cyclones all travel more than 1000 km and meet the intensification threshold.*

> 16) Line 196: In the figure caption what does the maximum depth refer to? Is this the maximum deepening in 24 hours? Or the total deepening over the cyclone lifecycle?

Thank you for this question. In this study, we use the cyclone depth as a measure of cyclone intensity, defined as the pressure difference between the cyclone center and the cyclone edge (Line 165). The maximum depth thus refers to the largest pressure difference reached at any point during the cyclone's life cycle — in other words, the cyclone's time of maximum intensity. This is calculated from hourly data along each track. We have changed 'maximum depth' into 'maximum cyclone depth' in the figure caption to make it more clear.

> 17) Line 212: Frontal circulations occur within 500km of the cyclone centre, particularly in the developing stages before frontal fracture occurs.

We agree that frontal circulations often occur within 500 km of the cyclone center, particularly during the early stages of development. Our intention was not to exclude all frontal circulations, but rather to avoid including precipitation that is too far removed from the cyclone core and may be influenced by nearby orography or unrelated synoptic features. We therefore consider it appropriate to include frontal circulations within this radius, as they are typically linked to dynamical ascent associated with the cyclone itself.

> 18) Line 214: Without observations, how do you know that the precipitation is allocated to the 'right' moisture sources?

Thank you for pointing this out. We acknowledge that the term *"right"* may be misleading, as we do not have observational evidence to confirm the correctness of the identified sources. We have therefore revised the sentence to clarify that the focus is on identifying *specific* moisture source regions, without implying absolute correctness.

*The number of days is chosen based on three factors: (1) computational cost and computer power, (2) numerical accuracy, and (3) the ability to allocate a significant proportion of precipitation to specific moisture sources.*

19) Line 227: Is the first timestep, the first 3 hours?

Yes, the first timestep is indeed the first three hours. We have revised the sentence to make this more clear:

*… and where a significant amount of moisture is lost during the first three hours back in time ($\Delta q < -0.1$ g kg-1 3h-1).*

20) Line 233: Is the start point of the trajectory the cyclone centre, or the location of the trajectory 8-days earlier?

The trajectory start is the location where the air parcel is released (at t = 0 h), so in the cyclone center. To improve clarity, we avoid using the term 'trajectory start' and instead refer to the physical location of the precipitation, i.e. 'the cyclone center'. This terminology better reflects the context of our analysis and avoids confusion with the direction of the backward trajectories.

We made the following changes (also taking into account comment 18 of reviewer 2):
*Line 225: Since we are interested in the moisture sources of precipitation, we first select the precipitating trajectories within the cyclone center, defined as trajectories where the relative humidity (RH) exceeds 80% at the trajectory start (t = 0 h), …*
*Line 233: Increases in specific humidity indicate moisture uptakes, while decreases (prior to reaching the cyclone center) suggest intermediate precipitation.*
*Line 253: In contrast, the moisture that the air parcel has gained at t = -168 h partly precipitates during the intermittent precipitation event, and will have a smaller contribution to the precipitation at the cyclone center.*
*Figure 2: 'Trajectory start' in the top diagram is changed to 'Origin'.*

21) Line 237: Is there sensitivity to this choice of threshold?

For this study, we used a threshold value of 0.075 g kg$^{-1}$ 3h$^{-1}$, based on the threshold of 0.025 g kg$^{-1}$ h$^{-1}$ that was proposed by Papritz et al. (2021). We did not experiment with the choice of threshold ourselves, as we closely followed the methodology of Papritz et al. (2021) and had no intention of deviating from it. However, the choice of threshold is extensively discussed in Sodemann (2025) and the authors of Papritz et al. (2021) did some experimenting as well. The threshold is introduced primarily to reduce the computational cost of the moisture source diagnostic by discarding small fluctuations of specific humidity along trajectories that have little impact on the identified moisture sources, and to filter out small fluctuations that are due to numerical noise.

A typical and well-tested value for the midlatitudes is 0.2 g kg$^{-1}$ 6h$^{-1}$. It is possible that this value may need to be higher for (sub)tropical regions, while in colder and drier conditions,

such as in polar or high-altitude regions, a lower threshold of 0.1 g kg⁻¹ 6h⁻¹ has been suggested (Sodemann and Stohl, 2009). Given that in our study we are using shorter time intervals and our domain covers a part of the subtropics, we have chosen to adopt the threshold value of 0.075 g kg⁻¹ 3h⁻¹. In principle, a low threshold could lead to more noise being included as moisture uptakes, while a higher threshold leads to a higher chance of discounting previous moisture uptakes, and thus may induce a bias towards more local moisture sources. However, the authors of the wintertime study found that applying or omitting the threshold led to a change in attributed precipitation by much less than 1%, indicating negligible sensitivity. Based on this, we consider the effect of the threshold to be negligible in our case as well.

22) Line 274: Does the maximum depth refer to a deepening of 35.2 hPa over the entire cyclone period in which case is it a maximum or should it be a total? Or is it the maximum deepening over a 24-hour time window, in which case the time period should be included.

The 'maximum depth of 32.5 hPa' refers to the largest pressure difference between the cyclone center and its edge – not a deepening over the full period, nor a 24-hour deepening rate. The time of cyclogenesis is mentioned, as is the time it takes to reach this maximum depth. Therefore, we believe that no revision to the sentence is necessary.

23) Line 293: What is meant by 'keep the moisture in place'? Typically, the winds near cyclones are relatively strong.

Thank you for pointing this out. We did not mean to suggest that the winds near cyclones are weak. Instead, we suggest that the winds over the moisture source regions – which are located further away from the cyclone itself – can be relatively weak prior to the cyclone's arrival. This allows for the evaporated moisture to remain near its sources instead of being transported away, which facilitates a build-up of moisture. As the developing cyclone propagates into the region, the moisture will be sucked into the warm sector.

We will revisit Line 293 and make this clearer.

24) Line 295: The authors might be interested in reading Demirdjian et al. (2023) which describes this process very nicely.
Demirdjian, R., Doyle, J.D., Finocchio, P.M. and Reynolds, C.A., 2023. Preconditioning and intensification of upstream extratropical cyclones through surface fluxes. Journal of the Atmospheric Sciences, 80(6), pp.1499-1517

Thank you for pointing us to the Demirdjian et al. (2023) paper, which we were not aware of. This paper describes how surface fluxes contribute to the interaction between clusters of cyclones, specifically by showing how surface sensible and latent heat fluxes moisten the airstream that connects a primary cyclone to an upstream developing secondary cyclone. This airstream feeds moist air parcels into the AR and WCB, and is closely related to the feeder airstream concept discussed in Dacre et al. (2019).

After reading the paper, we revisited our case study with this conceptual model in mind. While we do observe a large-scale primary cyclone (C1) preceding the development of the cyclone of interest (C2), the characteristics of our case differ from the typical cyclone cluster interactions described by Demirdjian et al. (2023). C1 appears to be embedded in a broader wave train with alternating cyclones and anticyclones, but C2 does not seem to develop upstream as part of this wave train. The development of C2 likely results from a smaller-scale disturbance along the baroclinic jet, and differs significantly in scale from C1. Over time, it propagates southward from C1, which remains largely stationary, and eventually overtakes it.

Given these differences with the results from the idealized model setup, we are hesitant to conclude that a clear upstream-developing cyclone or a well-defined primary-to-upstream feeder airstream can be identified in our case study. In particular, the specific configuration highlighted by Demirdjian et al. (2023), where the feeder airstream passes along the southern flank of an anticyclone through a region of strong latent and sensible heat fluxes, is not evident in our case.

We will revisit the case study section to incorporate our new insights and add a reference to Demirdjian et al. (2023) where appropriate, without directly linking our observations to the presence of a primary-to-upstream airstream.

25) Line 300: Figure 3 uses a colour scale that makes the fluxes on the warm side of the Gulf Stream appear to be positive. I had to look very carefully to see the 'sharp transition' from negative to positive. I suggest using a divergent red-blue colour scale instead.

Thank you for pointing this out. We will explore different colour scales to make sure the transition from negative to positive values is better visible.

26) Line 303: What is the surface evaporation elevated with respect to? No climatology of evaporation is shown so it is not clear if this is a climatological response or related to the passage of the cyclone.

This and the following comment, indicate that the term 'evaporation hotspots' needs more context. We therefore agree that it might be useful to explain how these regions relate to the climatology, to substantiate the effect of the passage of a cyclone. Additionally, the visual representation could be improved to make these anomalies more clearly visible, for example by using a different colour scale or applying normalization. We will explore various options for improving this figure, including the suggestions provided.

27) Line 305: Figure 4 uses a colour scale in which it is difficult to observe the 'evaporation hotspots'. Perhaps anomalies or normalised anomalies could be shown? I.e. subtract the climatological field and divide by the standard deviation of the climatology at each location. Then it would be possible to see how large anomalies are relative to the climatological mean, i.e. 1 std deviation larger/smaller than climatological mean?

See comment above.

28) Line 318: No evidence is provided to support the statement that the dynamical flow is setup by the primary cyclone, or that significant moisture accumulates ahead of the developing cyclone.

There are parts of the manuscript where it seems that we have been too speculative (see also comment 33 and comments 28 and 30 by reviewer 2). We will therefore rephrase these sentences so that it becomes clearer when we are hypothesising (using existing literature) or actually have evidence of our own that supports the statement.

In this case, the formulation is too strong. Nevertheless, once we plot the values in Figure 4 as anomalies with respect to the climatology, we can say whether the primary cyclone has significantly moistens the boundary layer ahead of the developing cyclone.

29) Line 326: Here the authors refer to the sources of moisture moving eastward. Presumably this is related to the fact that the cyclone is also moving eastward?

Yes, the eastward shift of the moisture sources is indeed a consequence of the cyclone moving eastward. The original sentence already describes the shift in both the cyclone's position and the moisture sources, but their connection could be stated more clearly.

*As the cyclone moves eastward throughout its life cycle, the sources of moisture also shift eastward in response to this motion (Fig. 5b, c), while most of the uptakes remain aligned with the north-eastward extension of the Gulf Stream Front.*

30) Line 332: How are long-range and local defined. Do they refer to a distance travelled by the trajectories over the 8-day period or are they the distance relative to a fixed feature, or the moving cyclone?

We have revised this line in the manuscript to clarify what we mean. For a more detailed discussion, please see our reply to the first general comment above.

31) Figure 5: It is useful to show the cyclone track, but I wonder if the position of the cyclone at 24hr intervals preceding the trajectory start time should be shown instead?

Thank you for this suggestion. It is indeed a good idea to also show the position of the cyclone center in 24-hour intervals prior to the time of precipitation, so that these locations can be compared with the location of the moisture sources. We will therefore experiment with adding additional symbols, both on their own and in combination with the cyclone track.

32) Line 351: I don't think that the trajectory figures 6a-c are very legible. It's difficult to identify the different pathways. If 80% of the moisture uptake occurs within 4 days of precipitation (line 515), would it be better to show shorter trajectories in figure 6 to make it more legible?

The rationale behind plotting the eight-day trajectories is to show the entire air parcel history. However, we will explore alternative representations, including shorter trajectories, to make the figure more legible.

33) Line 354: How do you know the parcels were lifted by convection?

There are parts of the manuscript where it seems we have been too speculative (see also comment 28 and comments 28 and 30 by reviewer 2). We will therefore revise this sentence to focus more clearly on the key takeaways of Figure 6, avoiding interpretations not supported by evidence. Specifically, rather than suggesting that the air parcels originating from the Gulf Stream region are located in the mid-troposphere due to convective uplift (for which we indeed have no evidence), we will emphasize the broader evolution of how the trajectories are distributed vertically.

*During the intensification phase, the precipitating air parcels originate from a range of vertical levels, and remain relatively unorganized. At the time of maximum depth, two distinct levels emerge: one characterized by strongly descending air parcels from higher up in the atmosphere, and another remaining close to the surface). Finally, as the cyclone matures and the velocity field becomes more organized, almost all precipitating trajectories originate from within the boundary layer, near the surface.*

34) Line 355: If the parcels are lifted by convection, how do they converge at the surface?

The trajectories we describe are those that eventually precipitate near the cyclone center and therefore must undergo ascent in the hours leading up to precipitation, which is preceded by convergence near the surface. In the days prior to this convergence, the trajectories originate from a range of vertical levels. Those descending from higher altitudes are likely associated with a dry intrusion, while others in the mid-troposphere may have experienced earlier lifting – potentially due to shallow convection in regions such as the Gulf Stream. However, all these parcels are eventually drawn towards the warm sector within the cyclone center due to the low-level convergence.

In light of this and the previous comment, we will revise these sentences and focus more on the key takeaways from Figure 6, rather than hypothesizing too much on the underlying processes for which we do not provide any evidence.

35) Line 359, 370, 570: Here and elsewhere, the descending trajectories behind the cyclone cold front are known as the dry intrusion. It might be helpful to refer to this term to enable links with previous work.

Thank you for this suggestion. We agree that it would be helpful to link the trajectories that we have calculated to known cyclone-relative airflows, such as the dry intrusion. However, we chose not to explicitly classify them as such, since many previous studies adopt threshold-based definitions (e.g., Raveh-Rubin, 2017, who defines dry intrusions as descending at least 400 hPa within 48 hours), when we prefer not to adopt such strict definitions. That said, we believe that it might be a good idea to refer to the term dry intrusion with reference to the conveyor belt model of Browning (1990, 1997), where it is more broadly defined as the descending air stream behind the cold front, and will make these changes.

36) Line 360, 578-9: Likewise, the air behind the primary cyclone cold front which is then 'fed into the secondary cyclone' is known as the feeder-airstream. It might be helpful to refer to this term to enable links with previous work.

Thank you for this suggestion. We agree that it is a good idea to term the air behind the primary cyclone cold front the feeder-airstream, with a reference to Dacre et al. (2019). We will make these changes.

37) Line 366: Use of the phrase 'came from' suggests air travelling towards the cyclone from the primary cyclone. Would a better description be that this air was swept up by the secondary cyclone as it travels into the region of enhanced moisture?

We prefer 'originate from' as an alternative for 'came from' to avoid implying stationarity. We have also added the suggested alternative to improve the sentence even more.

*Air parcels no longer originated in the Gulf Stream region but instead originated from the primary cyclone or further north along the Greenland coast before being swept up by the secondary cyclone.*

38) Line 410: You have not shown evidence that maximum vertical motion occurs before minimum MSLP? I think maximum precipitation typically occurs before maximum dynamical intensity due to limited moisture availability as cyclones travel polewards.

Thank you for pointing this out. We have indeed not shown evidence that maximum vertical motion occurs before minimum MSLP. While precipitation is influenced by both vertical motion and moisture availability, and it is plausible that poleward-moving cyclones are limited by moisture availability leading to an earlier peak in precipitation, we have not analysed this relationship in detail. To avoid overinterpretation, we will remove the statement about the timing of maximum vertical motion, as it is speculative and does not add essential insights beyond what is already said.

39) Line 429: Can you concentrate precipitation. Perhaps extending would be a better word to describe the shape of the precipitation pattern?

We agree that 'concentrate' might be somewhat confusing. We simply mean that the highest precipitation rates are found near the cyclone center. Therefore, we have revised the sentence as follows:

*In the Gulf Stream region, the highest precipitation rates are found near the cyclone center, displaying a near-symmetrical structure (Fig. 7a-c).*

40) Line 438: The precipitation does not significantly drop for Gulf Stream cyclones since the moisture availability remains high, even in the decay phase. By contrast, precipitation drops significantly for the other regions due to reduced dynamical forcing and reduced moisture availability.

Thank you for providing this additional interpretation. We have made a revision to the sentence, incorporating that the precipitation does not significantly drop for Gulf Stream cyclones, likely due to the high moisture availability.

41) Figure 7: Much of the precipitation pattern in the decaying phase is outside the domain shown. Could a larger domain be used to capture the full shape of the cyclone precipitation pattern in this phase?

We agree that much of the precipitation pattern is outside of the circular domain. However, our focus is specifically on precipitation in close vicinity of the cyclone center, to ensure that the precipitation is indeed dynamically and physically related to the cyclones. By increasing the domain we would risk including precipitation in the cold sector or along trailing cold fronts – features that can extend far southwards, and may be influenced by the larger upper-level trough or have their own dynamics.  In addition, we prefer not to use a larger domain, since a one-to-one comparison with the winter would no longer work.

42) Line 469: The authors describe the eastwards shift in the moisture sources as 'slight'. This is a subjective word. Could they calculate the magnitude of the eastward shift? This could then be compared with the eastward shift in the mean cyclone position to see if they are comparable.

It might indeed be interesting to compare the shift in moisture sources to the eastward shift in mean cyclone position, to see if the latter is fully responsible for the former. We will do our best to quantify the eastward shift, or otherwise avoid subjective terms.

43) Line 469: The eastward shift is also evident for the Gulf Stream cyclones.

Thank you for pointing this out. We will change the sentence from 'this shift is only evident for cyclones in the East Atlantic and Nordic Seas regions' to 'this shift is evident for all cyclones apart from the cyclones in the Labrador Sea'. As a side note, we will try to quantify this eastward shift for each subregion, and include those values as well (see our response to the previous comment).

44) Line 474: Local to what?

We have revised this line in the manuscript to clarify what we mean. For a more detailed discussion, please see our reply to the first general comment above.

45) Figure 8: Would it be possible to show the mean cyclone position at various stages in the cyclone lifecycle.

Thank you for this suggestion. It is indeed useful to show the mean cyclone track and the mean positions of the cyclone center at the different phases in the cyclone life cycle. We will add them to the figure.

46) Line 495: Within 1000km of what?

We have revised this line in the manuscript to clarify what we mean. For a more detailed discussion, please see our reply to the first general comment above.

47) Line 513: Would these terrestrial sources be the Great Lakes, or are they more widespread than this?

Zooming in on the moisture footprint of the cyclones intensifying in the Labrador Sea (Fig. 8), we observe that the terrestrial sources extend over a large part of eastern North America, not just the Great Lakes. However, we can attempt to directly estimate the contribution of the Great Lakes by checking whether the moisture source coordinates match those of the Great Lakes.

We will try to calculate this and add the value to the manuscript (in Line 513). However, it is important to note that our data are gridded onto the arrival domain – that is, the cyclone center – so the source latitudes and longitudes are weighted averages based on all moisture uptakes contributing to precipitation at each arrival point. As a result, the specific contribution from the Great Lakes may be underestimated, particularly if their moisture input is not dominant at those locations. Thus, the true contribution of the Great Lakes could be higher than our estimate will suggest.

48) Line 526: Sources 1000-1500km away from what?

We have revised this line in the manuscript to clarify what we mean. For a more detailed discussion, please see our reply to the first general comment above.

49) Line 539: Ascending trajectories ahead of the cold front are part of the WCB airflow. It might be helpful to refer to them as the ascending part of the WCB to enable links with previous work.

Thank you for this suggestion. We agree that it would be helpful to link the trajectories that we have calculated to known cyclone-relative airflows, such as the WCB. We have revised the sentences as follows:

*During all three phases in the cyclone life cycle, air parcels are concentrated in the lower troposphere (below 800 hPa) in the days leading up to precipitation, followed by a pronounced ascending motion within the last 12 hours before the precipitation. This ascent occurs primarily in the warm sector of the cyclone and is associated with the WCB airstream. As the air parcels are lifted in the WCB, they cool and reach saturation, leading to condensation and strong latent heat release that further enhances vertical motion.*

50) Line 540: What is the evidence for moisture accumulation near the cyclone centre?

We indeed show no evidence for moisture accumulation near the cyclone center, making this statement too speculative. Therefore, we have removed this part of the sentence. The new version of the sentence is given in the response to the previous comment.

51) Figure 11: These figures correspond well with figures 5 and 6 in Dacre et al. (2020), showing maximum latent heat fluxes behind the cyclone cold front as a result of large vertical gradients in specific humidity.
Dacre, H.F., Josey, S.A. and Grant, A.L., 2020. Extratropical-cyclone-induced sea surface temperature anomalies in the 2013–2014 winter. Weather and Climate Dynamics, 1(1), pp.27-44.

Thank you for pointing us to this paper. These figures offer a potential explanation for the region of strong surface evaporation in to the southwest of the developing cyclone, which is co-located with the cyclone's own cold sector. We have therefore included this explanation and a reference to the Dacre et al. (2020) paper in the following sentences (in which we also incorporate comment 35 and comment 31 of reviewer 2):

*One pronounced region of elevated evaporation appears southwest of the developing cyclone, co-located with the cyclone's cold sector. As shown by Dacre et al. (2020), the strong upward latent heat fluxes observed behind the cyclone's cold front are a result of sharp temperature and specific humidity gradients between the ocean surface and the overlaying air, with notably cooler and drier air at 10 meter. This environment allows cold, dry air that is descending behind the cold front – commonly referred to as the dry intrusion airstream (Browning 1990; Browning 1997) – to efficiently take up moisture that has evaporated from the ocean surface (Aemisegger and Papritz, 2018; Ilotoviz et al., 2021).*

52) Line 615: What about moisture evaporated in the subtropical high as referred to earlier in the paper?

The subtropical high is indeed associated with enhanced surface evaporation, as mentioned earlier in the manuscript. However, based on the current analysis, we cannot confirm whether this region actually contributes to the cyclone precipitation. As noted in our response to the second general comment, addressing this question would require us to either rephrase our research question or do additional analysis that provides more insight into the contribution of the subtropical high to the cyclone precipitation. If we manage to do this, we would consider including such a discussion in the conclusion.

53) Line 623: Here the authors refer to 'delivery' of moisture to the cyclone. Again, this implies the cyclone is stationary and that moisture is transported towards it from 'remote sources'. Is this really what is happening?

We have revised this line in the manuscript to clarify what we mean. For a more detailed discussion, please see our reply to the first general comment above.

54) Line 630: Source distances from what?

We have revised this line in the manuscript to clarify what we mean. For a more detailed discussion, please see our reply to the first general comment above.

55) Line 650: Dacre et al. (2023) use a very different method to that used here and estimate a mean moisture residence time of 36 hours. Do you think this is due to

methodological differences or because they focus on extreme winter cyclones where the ascent rates are typically higher than for summer cyclones?

Our estimated precipitation life time (approximately four days) is longer than the mean residence time of moisture associated with winter cyclone precipitation, which was found by Papritz et al. (2021) to be approximately three days using the same method. Therefore, we hypothesize that Dacre et al. (2023) estimated a mean moisture residence time of 36 hours, which is at least partly attributable to their focus on extreme winter cyclones. The ascent rates in these cyclones are typically higher than in summer cyclones, resulting in shorter moisture residence times. In addition, differences in methodological approaches will likely contribute to the discrepancy as well, as there can be significant differences between various moisture tracking methods (Benedict et al., 2024).

56) Line 657: You have not included a figure of cyclone-relative low-level winds so it is not possible to determine if a feeder airstream is present in your cyclone composites. Would it be possible to overlay cyclone-relative streamlines to see if the feeder airstream can be identified?

Thank you for this suggestion. We agree that it could be useful and will consider adding it to the figure. Adding cyclone relative low-level winds can give a first-order estimate of the direction of moisture transport, and whether feeder airstreams exists. Please also see our reply to the second general comment, in which we discuss this in more detail.

57) Line 671: This sentence suggests there are multiple ascending motions. Are the authors referring to frontal line convection as an alternative to the WCB airflow?

Thank you for pointing this out. We were simply referring to the ascending motion in the warm sector and in the WCB. We have therefore revised the sentence to use the singular 'motion' to avoid any confusion.

58) Line 676: Similar to what, the precipitation in winter cyclones?

Yes, similar to winter cyclones. To clarify this, we have revised the sentence as follows:

*… the higher moisture content in warmer air parcels results in similar precipitation to winter extratropical cyclones once saturation is reached.*

59) Line 680: How are you separating fronts and WCB airflow? They are generated by the same mechanism, so it seems strange to separate the precipitation associated with them. Perhaps you are referring to the horizontal and vertical parts of the WCB? Or to frontal line convection and WCB ascent?

Thank you for raising this. The original suggestion proposed to examine the contributions of different features to precipitation. In this context, we agree it is important to clarify whether we focus on *cyclone-related features* – in which case a more specific distinction should be made, for instance between frontal convection and WCB ascent, like you mention – or on broader *weather features*, as done in Konstali et al. (2024), who distinguish between fronts, cold air

outbreaks, cyclones, and atmospheric rivers (or moisture transport axes). We will revise these sentences to make it more clear which features can be explored.

Typographical errors
    60) Line 10: Why is the word 'extreme' in brackets?

Our intention was to indicate that while extratropical cyclones are often associated with extreme precipitation, this is not always the case — depending, for example, on how 'extreme' is defined. We wanted to say that these systems generally produce precipitation, and that this can include extremes. To improve clarity, we have revised the sentence to:

*"…and are known for their ability to produce precipitation, including extremes."*

    61) Line 42: Why is 'heavy' in brackets?

As in the previous comment, the word 'heavy' had been placed in brackets to reflect this nuance, but we have now removed the brackets to more clearly highlight the potential for high-impact precipitation associated with these systems.

    62) Line 97: Here and elsewhere, it is more standard to refer to the singular 'precipitating water' rather than the plural 'precipitating waters'.

Thank you for this suggestion. We will make the changes.

    63) Line 101: 'How' should be 'whether' since you haven't determined this yet.

Thank you for pointing this out. It should indeed be 'whether'.

    64) Line 164: You do not need cf.

Thank you for this comment. Instead of 'cf.', we added information on how the North Atlantic region is represented in the figure.

*… a cyclone must reach its maximum intensity within the North Atlantic region (defined by the black line in Fig. 1), …*

    65) Line 171: The number 2 needs units.

Thank you for spotting this mistake. We have included the units hPa.

    66) Line 216: I think interpolated would be a better description than 'traced'.

While we acknowledge that 'interpolated' could, in some contexts, be a more precise term, we have chosen to retain the use of 'traced'. The rationale behind this decision is to maintain consistency with the terminology employed in the trajectory calculation tool LAGRANTO, which has a specific function named 'trace' that derives atmospheric variables along air parcel

trajectories. Using an alternative term here could cause confusion, particularly for readers familiar with the tool and its functionality.

**Review 2**

This paper adopts a cyclone-centred perspective to evaluate the moisture sources and transport pathways of North Atlantic deep cyclone precipitation in summer. The paper strongly builds scientifically and methodologically on a previous paper (Papritz et al. 2021) focussing on winter deep North Atlantic cyclone precipitation. The paper is well written and the main findings are interesting and related to

i)   moisture residence times being relatively constant of about 4 days throughout the cyclone life cycle;

ii)  the moisture sources of cyclones originating from the tropics and making extratropical transition being mainly located in the subtropics and midlatitudes, with very limited amounts coming from the tropics directly

iii) contributions from different key geographical and cyclone-relative regions, such as land, the warm side of the Gulf Stream as well as evaporation from the cold sector of preceding cyclones, although the discussion mainly relates to geographic regions and not quantitatively to cyclone-relative regions.

I have a few minor comments mainly related to the writing and presentation of the results.

Thank you very much for reviewing our paper and your suggestions on how to improve the writing and presentation of the results. We appreciate your feedback and are glad that you found the main findings interesting. To improve clarity, we will emphasize the first two key findings more prominently in the conclusion section. We have also addressed all your specific comments as detailed below.

Comments 5, 24, 30, and 33 relate to Q2 regarding cyclone-relative moisture sources. Since reviewer 1 raised similar points, we refer to our response to their general comment 2 for a more detailed discussion.

1) Innovation: I think the paper could become a bit sharper in terms of its innovative contributions to science. In my reading, I got the impression that it was very closely following the preceding paper by Papritz et al. 2021 both in terms of scientific focus and methodological approach. The fact that summer deep cyclones are generally less studied has good reasons, they are rarer and less intense than winter cyclones. Therefore, the motivation for this study could be carved out a bit more convincingly. I do think there are good reasons to investigate summer cyclones separately, e.g. to investigate the dynamical impact of added moisture from the land sources, the role and moisture transport pathways related to cyclones from tropical origin, potential similarities with future warmer conditions with weaker baroclinicity also in winter, contrasts between cyclones over the ocean vs. over land… I encourage the authors to make a stronger case for their paper in the abstract and the introduction (e.g. at L. 51-55). The fact that summer cyclones are less studied does not make it a good reason to study them.

Thank you for raising this issue. The paper by Papritz et al. (2021) provides an essential foundation, since we have indeed adopted a similar methodological approach to allow for a direct comparison between summer and winter extratropical cyclones. However, this does not mean that the present paper cannot stand alone. Therefore, we agree that it is important to highlight why it is important to study summer storms separately.

We appreciate the reviewer's suggestions and agree that factors such as moisture originating from land sources, transport pathways related to cyclones from tropical origin, and potential similarities with future warmer climates (having reduced baroclinicity) provide a strong motivation. We will revise the abstract and introduction, especially lines 51–55, to more clearly emphasize these motivations and better define the novel contributions of our study. Rather than simply filling a gap due to limited prior work, we aim to offer new insights into the mechanisms governing the summertime cyclone-related water cycle.

2) L. 62: WCBs here I think Madonna et al. 2014 and Heitmann et al. 2024 should be referenced.
   And at L. 73 about the link between WCBs and cyclones: Binder et al. 2016.

Thank you for these helpful suggestions. We agree that it makes sense to refer to Madonna et al. (2014) and Heitmann et al. (2024) when introducing the WCB as a rising air stream responsible for precipitation. We have added these references to Lines 62-64, along with an additional reference to Browning (1990), which is also cited by the other two sources.

*This location coincides with the location of the warm conveyor belt (WCB), a rising airflow characterized by intense latent heat release and large amounts of precipitation (Browing, 1990; Heitmann et al., 2024; Madonna et al., 2014).*

In addition, we have incorporated a reference to Binder et al. (2016) in the discussion of the link between WCBs and extratropical cyclones. Line 73 has been revised as follows (also including comment 11 of reviewer 1):

*According to Eckhardt et al. (2004) and Pfahl et al. (2014), the WCB and extratropical cyclones are intrinsically linked. Furthermore, the WCB is capable of influencing the dynamics of the cyclones – a role that has been underscored in the context of cyclone intensification by Binder et al. (2016).*

3) WCBs: throughout the paper the authors should be much more cautious with their definition of the WCB and clearly define what they mean. Usually, this airstream is defined as ascending by 600 hPa or more in 48 h. It is very likely that in summer the airstreams are ascending less (see also the substantially lower frequency of WCBs in summer over the North Atlantic). Also, the convective parts of the ascent are probably missed in the approach chosen by the authors (Oertel et al. 2021) calculating the trajectories based on 3 hourly 3D wind fields on the still relatively coarse ERA5 grid.

Thank you for this comment. In response, we have revised the relevant sentences in the introduction to more clearly define both warm conveyor belts (WCBs) and atmospheric rivers (ARs), highlighting that these features are both airstreams within the warm sector of

extratropical cyclones that start to ascent. However, rather than applying the strict trajectory-based definition of WCBs – defined as parcels ascending by at least 600 hPa within 48 hours – we have intentionally adopted a looser, more qualitative usage of the term in this study. Our focus is on the airstreams that transport warm, moist air poleward and upward, without explicitly identifying WCBs via trajectory thresholds. This is particularly relevant in summer, when, as you mentioned as well, ascent rates are generally lower. Instead, in Section 4.4, we discuss the presence of weaker ascent in these airstreams during summer, rather than suggesting that WCBs are absent.

Regarding the second part of the comment, we agree that convective components of ascent are likely underrepresented due to the use of 3-hourly 3D wind fields on the relatively coarse ERA5 grid. This could indeed result in an underestimation of rapid convective ascent, particularly during summer. We now explicitly acknowledge this limitation in Section 4.4. Nevertheless, previous studies (e.g., Binder et al., 2020) have shown that ERA5 is still able to capture the large-scale structure of WCB cloud features, including their location and cloud phase. Given that our analysis emphasizes broad structural and thermodynamic characteristics rather than detailed convective dynamics, we consider ERA5 adequate for the goals of this study.

4) L. 76-87: I don't understand the use of discussing atmospheric rivers in such great detail since they are not identified or discussed further in the results of this paper. I would suggest shortening and shifting the discussion on the potential role and link with ARs to the conclusion.

Thank you for your comment. We agree that the initial discussion of ARs in the introduction was a bit too detailed, especially given that ARs are not directly identified or analysed in the results. In response, we have revised this section to be more concise and focused, emphasizing the potential role of ARs as moisture sources for warm-sector precipitation. The potential link between ARs and the studied precipitation patterns is already briefly acknowledged in the conclusion, and we feel that no further changes are needed there.

5) Q2 is not addressed in a cyclone-relative way. I do think that the authors would have the necessary data and tools to address this question, with a bit more coding work and gridding the uptakes for different times relative to cyclone maximum depth.

Thank you for raising this point. Unfortunately, this analysis is more difficult than it seems, because the tool that we use only provides moisture uptake as an accumulated footprint over the eight-day period rather than at individual time steps. Please see our reply to the second general comment of reviewer 1, in which we discuss this in more detail.

6) L. 117: Here the Lagrangian method used should also be referenced (Sodemann et al. 2008).

Thank you for the suggestion. At this point in the text, our intention is to introduce the Lagrangian approach in general terms, in order to contrast it with Eulerian methods. We do not yet refer to our specific method. In light of comment 7, we have revised the corresponding sentences to make this comparison clearer, and we hope this now better justifies why the

method by Sodemann et al. (2008) is not referenced here. The specific methodology we use is introduced and referenced in detail later in the manuscript.

7) L. 120: these are not adequate references for the use of Lagrangian moisture source identification to distinguish between different air streams. For example Pfahl et al. 2014 could be a good option.

Thank you for this comment. We agree that Pfahl et al. (2014) is a more appropriate reference when referring to the use of Lagrangian moisture source identification to distinguish between different air streams. We have now included this reference specifically in that context. The original references (Gimeno et al., 2012; Pérez-Alarcón et al., 2022) are more relevant for discussion on Eulerian versus Lagrangian approaches, and we have moved them earlier in the sentence accordingly. We have also revised the sentence structure to make the comparison between the two approaches clearer and to ensure that all references are cited in the most appropriate places.

*Lagrangian methods follow individual air parcels as they move in space and time, while Eulerian methods analyse moisture budgets at fixed locations (Gimeno et al., 2012). Although an Eulerian approach is also suitable for constructing a moisture budget (e.g. van der Ent and Tuinenburg, 2017), the advantage of the Lagrangian approach is that it allows for the quantification of moisture uptakes along air parcel trajectories and provides a high spatial resolution of moisture source diagnostics (Gimeno et al., 2012; Pérez-Alarcón et al., 2022). Furthermore, it enables the distinction between different air streams, allowing for the analysis of their associated moisture sources separately (e.g. Pfahl et al., 2014).*

8) L. 123: if the North Atlantic was studied several times before, then mention several studies. Here maybe e.g Gimeno et al. 2012, Perez-Alarcon et al. 2022 could be good options.

Thank you for this comment. We have included one of your suggestions (Perez-Alarcon et al. 2022) and added another one (Coll-Hidalgo et al., 2025).

*… as the North Atlantic has been studied several times before (e.g. Chang and Song, 2006; Coll-Hidalgo et al., 2025; Pérez-Alarcón et al., 2022), …*

9) L. 160: "Thereafter a new time axis … is defined", I think it's not a new axis, just a new time of reference.

Thank you for this suggestion. We have revised the sentence as follows:

*Thereafter, a new time axis of reference relative to the cyclone life cycle is defined.*

10) L. 168-170: Why did you exclude these cyclones? Some of them might also have made extratropical transition and actually be quite interesting to study in more detail.

We exclude these cases, as these cyclones already begin to exhibit wintertime characteristics due to increased baroclinicity later in the season. Our goal is to analyse summer cyclones only

under summertime characteristics, which are characterized by weaker baroclinicity and generally weaker vertical motion. We fully agree that cyclones undergoing extratropical transition are scientifically interesting – particularly with regard to how their characteristics change throughout the year, especially during the transition seasons. However, we believe that these systems should to be studied separately, rather than be included as an essential part of this study.

11) L. 181: "exhibit strong movement" what does this mean exactly?

We argue that the cyclones in our subset exhibit strong movement because they can cross the Atlantic basin and make landfall in Europe. We quantify this further using the definition of a moving cyclone. To better link the statement to the definition, we updated the text (also incorporating comment 15 of reviewer 1).

12) Fig. 1: add contours of track density to help interpret Fig. 8.

Thank you for this suggestion. We will consider adding the contours of the track density to the figure, while keeping in mind that the figure should remain legible. Another alternative we are considering is adding the mean cyclone track and the mean location of the cyclone centers in 24-hour intervals before to the precipitation.

13) L. 202: add Wernli and Davies 1997 for LAGRANTO to reference the original publication as well as.

Thank you for pointing this out. We have added the reference.

14) L. 212: justify the 8-days based on studies about the moisture residence time in this region and season.

We will justify the 8-days using the studies by Gimeno et al. (2012) and Läderach and Sodemann (2016).

15) L. 214: "LAGRANTO's ability to allocate a significant portion of precipitation to the right moisture sources" not sure I understand what you mean here. Do you mean Watersip instead of LAGRANTO? And how do you know what the "right sources" are?

Thank you for pointing this out. It is indeed not LAGRANTO itself that attributes precipitation to moisture sources, but rather the WaterSip diagnostic applied to the LAGRANTO trajectories. As the WaterSip tool will be introduced a few sentences later, we chose to stick to a general term here.
We also acknowledge that the term *"right"* may be misleading, as we do not have observational evidence to confirm the correctness of the identified sources. We have therefore revised the sentence to clarify that the focus is on identifying *specific* moisture source regions, without implying absolute correctness.

*The number of days is chosen based on three factors: (1) computational cost and computer power, (2) numerical accuracy, and (3) the ability to allocate a significant proportion of precipitation to specific moisture sources.*

    16) L. 219: Did you use the official Watersip code? Or an own implementation in which case it would be clearer to simply reference the original publication of the algorithm with Sodemann et al. 2008 and not call the algorithm Watersip. For reproducibility it would be easiest if you provided a link to the code used.

We used the official WaterSip code (version 3) for our analysis. To clarify this, we have updated the reference accordingly and included a better description and the version number when the tool is properly introduced in the manuscript.

*Line 228: Using these variables, the moisture budget of the air parcel can be constructed by applying the moisture source diagnostic WaterSip (Sodemann, 2025).*
*Line 230: In this study, we employ WaterSip (version 3), a software tool that implements the widely used Lagrangian moisture source and transport diagnostic developed by Sodemann et al. (2008), to identify the moisture sources (Sodemann, 2025).*

    17) L. 239: in the 48 h around maximum cyclone depth.

We have revised the sentence to incorporate the 48-hour window.

*Once all the moisture uptake locations are identified, the absolute uptake amount is translated into a moisture source footprint for  every 3 hours within the 48-hour window around maximum cyclone (from -24 h to 24 h).*

    18) L. 255: at arrival in the cyclone (trajectory start is confusing).

We revised the sentence. See also comment 20 of reviewer 1 for similar changes in the manuscript.

    19) L. 370: I would not call this the upper troposphere, there are very few trajectories coming from above 500 hPa.

We have rephrased the sentence as 'originate from higher up in the troposphere'.

    20) L. 376: implying that local moisture recycling is becoming important: what does that mean exactly?

In this context, we mean that the moisture that precipitates within the cyclone center is re-evaporated from the surface and reused by the same cyclone.

    21) L. 415-416: here Aemisegger and Papritz, 2018 would be more fitting.

We have replaced the reference *Aemisegger and Sjolte, 2018* by *Aemisegger and Papritz, 2018*.

22) L. 453: It's not clear what the initial moisture is: the diagnosed precipitation?

With 'initial moisture' we mean the specific humidity carried by each air parcel at the start of the 8-day backward trajectory, i.e. in the cyclone center. We have revised the sentence as follows:

*Overall, WaterSip effectively attributes the majority of precipitation to its source region. For approximately 90% of all precipitating waters, over 50% of the specific humidity in the air parcel at the cyclone center could be accounted for within the 8-day period (Fig. A3).*

23) L . 465: I think this is really an important point of this paper. It should be emphasised more. This contradicts the usual assumption that in cyclones from the tropics making extratropical transition, subtropical or even tropical moisture gets exported into the midlatitudes.

Thank you for pointing this out. We agree that this is an important finding that deserves to be emphasized more clearly in the manuscript. Our results contradicts the common assumption that extratropical cyclones, especially those undergoing extratropical transition, are predominantly fuelled by subtropical or tropical moisture. We will revise Lines 464–469 to better underscore this finding, and we will revisit it in the conclusion section as well. We will also include a reference to Dacre and Clark (2025), who support this result.

24) L. 469-472: this raises the question of cyclone-relative sources, which is not addressed quantitatively in this paper.

Thank you for raising this point. Please see our reply to the second general comment of reviewer 1, in which we discuss this in more detail.

25) L. 474: Here "local" needs to be defined more quantitatively. Local relative to the cyclone?

We have revised this line in the manuscript to clarify what we mean. For a more detailed discussion, please see our reply to the first general comment from reviewer 1.

26) Fig. 9b: I think panel b does not make much sense given the relatively large North-South temperature gradient.

Thank you for raising this. We will explore an alternative for this panel, or otherwise remove the panel.

27) Fig. 9d: make clear that panel d is based on the explained fraction.

In figure 9, all panels are based on the fraction of precipitation that was attributed to a source, and therefore 'explained'. We realize the potential confusion may arise because panels a–c describe characteristics of the source regions, while panel d shows a transport-related

characteristic (residence time). To avoid any confusion, we will make sure to repeat this when introducing the figure in the text and revisit the y-axis labels to better reflect this difference.

28) L. 526: "…hinting at cold-air advection" I think this is speculative.

There are parts of the manuscript where it seems that we have been too speculative (see also comment 30 and 28 and 33 by reviewer 1). We will therefore rephrase these sentences so that it becomes clearer when we are hypothesising (using existing literature) or actually have evidence of our own that supports the statement.

29) L. 545: then I would say they are not WCB trajectories. What is the role of convection and the relatively coarse spatial and temporal resolution of the ERA5 data?

As noted in our response to comment 3, we prefer not to state that WCBs are absent simply because the ascent does not meet the typical threshold of 600 hPa in 48 hours. This threshold is primarily based on wintertime WCBs, and applying it to summertime cyclones may exclude relevant airstreams, as ascent tends to be weaker in summer due to reduced baroclinicity. Instead of quantifying how many trajectories meet that threshold, we prefer to focus on the presence of an ascending airstream consistent with the conceptual WCB. It seems that even in the original conveyor belt model described by Browning (1990), such airstreams are still considered part of the WCB – albeit with weaker vertical motion.

Regarding the role of ERA5, we agree this is important to acknowledge. The relatively coarse spatial and temporal resolution of ERA5, along with its parameterization of convective processes, may underestimate the strength and vertical extent of ascent, especially in cases of deep convection like in the WCB. We have clarified this point in the manuscript.

30) Section 4.5: the discussion in this section is a bit speculative without cyclone-relative analysis of the moisture sources.

Thank you for raising this point. In this section, we analyse the spatial distribution of surface evaporation within and around the cyclone center, rather than the actual moisture sources. We will revisit this section and clarify that certain sentences are hypotheses – based on existing literature to make it less speculative – and not conclusions. Please also see our reply to the second general comment of reviewer 1, in which we discuss this in more detail.

31) L. 570: "… facilitating strong upward latent heat fluxes": yes and what matters even more for your study, is that given that the subsiding air is dry, it's efficiency in taking up humidity is large (Aemisegger and Papritz 2018).

Agreed. We have revised the sentence, please see our response to comment 51 of reviewer 1 for the changes.

32) L. 571: here maybe Illotovitz et al. 2021 about the impact of the dry intrusion on boundary layer dynamics and surface fluxes would be a good reference.

Agreed. We have revised the sentence, please see our response to comment 51 of reviewer 1 for the changes.

33) L. 657: The fact that you do not find a feeder airstream is due to the fact that you do not look at the moisture sources in a cyclone-relative perspective. Without such a cyclone-relative analysis (which I do think is feasible and which I would strongly recommend), you cannot really make this statement. If you really do not find a "feeder airstream" in your summer cyclone based on such an analysis this would reveal an interesting contrast and rise new questions about the cyclone-relative moisture cycling in summer vs. winter cyclones.

Indeed, without the cyclone-relative moisture sources it is difficult to identify a feeder airstream. Performing the cyclone-relative analysis is more difficult than it seems, because the tool that we use only provides moisture uptake as an accumulated footprint over the eight-day period rather than at individual time steps. We therefore plan to explore the option of adding cyclone relative winds, as this gives a first-order estimate of the direction of moisture transport, and whether feeder airstreams exists. Please also see our reply to the second general comment of reviewer 1, in which we discuss this in more detail.

34) L. 688: "… or from the developing cyclone's own cold sector (which appears to be more important in summer)", this sounds contradictory with the explanation at L. 573ff, where the authors discuss that this moisture recycling pathway is unlikely due to the necessary crosscold front motion of the air parcels that would rain out in the same cyclone. I recommend to rephrase this according to the statements made earlier in the paper.

Thank you for pointing this out. You are absolutely right, we do not expect moisture that evaporates in a cyclone's own cold sector to contribute directly to its own precipitation. We will revise the sentence to focus on 'cold-air advection within the cyclone's cold sector,' which, in winter, can serve as a source of moisture for precipitation in a subsequent cyclone.

**Community comment**

Although the authors would not necessarily be aware of this work, a recent article with similar aims and methods has been published (Coll-Hidalgo et al. 2025)
https://www.sciencedirect.com/science/article/pii/S0169809525002972?via%3Dihub#f0025
I believe that the manuscript would benefit from a comparison with the methods, results and conclusions of this article.

Thank you for pointing us to this paper, which we were not aware of. While the overall aim is closely aligned with ours, namely to investigate moisture sources for precipitation in North Atlantic extratropical cyclones, there are also important methodological differences. Notably, their study focuses on the winter season, employs downscaled ERA5 data using the WRF mesoscale model, and determines the moisture sources for three specific cyclone features: the cyclone radius (which aligns with our approach), the warm conveyor belt (WCB) footprint, and the square-root spiral area. These differences make a direct, one-to-one comparison difficult, as differences in seasonality, model resolution, and cyclone feature definitions can

all influence the results. That said, we agree that comparing findings based on the cyclone radius framework is particularly relevant, and we will reflect on their main results in our discussion section.

**References**

Benedict, I., Weijenborg, C., van der Ent, R., Keune, J., Koren, G., and Kalverla, P.: A moisture tracking intercomparison study - Addressing the uncertainty in modelling the origins of precipitation, EMS Annual Meeting 2024, Barcelona, Spain, 1–6 Sep 2024, EMS2024-1040, https://doi.org/10.5194/ems2024-1040, 2024.

Binder, H., Boettcher, M., Joos, H., Sprenger, M., and Wernli, H.: Vertical cloud structure of warm conveyor belts – a comparison and evaluation of ERA5 reanalysis, CloudSat and CALIPSO data, Weather Clim. Dynam., 1, 577–595, https://doi.org/10.5194/wcd-1-577-2020, 2020.

Browning, K.: Organization of clouds and precipitation in extratropical cyclones, in: Extratropical cyclones, 129–153, Springer, https://doi.org/10.1007/978-1-944970-33-8_8, 1990.

Browning, K. A.,: The dry intrusion perspective of extra-tropical cyclone development. Meteor. Appl., 4, 317–324, https://doi.org/10.1017/S1350482797000613, 1997.

Chang, E. K. M., Lee, S., and Swanson, K. L.: Storm track dynamics, J. Climate, 15, 2163–2183, https://doi.org/10.1175/1520-0442(2002)015<02163:STD>2.0.CO;2, 2002.

Coll-Hidalgo, P., Nieto, R., Fernández-Alvarez, J. C., and Gimeno, L.: Assessment of the origin of moisture for the precipitation of North Atlantic extratropical cyclones: Insights from downscaled ERA5, Atmos. Res., 324, 108205, https://doi.org/10.1016/j.atmosres.2025.108205, 2025.

Dacre, H. F. and Clark, P. A.: A kinematic analysis of extratropical cyclones, warm conveyor belts and atmospheric rivers, npj Clim. Atmos. Sci., 8, 97, https://doi.org/10.1038/s41612-025-00942-z, 2025.

Dacre, H. F., Martinez-Alvarado, O., and Mbengue, C. O.: Linking atmospheric rivers and warm conveyor belt airflows, J. Hydrometeorol., 20, 1183–1196, https://doi.org/10.1175/JHM-D-18-0175.1, 2019.

Gimeno, L., Stohl, A., Trigo, R. M., Dominguez, F., Yoshimura, K., Yu, L., Drumond, A., Durán-Quesada, A. M., and Nieto, R.: Oceanic and terrestrial sources of continental precipitation, Rev. Geophys., 50, RG4003, 10.1029/2012RG000389, 2012.

Hartmann, D. L.: Global Physical Climatology, Academic Press, San Diego, CA, 408 pp., ISBN: 9780123285300, 1994.

Läderach, A., and Sodemann, H.: A revised picture of the atmospheric moisture residence time, Geophys. Res. Lett., 43, 924–933, doi:10.1002/2015GL067449, 2016.

Oertel, A., Boettcher, M., Joos, H., Sprenger, M., Konow, H., Hagen, M., and Wernli, H.: Convective activity in an extratropical cyclone and its warm conveyor belt – a case-study combining observations and a convection-permitting model simulation, Q. J. Roy. Meteorol. Soc., 145, 1406–1426, 2019.

Papritz, L., Aemisegger, F., and Wernli, H.: Sources and transport pathways of precipitating waters in cold-season deep North Atlantic cyclones, J. Atmos. Sci., 78, 3349–3368, https://doi.org/10.1175/JAS-D-21-0105.1, 2021.

Peixoto, J. P. and Oort, A. H.: Physics of Climate, American Institute of Physics, New York, 520 pp., ISBN: 9780883187128, 1992.

Raveh-Rubin, S.: Dry Intrusions: Lagrangian Climatology and Dynamical Impact on the Planetary Boundary Layer. J. Climate, 30, 6661–6682, https://doi.org/10.1175/JCLI-D-16-0782.1, 2017.

Sodemann, H.: The Lagrangian moisture source and transport diagnostic WaterSip V3.2, EGUsphere [preprint], https://doi.org/10.5194/egusphere-2025-574, 2025.

Sodemann, H., and Stohl, A.: Asymmetries in the moisture origin of Antarctic precipitation, Geophys. Res. Lett., 36, L22803, doi:10.1029/2009GL040242, 2009.

---

## Author Response (AR1)

**Precipitation, Moisture Sources and Transport Pathways associated with Summertime North Atlantic Deep Cyclones**

**Authors' response**

We thank the two reviewers for their insightful comments. In the following, we outline in red how we have addressed their comments. Changes to the manuscript are italicized, with the corresponding line numbers referring to the tracked-changes version of the manuscript.

**Review 1**

This paper aims to evaluate the moisture sources for summertime extratropical cyclones. It employs a Lagrangian back trajectory method to determine the sources and sinks of moisture for parcels that result in precipitation at the centre of cyclones. Overall, the paper is well written, the analysis interesting and conclusions a valuable contribution to the literature. I have made quite a lot of specific comments, but they are largely points of clarification rather than extensive requests for further analysis. I recommend that this paper would be suitable for publication in WCD if the points below are addressed.

Thank you very much for reviewing our paper and all your suggestions on how to improve the writing and clarify certain parts. We are glad to hear that you find the analysis interesting and the conclusions a valuable contribution to the literature! A more detailed answer per comment is given below.

**General comments**

1) Throughout the paper the authors refer to moisture being 'transported from remote regions to the centre of the cyclone' or the 'delivery' of moisture to the cyclone. These phrases imply that the cyclone is stationary, and that moisture is transported towards it. How good is the approximation of a stationary cyclone? If the cyclone is not stationary, then moisture may remain 'local' to the cyclone centre, but still undergo 'long-range transport' as it moves with the cyclone. Please could the authors be more specific with their phrasing and define what they mean by local and remote.

Thank you for pointing this out. We are happy to clarify this aspect of our paper. We agree that our phrasing may have unintentionally implied a stationary cyclone, which is not the case in our analysis. In fact, we explicitly account for the movement of cyclones by following them throughout their life cycle. Every three hours, we identify moisture sources relative to the current cyclone location and timing of precipitation within the cyclone center, rather than a fixed geographic point that remains the same throughout the life cycle.

However, once the air parcels are released from their starting positions, we only follow them eight days back in time, without tracking the cyclone's position during this eight-day period. Therefore, the *source distance* we refer to in the manuscript is the great-circle distance between the location where precipitation occurs (within the cyclone center) and the origin of the moisture, rather than the distance between the moisture origin and the cyclone center at the time of the uptake. In this context, the moisture sources and the way the moisture is transported is defined in a

**geographical frame of reference**. A **cyclone-relative framework**  is adopted when we discuss the spatial distribution of precipitation and surface evaporation around the cyclone center.

We agree that we can be more specific about the definition and the frame of reference when we state the numbers. In addition, as you mentioned in one of your specific comments, a short source distance does not necessarily imply short-range transport, because the air parcels may have moved with the cyclone as it travelled. Therefore, we agree that the terms "local" and "remote" can be misleading in this context. To avoid this confusion we have made the following changes in the manuscript:

1. To avoid implying that the cyclones are stationary we revised 'coming from' to 'originates', e.g. in line 37:

   *By tracking precipitating air parcels back in time we find that the moisture originates from areas of strong ocean evaporation.*

2. We better defined terms like "local" and "remote", by emphasising source origin relative to precipitation location when quantifying/describing distances. For example, we revised line 27 as follows:

   *As cyclones mature, distances between the moisture source and the location where the moisture rains out decrease*

3. Clarify that the cyclone is moving and that air parcels can undergo long-range transport even when the source distance is small. We briefly mention this in line 101:

   *The moisture that precipitates in the cyclone centers may have originated locally and remained close to the center as it propagated, or it may have been sourced from more distant areas.*

   2) Q2 states that cyclone-relative moisture sources will be determined. While cyclone-relative evaporation is shown (figure 11) I could not find any cyclone relative moisture source analysis in the paper. In a cyclone-relative frame of reference, it would be necessary to subtract the cyclone motion from that of the Lagrangian trajectories. However, since many of the cyclones will not exist for 8-days I'm not sure how this would be performed. As noted in comment 56, simply showing cyclone-relative streamlines or wind vectors overlaid in figure 11 would go some way to answering the question, but as it is, I do not think the authors have answered part of Q2.

We thank the reviewer for pointing this out and agree that our current Q2 is not fully answered with our analysis. Our current performed analysis, which is based on output that provides moisture uptake as an accumulated footprint over the eight-day period rather than at individual time steps, limits us to determine the cyclone-relative moisture sources. An alternative method, such as subtracting the cyclone motion from the Lagrangian trajectories to estimate relative source locations, is indeed not

straightforward since many of the cyclones will not exist for eight days, as is correctly pointed out by the reviewer.

To ensure our research question is fully addressed, we have revised Q2 to focus specifically on identifying the moisture sources of the precipitating air parcels in **geographical frame of reference** only: *"What are the geographical moisture sources of the precipitating air parcels?"* While we have also gained some insights into the **cyclone-relative frame of reference** from the case study and from analysing surface evaporation on a cyclone-centered grid, we present these findings more as preliminary explorations or first-order estimates, rather than a definitive answer to the research question.

It is also important to note that the analysis of surface evaporation on a cyclone-centered grid has become more robust since we included the cyclone relative winds to figure 11 (see comment 56, review 1 for the updated figure, and comment 30, review 2 for the interpretation), as suggested by the reviewer in Comment 56. This addition provides a first-order estimate of the direction of moisture transport and helps assess whether regions of high surface evaporation are likely to contribute moisture to precipitation in the cyclone center, thereby identifying them as potential moisture sources.

Specific comments
    3) Line 14: The data source should be referred to here, i.e. ERA5

Thank you for this comment. The line has been revised as follows:

*For this purpose, 8-day backward trajectories are calculated for all air parcels in the vicinity of cyclone centers with ERA5 reanalysis data, for a subset of the most intense summertime cyclones over the North Atlantic.*

    4) Line 17: Do the authors mean 'beneath the warm conveyor belt', rather than 'in the WCB' or are they suggesting that some precipitation associated with the WCB evaporates within the airflow?

Thank you for pointing this out. Our intention was to express that the bulk of the precipitation occurs near the cyclone center and is co-located with the WCB. We agree that the phrasing could be more precise and have revised the sentence to say "beneath the warm conveyor belt". We have revised line 20, and made the same change in Line 699.

    5) Line 24: It is not clear what the authors mean by remote and localised sources. Are they referring to distance relative to the cyclone position at some fixed point along its track, or relative to a moving cyclone?

We have revised this line in the manuscript to clarify what we mean. For a more detailed discussion, please see our reply to the first general comment above.

    6) Line 32: The authors refer to moisture 'coming from' specific source regions. This suggests somehow that the atmosphere is stationary, and that the moisture is transported through that stationary atmosphere. Perhaps 'originate' would be more specific term to use?

We have revised this line in the manuscript and changed it to 'originates from'. For a more detailed discussion, please see our reply to the first general comment above.

7) Line 38: There are many earlier papers that describe the role of extratropical cyclones in redistributing energy, moisture and momentum. Please refer to original sources where possible.

Thank you for this comment. Instead of Field and Wood (2007), Neu et al. (2013) and Papritz et al. (2021), we have now referred to Chang et al. (2002), Hartmann (1994) and Peixoto and Oort (1992):

*Extratropical cyclones make up an important component of the global atmospheric circulation by redistributing energy, moisture and momentum from lower latitudes to the polar regions (Chang et al., 2002; Hartmann, 1994; Peixoto and Oort,1992).*

8) Line 63: This sentence seems strangely worded to me since it implies that the convergence on the warm side of the cold front is coincidentally co-located with the WCB. In reality, the crossfront ageostrophic circulation causing convergence near the surface increases frontogenesis along the cold front strengthening the horizontal part of the WCB. The convergence also leads to ascent strengthening the vertical part of the WCB. Thus, there is no coincidence but direct cause and effect (see Dacre and Clark, 2025) Dacre, H.F. and Clark, P.A., 2025. A kinematic analysis of extratropical cyclones, warm conveyor belts and atmospheric rivers. npj Climate and Atmospheric Science, 8(1), p.97.

Thank you for pointing us to the Dacre and Clark (2025) paper, which we were not aware of. We agree that our original phrasing may have suggested a coincidental co-location between low-level convergence and the WCB, rather than explaining the underlying relationship. We have revised the text in the introduction (lines 73-79) to reflect this cause-and-effect relationship more accurately and have incorporated a reference to Dacre and Clark (2025) to support this explanation.

*Within these systems, there is low-level convergence of moist and warm air within the warm sector – the region of warm air between the cold and warm front that extends from the cyclone center in a northeasterly direction (Ahrens, 2009; Chang and Song, 2006; Papritz et al., 2021). This convergence is driven by a cross-front ageostrophic circulation on the warm side of the cold front, which strengthens the cold front and enhances upward motions (Dacre and Clark, 2025). The rising airflow is often termed the warm conveyor belt (WCB), and it is intrinsically linked to the extratropical cyclone (Eckhardt et al., 2004; Pfahl et al., 2014).*

9) Line 65: Here it is stated that large-scale vertical motion lifts the WCB above the warm front. What is causing the large-scale vertical motion? Are the authors referring to divergence aloft downstream of a planetary-scale Rossby wave or some other mechanism?

With 'large-scale vertical motion' we are referring to the ascending motion in the warm sector, which is driven by the cross-front ageostrophic circulation. As you also

point out in the previous comment, this circulation leads to convergence near the surface on the warm side of the cold front, which strengthens the front and supports ascent. This is the same mechanism that fuels the WCB, eventually lifting it above the warm front. In the manuscript, we have revised this part of the introduction, emphasizing the underlying cause of vertical motion in the WCB, without needing to link it to divergence aloft downstream of a planetary-scale Rossby wave or some other mechanism, as we also do not provide any evidence that supports this hypothesis. We have also reworded line 86 itself:

*Over time, the WCB can overrun the warm front and reach the upper troposphere…*

> 10) Line 71: I do not see follow the logic here. The fact that WCB ascend from the boundary layer to the upper-troposphere in 2-days does not imply that they transport more moisture to the upper troposphere than shallow convective processes, or that they are the main sink of precipitable water in the atmosphere. Please add more details to explain this sentence.

Thank you for pointing this out. We agree that the original phrasing implied a causal relationship. Our intention was not to suggest that the rapid ascent of WCBs alone makes them the main sink of precipitable water. Rather, we wanted to highlight the efficiency with which WCBs transport moisture vertically from the boundary layer to the upper troposphere over relatively short timescales. We have revised this part of the introduction (lines 92-98) to clarify this point.

*In the WCB, moisture can be transported from the boundary layer to the upper troposphere in roughly two days (Eckhardt et al., 2004). This level of efficiency, in combination with the extent of the uplift, results in a substantial amount of precipitation being generated in the WCB.*

> 11) Line 74: The WCB is generated by the same mechanisms that are responsible for the formation of extratropical cyclone fronts, therefore they are 'intrinsically linked' not 'often linked'.

Thank you for this suggestion. Line 78 has been revised as follows:

*The rising airflow is often termed the warm conveyor belt (WCB), and it is intrinsically linked to the extratropical cyclone (Eckhardt et al., 2004; Pfahl et al., 2014).*

> 12) Line 78: If you are saying that moisture is 'transported from remote regions to the centre of the cyclone' then are you assuming that the cyclone is stationary? If not, you need to consider that the moisture may have remained local to the centre of the cyclone, but that it moved with the cyclone as it travelled.

We have revised this line in the manuscript to clarify what we mean. For a more detailed discussion, please see our reply to the first general comment above.

> 13) Line 81: ARs are generated by the same processes that create the WCB airflow, thus they do not exist without cyclone airflows. The apparent

distinction is merely a matter of choice of threshold and the different methods used to identify them.

Thank you for raising this point. We agree that the apparent distinction between WCBs and ARs largely stems from differences in how they are defined and identified. Therefore, we have revised this part of the introduction to clarify that WCBs and ARs are indeed generated by the same underlying mechanisms and often have spatial overlap. We now explicitly note that WCBs are defined as strongly ascending cyclone-relative airflows, while ARs are identified as Earth-relative moisture plumes.

*ARs are long and narrow filaments of high vertically integrated moisture (Pfahl et al., 2014). Although WCBs and ARs are typically defined differently – WCBs as strongly ascending cyclone-relative airflows and ARs as Earth-relative moisture plumes (Dacre et al., 2019) – they are closely linked to cyclone development and spatial overlap often exists (Dacre and Clark, 2025; Knippertz et al., 2018).* (lines 106-111)

14) Line 142: Are there any limitations to using ERA5 precipitation in this study?

There are indeed some limitations in using ERA5 reanalysis data, including its coarse spatial resolution, the parameterization of convective processes and associated vertical motion, and an underestimation of intense, localized precipitation. In our study, this particularly affects the representation of diabatic processes, which typically occur at scales smaller than the model can resolve and are therefore parameterized (Binder et al., 2020). For instance, WCBs over the North Atlantic are often linked to deep convection (Oertel et al., 2019), which may be underestimated in ERA5, potentially also reducing the ascending motion. However, ERA5 benefits from data assimilation, which integrates observational data into the model and helps compensate for some of these shortcoming. As shown by Binder et al. (2020), ERA5 can still capture the large-scale structure of WCB clouds, including their location and thermodynamic cloud phase, which is sufficient for the scope of our study.

Moreover, ERA5 offers a physically consistent dataset with global coverage and high temporal resolution, which is of great importance when studying cyclone life cycles in regions with sparse observational data, such as over the ocean. Given current data availability, it remains the most suitable dataset for this type of analysis. Finally, the main advantage of ERA5 is that the meteorological fields remain physically consistent – that is, that temperature, moisture, and vertical motion evolve coherently with the precipitation field.

We have slightly revised the relevant paragraph (Line 178-181) in the manuscript to more clearly acknowledge these limitations.

15) Line 183: Over what period is the intensification measured? 24 hours or the entire cyclone lifecycle?

The intensification is measured from the time of cyclogenesis to the time of maximum intensity. We have updated the text (also incorporation comment 11 of reviewer 2) to reflect this definition more clearly.

*To quantify this, we apply the definition of a moving cyclone proposed by Eckhardt et al. (2004), which defines a moving cyclone as one that during its life cycle travels at least 1000 km and intensifies by more than 10 hPa. Based on this criterion, the selection is limited to three stationary cyclones, while the remaining 685 cyclones all travel more than 1000 km and meet the intensification threshold.* (lines 218-221)

16) Line 196: In the figure caption what does the maximum depth refer to? Is this the maximum deepening in 24 hours? Or the total deepening over the cyclone lifecycle?

Thank you for this question. In this study, we use the cyclone depth as a measure of cyclone intensity, defined as the pressure difference between the cyclone center and the cyclone edge (Line 195). The maximum depth thus refers to the largest pressure difference reached at any point during the cyclone's life cycle — in other words, the cyclone's time of maximum intensity. This is calculated from hourly data along each track. We have changed 'maximum depth' into 'maximum cyclone depth' in the figure caption to make it more clear.

17) Line 212: Frontal circulations occur within 500km of the cyclone centre, particularly in the developing stages before frontal fracture occurs.

We agree that frontal circulations often occur within 500 km of the cyclone center, particularly during the early stages of development. Our intention was not to exclude all frontal circulations, but rather to avoid including precipitation that is too far removed from the cyclone core and may be influenced by nearby orography or unrelated synoptic features. We therefore consider it appropriate to include frontal circulations within this radius, as they are typically linked to dynamical ascent associated with the cyclone itself.

18) Line 214: Without observations, how do you know that the precipitation is allocated to the 'right' moisture sources?

Thank you for pointing this out. We acknowledge that the term *"right"* may be misleading, as we do not have observational evidence to confirm the correctness of the identified sources. We have therefore revised the sentence to clarify that the focus is on identifying *specific* moisture source regions, without implying absolute correctness.

*The number of days is chosen based on three factors: (1) computational cost and computer power, (2) numerical accuracy, and (3) the ability to allocate a significant proportion of precipitation to specific moisture sources.* (lines 254-256)

19) Line 227: Is the first timestep, the first 3 hours?

Yes, the first timestep is indeed the first three hours. We have revised the sentence to make this more clear:

*… and where a significant amount of moisture is lost during the first three hours back in time ($\Delta q < -0.1$ g kg-1 3h-1).*

20) Line 233: Is the start point of the trajectory the cyclone centre, or the location of the trajectory 8-days earlier?

The trajectory start is the location where the air parcel is released (at t = 0 h), so in the cyclone center. To improve clarity, we avoid using the term 'trajectory start' and instead refer to the physical location of the precipitation, i.e. 'the cyclone center'. This terminology better reflects the context of our analysis and avoids confusion with the direction of the backward trajectories.

We made the following changes (also taking into account comment 18 of reviewer 2):
*Line 269: Since we are interested in the moisture sources of precipitation, we first select the precipitating trajectories within the cyclone center, defined as trajectories where the relative humidity (RH) exceeds 80% at the trajectory start (t = 0 h), …*
*Line 276: Increases in specific humidity indicate moisture uptakes, while decreases (prior to reaching the cyclone center) suggest intermediate precipitation.*
*Line 298: In contrast, the moisture that the air parcel has gained at t = -168 h partly precipitates during the intermittent precipitation event, and will have a smaller contribution to the precipitation at the cyclone center.*
*Figure 2: 'Trajectory start' in the top diagram is changed to 'Origin'.*

21) Line 237: Is there sensitivity to this choice of threshold?

For this study, we used a threshold value of 0.075 g kg$^{-1}$ 3h$^{-1}$, based on the threshold of 0.025 g kg$^{-1}$ h$^{-1}$ that was proposed by Papritz et al. (2021). We did not experiment with the choice of threshold ourselves, as we closely followed the methodology of Papritz et al. (2021) and had no intention of deviating from it. However, the choice of threshold is extensively discussed in Sodemann (2025) and the authors of Papritz et al. (2021) did some experimenting as well. The threshold is introduced primarily to reduce the computational cost of the moisture source diagnostic by discarding small fluctuations of specific humidity along trajectories that have little impact on the identified moisture sources, and to filter out small fluctuations that are due to numerical noise.

A typical and well-tested value for the midlatitudes is 0.2 g kg$^{-1}$ 6h$^{-1}$ . It is possible that this value may need to be higher for (sub)tropical regions, while in colder and drier conditions, such as in polar or high-altitude regions, a lower threshold of 0.1 g kg$^{-1}$ 6h$^{-1}$ has been suggested (Sodemann and Stohl, 2009). Given that in our study we are using shorter time intervals and our domain covers a part of the subtropics, we have chosen to adopt the threshold value of 0.075 g kg$^{-1}$ 3h$^{-1}$. In principle, a low threshold could lead to more noise being included as moisture uptakes, while a higher threshold leads to a higher chance of discounting previous moisture uptakes, and thus may induce a bias towards more local moisture sources. However, the authors of the wintertime study found that applying or omitting the threshold led to a change in attributed precipitation by much less than 1%, indicating negligible sensitivity.  Based on this, we consider the effect of the threshold to be negligible in our case as well.

22) Line 274: Does the maximum depth refer to a deepening of 35.2 hPa over the entire cyclone period in which case is it a maximum or should it be a total? Or

is it the maximum deepening over a 24-hour time window, in which case the time period should be included.

The 'maximum depth of 32.5 hPa' refers to the largest pressure difference between the cyclone center and its edge – not a deepening over the full period, nor a 24-hour deepening rate. The time of cyclogenesis is mentioned, as is the time it takes to reach this maximum depth. Therefore, we believe that no revision to the sentence is necessary.

23) Line 293: What is meant by 'keep the moisture in place'? Typically, the winds near cyclones are relatively strong.

Thank you for pointing this out. We did not mean to suggest that the winds near cyclones are weak. Instead, we suggest that the winds over the moisture source regions – which are located further away from the cyclone itself – can be relatively weak prior to the cyclone's arrival. This allows for the evaporated moisture to remain near its sources instead of being transported away, which facilitates a build-up of moisture. As the developing cyclone propagates into the region, the moisture will be sucked into the warm sector.

We have revised the text and moved it to section 3.2 (lines 424-427), where it makes more sense.

*Upon descending into the boundary layer, they gained substantial moisture through mixing. Moreover, surface winds are relatively weak prior to the arrival of the secondary cyclone, facilitating a build-up of moisture. As the developing cyclone propagates into the region, the accumulated moisture is likely entrained into the warm sector and connected into the WCB and the AR through a feeder airstream, according to the mechanism proposed by Dacre et al. (2019).*

24) Line 295: The authors might be interested in reading Demirdjian et al. (2023) which describes this process very nicely.
Demirdjian, R., Doyle, J.D., Finocchio, P.M. and Reynolds, C.A., 2023. Preconditioning and intensification of upstream extratropical cyclones through surface fluxes. Journal of the Atmospheric Sciences, 80(6), pp.1499-1517

Thank you for pointing us to the Demirdjian et al. (2023) paper, which we were not aware of. This paper describes how surface fluxes contribute to the interaction between clusters of cyclones, specifically by showing how surface sensible and latent heat fluxes moisten the airstream that connects a primary cyclone to an upstream developing secondary cyclone. This airstream feeds moist air parcels into the AR and WCB, and is closely related to the feeder airstream concept discussed in Dacre et al. (2019).

After reading the paper, we revisited our case study with this conceptual model in mind and decided to expanded our description of the surface fluxes in the cyclone's cold sector and added a reference to Demirdjian et al. (2023) in section 3.1 of the case study.

*The region where colder air is advected over a warmer underlying sea surface represents the cyclone's cold sector, and is located to the southwest of the cyclone center. Here, we observe strong upward fluxes of sensible and latent heat (Fig. 4a-c), which will result in a moistening of the boundary layer, thereby increasing the buoyancy and deepening the boundary layer (Demirdjian et al., 2023; Papritz et al. 2015).* (lines 337-340)

In the idealized model setup in Demirdjian et al. (2023), the secondary cyclone develops upstream and there is a well-defined primary-to-upstream (PTU) feeder airstream that passes along the southern flank of an anticyclone through a region of strong latent and sensible heat fluxes. This is not evident in our case. We therefore chose not to directly link it to the PTU airstream, but included a discussion of the primary-secondary cyclone configuration and how it differs from the cyclone cluster case described by Demirdjian et al. (2023):

*An examination of the SLP contours reveals that C1 is embedded within a larger wave train, with downstream alternating cyclones and anticyclones. […] To the west of this region of intense surface evaporation, we observe the genesis of C2 (Fig. 3c). Given that C2 does not develop upstream as part of the prevailing wave train and the cyclone is considerably smaller in scale compared to C1, it seems likely that the cyclogenesis of C2 is the result of a small-scale disturbance along the baroclinic jet. Therefore, this particular case appears to deviate from the typical cyclone cluster as described in the literature (e.g., Bjerknes and Solberg, 1992; Demirdjian et al., 2023) in that the two cyclones do not follow a similar track, but instead, C1 becomes stagnant and C2 propagates to the south of it and eventually overtakes it.* (lines 326-327 and lines 340-346)

25) Line 300: Figure 3 uses a colour scale that makes the fluxes on the warm side of the Gulf Stream appear to be positive. I had to look very carefully to see the 'sharp transition' from negative to positive. I suggest using a divergent red-blue colour scale instead.

Thank you for pointing this out. In the original figure, the red–blue colour scale included two similar shades of yellow in the middle, making it difficult to distinguish positive from negative values. This made it difficult to interpret whether the fluxes on the warm side of the Gulf Stream Front and in the subtropics were positive or negative. We have therefore revised the colour scale so that the warm orange/yellow tones are more clearly distinguishable from the cooler blue shades. The updated version of the figure is shown below.

[Figure]

**Figure 3: Sea-air potential temperature difference ($\theta_{SST} - \theta_{900hPa}$, in K) associated with an East Atlantic cyclone, for t = -72 h (4 May 2019, 10:00 UTC) to t = 24 h (8 May 2019, 10:00 UTC) in 12-hourly intervals. Additionally shown are SLP contours (in intervals of 5 hPa; dark green lines), the cyclone track (white line), and the cyclone center location (white cross labelled C2; visible only from -48 hours onward, as cyclogenesis did not occur until -59 hours). Label C1 represents the location of the preceding primary cyclone.**

26) Line 303: What is the surface evaporation elevated with respect to? No climatology of evaporation is shown so it is not clear if this is a climatological response or related to the passage of the cyclone.

This and the following comment, indicate that the 'evaporation hotspots' need to be more visible. To address this, we slightly adjusted the colour scale so that lighter shades (representing lower values) and darker shades (representing higher values) are more visually distinct. The revised figure is shown below.

[Figure]

**Figure 4: Surface evaporation (in mm day⁻¹) associated with an East Atlantic cyclone, for t = -72 h (4 May 2019, 10:00 UTC) to t = 24 h (8 May 2019, 10:00 UTC) in 12-hourly intervals. Additionally shown are SLP contours (in intervals of 5 hPa; dark green lines), the cyclone track (white line), and the cyclone center location (white cross labelled C2; visible only from -48 hours onward, as cyclogenesis did not occur until -59 hours). Label C1 represents the location of the preceding primary cyclone.**

We also explored alternative visualizations by plotting anomalies instead of absolute values (see figure below). This approach helps clarify how the regions compare to the climatology and reveals the effect of the passage of a cyclone. However, with this type of visualization (and when applying normalization), the main evaporation hotspots that emerge are located in the cold sectors of the cyclones. In contrast, the regions with climatologically high evaporation, such as the warm side of the Gulf Stream Front and the subtropical high, do not appear as anomalies, even though they are important sources of moisture. For this reason, we have chosen to include the absolute values rather than the anomalies.

[Figure]

(a) t = -72 h  (b) t = -60 h  (c) t = -48 h
(d) t = -36 h  (e) t = -24 h  (f) t = -12 h
(g) t = 0 h  (h) t = 12 h  (i) t = 24 h

−10  −8  −6  −4  −2  0  2  4  6  8  10
Surface evaporation anomalies [mm day$^{-1}$]

27) Line 305: Figure 4 uses a colour scale in which it is difficult to observe the 'evaporation hotspots'. Perhaps anomalies or normalised anomalies could be shown? I.e. subtract the climatological field and divide by the standard deviation of the climatology at each location. Then it would be possible to see how large anomalies are relative to the climatological mean, i.e. 1 std deviation larger/smaller than climatological mean?

See comment above.

28) Line 318: No evidence is provided to support the statement that the dynamical flow is setup by the primary cyclone, or that significant moisture accumulates ahead of the developing cyclone.

There are parts of the manuscript where it seems that we have been too speculative (see also comment 33 and comments 28 and 30 by reviewer 2). In this case, the formulation is too strong, so we have decided to remove this statement and instead focused directly on the precipitation and moisture sources.

*For this East Atlantic cyclone, precipitation peaked 12 hours before the cyclone reached its maximum depth, after which  it notably decreased (not shown). The*

*corresponding moisture sources of the precipitation falling at three distinct times during the cyclone life cycle – the intensification phase (t = -24 h), the mature stage (t = 0 h), and the decay phase (t = 24 h) – are shown in Fig. 5.* (lines 376-378)

29) Line 326: Here the authors refer to the sources of moisture moving eastward. Presumably this is related to the fact that the cyclone is also moving eastward?

Yes, the eastward shift of the moisture sources is indeed a consequence of the cyclone moving eastward. The original sentence already describes the shift in both the cyclone's position and the moisture sources, but their connection could be stated more clearly.

*As the cyclone moves eastward throughout its life cycle, the sources of moisture undergo a similar shift, moving in an eastward direction and being closer to the cyclone center (Fig. 5b, c), while most of the uptakes remain aligned with the north-eastward extension of the Gulf Stream Front.* (lines 383-385)

30) Line 332: How are long-range and local defined. Do they refer to a distance travelled by the trajectories over the 8-day period or are they the distance relative to a fixed feature, or the moving cyclone?

By "long-range," we mean that the moisture originates far from where the precipitation eventually falls, whereas "local" indicates a nearby source. Rather than rephrasing this in the manuscript, we have chosen to remove this interpretation and focus on the actual calculated source distances. This allows us to avoid speculation about the underlying processes, which are discussed later in this section.

31) Figure 5: It is useful to show the cyclone track, but I wonder if the position of the cyclone at 24hr intervals preceding the trajectory start time should be shown instead?

Thank you for this suggestion. It is indeed a good idea to also show the position of the cyclone center in 24-hour intervals prior to the time of precipitation, so that these locations can be compared with the location of the moisture sources. The revised figure, including these locations, is shown below.

[Figure]

**Figure 5: Moisture source footprint of uptakes contributing to precipitation in the cyclone center of an East Atlantic cyclone, for (a) t = -24 h (6 May 2019, 10:00 UTC), (b) t = 0 h (7 May 2019, 10:00 UTC), and (c) t = 24 h (8 May 2019, 10:00 UTC). Additionally shown are MSLP (in intervals of 5 hPa; dark green contours), the cyclone track (white line), and the cyclone center location at the time of the precipitation (white cross). Cyclone center positions at 24-hour intervals prior to the precipitation are indicated by colored dots. The weighted mean accounted fraction and source distance are given in the top right of each panel.**

32) Line 351: I don't think that the trajectory figures 6a-c are very legible. It's difficult to identify the different pathways. If 80% of the moisture uptake occurs within 4 days of precipitation (line 515), would it be better to show shorter trajectories in figure 6 to make it more legible?

The rationale behind plotting the eight-day trajectories is to show the entire air parcel history. In the figure below, we have plotted four-day trajectories. Such shorter trajectories do not show that the parcels originate from the primary cyclone, travel all the way from the North American continent, or move along the Greenland coast. So we prefer not to include this figure in the manuscript.

[Figure]

However, we acknowledge that the current figure is not very legible. To improve clarity, we have decided to plot only the 20% of trajectories associated with the most intense precipitation, like we do for figures 6d-f. The revised figure is shown below.

[Figure]

33) Line 354: How do you know the parcels were lifted by convection?

There are parts of the manuscript where it seems we have been too speculative (see also comment 28 and comments 28 and 30 by reviewer 2). We have therefore revised this section so that it focusses more clearly on the key takeaways of Figure 6, avoiding interpretations not supported by evidence. Specifically, rather than

suggesting that the air parcels originating from the Gulf Stream region are located in the mid-troposphere due to convective uplift (for which we indeed have no evidence), we have emphasized the broader evolution of how the trajectories are distributed vertically.

*During the intensification phase, the precipitating air parcels originate from a range of vertical levels (Fig. 6d) and appear relatively unorganized, also in a geographical frame of reference (Fig. 6a). Some trajectories flow along the Gulf Stream front, where the strong SST gradient drives moisture uptake far from where the eventual precipitation falls.* (lines 413-416)

   34) Line 355: If the parcels are lifted by convection, how do they converge at the surface?

The trajectories we describe are those that eventually precipitate near the cyclone center and therefore must undergo ascent in the hours leading up to precipitation, which is preceded by convergence near the surface. In the days prior to this convergence, the trajectories originate from a range of vertical levels. Those descending from higher altitudes are likely associated with a dry intrusion, while others in the mid-troposphere may have experienced earlier lifting – potentially due to shallow convection in regions such as the Gulf Stream. However, all these parcels are eventually drawn towards the warm sector within the cyclone center due to the low-level convergence.

In light of this and the previous comment, we have removed the part on convective uplift and tried to better describe the convergence at the surface.

*Regardless of the origin, all parcels are eventually drawn into the warm sector near the cyclone center by low-level convergence, where they ascend rapidly just before the onset of precipitation.* (lines 456-458)

   35) Line 359, 370, 570: Here and elsewhere, the descending trajectories behind the cyclone cold front are known as the dry intrusion. It might be helpful to refer to this term to enable links with previous work.

Thank you for this suggestion. We agree that it is helpful to link the trajectories that we have calculated to known cyclone-relative airflows, such as the dry intrusion. However, we chose not to explicitly classify them as such, since many previous studies adopt threshold-based definitions (e.g., Raveh-Rubin, 2017, who defines dry intrusions as descending at least 400 hPa within 48 hours), when we prefer not to adopt such strict definitions. That said, we believe that it is a good idea to refer to the term dry intrusion with reference to the conveyor belt model of Browning (1990, 1997), where it is more broadly defined as the descending air stream behind the cold front, and made these changes.

*Line 421: Some of these parcels experienced intermittent precipitation, remained aloft and subsequently joined cold, dry air in the mid-troposphere. Together, they descended behind the cold front of the primary cyclone, resembling the dry intrusion airstream described in Browning's conveyor belt model (1990, 1997).*

*Line 670: This environment allows cold, dry air that is descending behind the cold front – commonly referred to as the dry intrusion airstream (Browning 1990; Browning 1997)…*

36) Line 360, 578-9: Likewise, the air behind the primary cyclone cold front which is then 'fed into the secondary cyclone' is known as the feeder-airstream. It might be helpful to refer to this term to enable links with previous work.

Thank you for this suggestion. We agree that it is a good idea to term the air behind the primary cyclone cold front the feeder-airstream, with a reference to Dacre et al. (2019). We made the following change, also incorporating text that was considered unclear in comment 23 and makes more sense in this context.

*Upon descending into the boundary layer, they gained substantial moisture through mixing. Moreover, surface winds are weak prior to the arrival of the secondary cyclone, facilitating a build-up of moisture. As the developing cyclone propagates into the region, the accumulated moisture is likely entrained into the warm sector and connected into the WCB and the AR through a feeder airstream, according to the mechanism proposed by Dacre et al. (2019).* (lines 424-427)

37) Line 366: Use of the phrase 'came from' suggests air travelling towards the cyclone from the primary cyclone. Would a better description be that this air was swept up by the secondary cyclone as it travels into the region of enhanced moisture?

We prefer 'originate from' as an alternative for 'came from' to avoid implying stationarity. We have also added the suggested alternative to improve the sentence even more.

*Both airstreams originated in the Gulf Stream region but instead originated from the primary cyclone or further north along the Greenland coast before being swept up by the secondary cyclone.*

38) Line 410: You have not shown evidence that maximum vertical motion occurs before minimum MSLP? I think maximum precipitation typically occurs before maximum dynamical intensity due to limited moisture availability as cyclones travel polewards.

Thank you for pointing this out. We have indeed not shown evidence that maximum vertical motion occurs before minimum MSLP. While precipitation is influenced by both vertical motion and moisture availability, and it is plausible that poleward-moving cyclones are limited by moisture availability leading to an earlier peak in precipitation, we have not analysed this relationship in detail. To avoid overinterpretation, we will remove the statement about the timing of maximum vertical motion, as it is speculative and does not add essential insights beyond what is already said.

39) Line 429: Can you concentrate precipitation. Perhaps extending would be a better word to describe the shape of the precipitation pattern?

We agree that 'concentrate' might be somewhat confusing. We simply mean that the highest precipitation rates are found near the cyclone center. Therefore, we have revised the sentence as follows:

*In the Gulf Stream region, the highest precipitation rates are found near the cyclone center, displaying a near-symmetrical structure (Fig. 7a-c).*

    40) Line 438: The precipitation does not significantly drop for Gulf Stream cyclones since the moisture availability remains high, even in the decay phase. By contrast, precipitation drops significantly for the other regions due to reduced dynamical forcing and reduced moisture availability.

Thank you for providing this additional interpretation. We have made a revision to the sentence, incorporating that the precipitation does not significantly drop for Gulf Stream cyclones, likely due to the high moisture availability.

    41) Figure 7: Much of the precipitation pattern in the decaying phase is outside the domain shown. Could a larger domain be used to capture the full shape of the cyclone precipitation pattern in this phase?

We agree that much of the precipitation pattern is outside of the circular domain. However, our focus is specifically on precipitation in close vicinity of the cyclone center, to ensure that the precipitation is indeed dynamically and physically related to the cyclones. By increasing the domain we would risk including precipitation in the cold sector or along trailing cold fronts – features that can extend far southwards, and may be influenced by the larger upper-level trough or have their own dynamics. In addition, we prefer not to use a larger domain, since a one-to-one comparison with the winter would no longer work.

    42) Line 469: The authors describe the eastwards shift in the moisture sources as 'slight'. This is a subjective word. Could they calculate the magnitude of the eastward shift? This could then be compared with the eastward shift in the mean cyclone position to see if they are comparable.

Thank you for the suggestion to compare the eastward shift in moisture sources with the displacement of the mean cyclone center. To address this, we calculated the spherical center of mass of the mean moisture footprints at all timesteps within the study period (−24 to +24 h, in 3-hourly steps) and compared its movement with that of the mean cyclone center. Both were characterized in terms of propagation direction and distance traveled. We also quantified the compactness of the moisture footprint, by calculating the weighted mean standard deviation of the source latitude and longitude positions. The results are summarized in the figure below, which we will include in the supplementary material.

[Figure]

**Figure A4: Movement of the moisture footprint and the mean cyclone track, characterized in terms of (a)-(d) propagation direction, (e) distance travelled, and (f) moisture footprint spread, which is defined as the weighted mean standard deviation of the source latitude and longitude positions. The coordinates of the moisture footprint are found by calculating the spherical center of mass of the mean moisture footprints. We then used the great-circle distance to track its movement and compared it with that of the mean cyclone center.**

In the manuscript, we have replaced line 469 and the subsequent paragraph with the following text:

*Another important observation of the moisture source footprint is its large spatial extent, especially during the intensification phase. This suggests that North Atlantic cyclones can access moisture reservoirs distributed across large parts of the ocean, rather than depending solely on moisture that has evaporated in close proximity of the cyclone center. As the cyclones evolve, the spatial distribution of the moisture footprints becomes more compact, particularly for those that reach their maximum intensity in the eastern part of the North Atlantic. This is shown by a decreasing standard deviation of the sources' latitude and longitude positions (Fig. A4f). As the footprints become more compact, they also shift in space. Across all subregions, the moisture sources remain aligned with the warm side of the Gulf Stream Front, moving in a northerly direction, consistent with the movement of the mean cyclone centers (Figs.8 and A4a-d). During the decay phase, however, the mean moisture footprints in the Labrador Sea and Nordic Seas tend to shift more northward relative to the mean cyclone track. This north-eastward shift of the moisture footprint exceeds the movement of the mean cyclone center in all subregions except in the Gulf Stream region (Fig. A4e). This implies that, over time, moisture sources move closer to the cyclone center. At this stage, local moisture recycling becomes more important, although overall moisture uptake is lower than during the intensification phase.* (lines 550-564)

43) Line 469: The eastward shift is also evident for the Gulf Stream cyclones.

Thank you for pointing this out. We have replaced this sentence with new information (see previous comment).

44) Line 474: Local to what?

We have revised "local moisture" to "moisture that has evaporated in close proximity of the cyclone center".

45) Figure 8: Would it be possible to show the mean cyclone position at various stages in the cyclone lifecycle.

Thank you for this suggestion. It is indeed useful to show the mean cyclone track and the mean positions of the cyclone center at the different phases in the cyclone life cycle. We have added them to the figure.

46) Line 495: Within 1000km of what?

We have revised this line in the manuscript to clarify what we mean. For a more detailed discussion, please see our reply to the first general comment above.

47) Line 513: Would these terrestrial sources be the Great Lakes, or are they more widespread than this?

Zooming in on the moisture footprint of the all cyclones (see figure below), we observe that the terrestrial sources extend over a large part of eastern North America, not just the Great Lakes.

[Figure]

48) Line 526: Sources 1000-1500km away from what?

We have revised this line in the manuscript to clarify what we mean. For a more detailed discussion, please see our reply to the first general comment above.

49) Line 539: Ascending trajectories ahead of the cold front are part of the WCB airflow. It might be helpful to refer to them as the ascending part of the WCB to enable links with previous work.

Thank you for this suggestion. We agree that it is helpful to link the trajectories that we have calculated to known cyclone-relative airflows, such as the WCB. We have revised the sentences as follows:

*This ascent occurs primarily in the warm sector of the cyclone and is associated with the WCB airstream. As the air parcels are lifted in the WCB, they cool and reach saturation, leading to condensation and strong latent heat release that further enhances vertical motion.* (lines 631-633)

50) Line 540: What is the evidence for moisture accumulation near the cyclone centre?

We indeed show no evidence for moisture accumulation near the cyclone center, making this statement too speculative. Therefore, we have removed this part of the sentence. The new version of the sentence is given in the response to the previous comment.

51) Figure 11: These figures correspond well with figures 5 and 6 in Dacre et al. (2020), showing maximum latent heat fluxes behind the cyclone cold front as a result of large vertical gradients in specific humidity.
Dacre, H.F., Josey, S.A. and Grant, A.L., 2020. Extratropical-cyclone-induced sea surface temperature anomalies in the 2013–2014 winter. Weather and Climate Dynamics, 1(1), pp.27-44.

Thank you for pointing us to this paper. These figures offer a potential explanation for the region of strong surface evaporation in to the southwest of the developing cyclone, which is co-located with the cyclone's own cold sector. We have therefore included this explanation and a reference to the Dacre et al. (2020) paper in the following sentences (in which we also incorporate comment 35 and comment 31 of reviewer 2):

*One pronounced region of elevated evaporation appears southwest of the developing cyclone, co-located with the cyclone's cold sector. As shown by Dacre et al. (2020), the strong upward latent heat fluxes observed behind the cyclone's cold front are a result of sharp temperature and specific humidity gradients between the ocean surface and the overlaying air, with notably cooler and drier air at 10 meter. This environment allows cold, dry air that is descending behind the cold front – commonly referred to as the dry intrusion airstream (Browning 1990; Browning 1997) – to efficiently take up moisture that has evaporated from the ocean surface (Aemisegger and Papritz, 2018; Ilotoviz et al., 2021).* (lines 667-673)

52) Line 615: What about moisture evaporated in the subtropical high as referred to earlier in the paper?

The subtropical high is indeed associated with enhanced surface evaporation, as mentioned earlier in the manuscript. However, based on the current analysis, we cannot confirm whether this region actually contributes to the cyclone precipitation. Moreover, the cyclone-relative winds in figure 11 show no northerly component, indicating that the winds are not directed towards the cyclone center. Therefore, we have decided not to include it in the conclusion.

53) Line 623: Here the authors refer to 'delivery' of moisture to the cyclone. Again, this implies the cyclone is stationary and that moisture is transported towards it from 'remote sources'. Is this really what is happening?

We have revised this line in the manuscript to clarify what we mean. For a more detailed discussion, please see our reply to the first general comment above.

54) Line 630: Source distances from what?

We have revised this line in the manuscript to clarify what we mean. For a more detailed discussion, please see our reply to the first general comment above.

55) Line 650: Dacre et al. (2023) use a very different method to that used here and estimate a mean moisture residence time of 36 hours. Do you think this is due to methodological differences or because they focus on extreme winter cyclones where the ascent rates are typically higher than for summer cyclones?

Our estimated precipitation life time (approximately four days) is longer than the mean residence time of moisture associated with winter cyclone precipitation, which was found by Papritz et al. (2021) to be approximately three days using the same method. Therefore, we hypothesize that Dacre et al. (2023) estimated a mean moisture residence time of 36 hours, which is at least partly attributable to their focus on extreme winter cyclones. The ascent rates in these cyclones are typically higher than in summer cyclones, resulting in shorter moisture residence times. In addition, differences in methodological approaches will likely contribute to the discrepancy as well, as there can be significant differences between various moisture tracking methods (Benedict et al., 2024).

56) Line 657: You have not included a figure of cyclone-relative low-level winds so it is not possible to determine if a feeder airstream is present in your cyclone composites. Would it be possible to overlay cyclone-relative streamlines to see if the feeder airstream can be identified?

Thank you for this suggestion. We have added the cyclone-relative winds to the figure, since it gives a first-order estimate of the direction of moisture transport, and whether feeder airstreams exists. The revised figure is given below.

[Figure]

**Figure 11: Cyclone-relative composites of the surface evaporation (in mm day⁻¹) of all cyclones in the North Atlantic, for t = -72 h to t = 24 h in 12-hourly intervals. Surface evaporation is averaged over 12-hourly intervals centered on the given relative time t. Vectors of the cyclone-relative winds at 900 hPa are also shown, defined as the actual 900-hPa winds minus the 12-hour average cyclone propagation speed. Additionally shown are MSLP (in intervals of 5 hPa; dark green contours), the cyclone center location (black dot), and surrounding circle with a radius of 500 km (black circle). The number of cyclones contributing to the composite is indicated in the top right of each panel.**

However, instead of discussing the feeder airstream concept, which is difficult without a cyclone-relative analysis, we have chosen to revise the sentence (Line 789) to focus more on what is (not) observed in Figure 11.

*In the composites of all 688 cyclones, however, we could not identify a primary cyclone in the MSLP contours, nor a region of enhanced surface evaporation to the northeast.*

57) Line 671: This sentence suggests there are multiple ascending motions. Are the authors referring to frontal line convection as an alternative to the WCB airflow?

Thank you for pointing this out. We were simply referring to the ascending motion in the warm sector and in the WCB. We have therefore revised the sentence to use the singular 'motion' to avoid any confusion.

58) Line 676: Similar to what, the precipitation in winter cyclones?

Yes, similar to winter cyclones. To clarify this, we have revised the sentence as follows:

*… the higher moisture content in warmer air parcels results in similar precipitation to winter extratropical cyclones once saturation is reached.*

59) Line 680: How are you separating fronts and WCB airflow? They are generated by the same mechanism, so it seems strange to separate the precipitation associated with them. Perhaps you are referring to the horizontal and vertical parts of the WCB? Or to frontal line convection and WCB ascent?

Thank you for raising this. The original suggestion proposed to examine the contributions of different features to precipitation. In this context, we agree it is important to clarify whether we focus on *cyclone-related features* – in which case a more specific distinction should be made, for instance between frontal convection and WCB ascent, like you mention – or on broader *weather features*, as done in Konstali et al. (2024), who distinguish between fronts, cold air outbreaks, cyclones, and atmospheric rivers (or moisture transport axes). We have, however, removed the entire paragraph, but incorporating some of the relevant information earlier in the discussion when addressing the different precipitation-targeting methods (see community comment).

*This finding underscores a broader point that different feature-identification frameworks emphasize different aspects of the same storm system, even though many features share similar generation mechanisms. Consequently, this can result in different outcomes. Coll-Hidalgo et al. (2025) already explored this sensitivity; however, extending this research with systematic approaches that also capture more general weather features, such as ARs and fronts, using the methodology of Konstali et al. (2024), would further clarify each feature's contribution to precipitation in summer cyclones. A meaningful comparison can for example be made between precipitation attributed to fronts and that within the WCB footprint, as both are generated by the same underlying mechanisms.* (lines 778-784)

Typographical errors

60) Line 10: Why is the word 'extreme' in brackets?

Our intention was to indicate that while extratropical cyclones are often associated with extreme precipitation, this is not always the case — depending, for example, on how 'extreme' is defined. We decided to remove the brackets to more clearly highlight the potential for high-impact precipitation associated with these systems.

61)Line 42: Why is 'heavy' in brackets?

As in the previous comment, the word 'heavy' had been placed in brackets to reflect this nuance, but we have now removed the brackets to more clearly highlight the potential for high-impact precipitation associated with these systems.

62)Line 97: Here and elsewhere, it is more standard to refer to the singular 'precipitating water' rather than the plural 'precipitating waters'.

Thank you for this suggestion. We have changed 'precipitating waters' to either 'precipitating air parcels' or 'precipitating water'.

63)Line 101: 'How' should be 'whether' since you haven't determined this yet.

Thank you for pointing this out. It should indeed be 'whether'.

64)Line 164: You do not need cf.

Thank you for this comment. Instead of 'cf.', we added information on how the North Atlantic region is represented in the figure.

*… a cyclone must reach its maximum intensity within the North Atlantic region (defined by the black line in Fig. 1), …*

65)Line 171: The number 2 needs units.

Thank you for spotting this mistake. We have included the units hPa.

66)Line 216: I think interpolated would be a better description than 'traced'.

While we acknowledge that 'interpolated' could, in some contexts, be a more precise term, we have chosen to retain the use of 'traced'. The rationale behind this decision is to maintain consistency with the terminology employed in the trajectory calculation tool LAGRANTO, which has a specific function named 'trace' that derives atmospheric variables along air parcel trajectories. Using an alternative term here could cause confusion, particularly for readers familiar with the tool and its functionality.

**Review 2**

This paper adopts a cyclone-centred perspective to evaluate the moisture sources and transport pathways of North Atlantic deep cyclone precipitation in summer. The paper strongly builds scientifically and methodologically on a previous paper (Papritz et al. 2021) focussing on winter deep North Atlantic cyclone precipitation. The paper is well written and the main findings are interesting and related to

i)    moisture residence times being relatively constant of about 4 days throughout the cyclone life cycle;

ii)   the moisture sources of cyclones originating from the tropics and making extratropical transition being mainly located in the subtropics and midlatitudes, with very limited amounts coming from the tropics directly

iii)  contributions from different key geographical and cyclone-relative regions, such as land, the warm side of the Gulf Stream as well as evaporation from the cold sector of preceding cyclones, although the discussion mainly relates to geographic regions and not quantitatively to cyclone-relative regions.

I have a few minor comments mainly related to the writing and presentation of the results.

Thank you very much for reviewing our paper and your suggestions on how to improve the writing and presentation of the results. We appreciate your feedback and are glad that you found the main findings interesting. To improve clarity, we have emphasized the first two key findings more prominently in the conclusion section. We have also addressed all your specific comments as detailed below.

Comments 5, 24, 30, and 33 relate to Q2 regarding cyclone-relative moisture sources. Since reviewer 1 raised similar points, we refer to our response to their general comment  2 for a more detailed discussion.

1) Innovation: I think the paper could become a bit sharper in terms of its innovative contributions to science. In my reading, I got the impression that it was very closely following the preceding paper by Papritz et al. 2021 both in terms of scientific focus and methodological approach. The fact that summer deep cyclones are generally less studied has good reasons, they are rarer and less intense than winter cyclones. Therefore, the motivation for this study could be carved out a bit more convincingly. I do think there are good reasons to investigate summer cyclones separately, e.g. to investigate the dynamical impact of added moisture from the land sources, the role and moisture transport pathways related to cyclones from tropical origin, potential similarities with future warmer conditions with weaker baroclinicity also in winter, contrasts between cyclones over the ocean vs. over land… I encourage the authors to make a stronger case for their paper in the abstract and the introduction (e.g. at L. 51-55). The fact that summer cyclones are less studied does not make it a good reason to study them.

Thank you for raising this issue. The paper by Papritz et al. (2021) provides an essential foundation, since we have indeed adopted a similar methodological approach to allow for a direct comparison between summer and winter extratropical cyclones. However, this does not mean that the present paper cannot stand alone. Therefore, we agree that it is important to highlight why it is important to study summer storms separately.
We appreciate the reviewer's suggestions and agree that especially potential similarities with future warmer climates (having reduced baroclinicity) provides a strong motivation. We have revised the abstract and introduction to more clearly emphasize this motivation.

In the abstract (lines 11-14):
*Nevertheless, studying summer cyclones is particularly relevant in the context of climate change, as future warming is expected to increase atmospheric moisture while reducing baroclinicity. This makes present-day summer conditions an analogue for future winter cyclones and critical for understanding how summertime cyclones themselves may evolve in a warmer climate.*

In the introduction (lines 54-60 and lines 64-68):
*Given their significant role, accurately forecasting these precipitation extremes and understanding how they are projected to change under global warming has received increasing scientific attention. Improving the representation of extratropical cyclones in climate models therefore remains a key component of the work of the Intergovernmental Panel on Climate Change (IPCC) (Catto, 2016). Recent analyses of the latest generation of CMIP6 models, for example, provide new insights into the projected changes, including a decrease in cyclone frequency but more intense precipitation associated with them (Priestley and Catto, 2022). […] This lack of attention is particularly striking given the anticipated increase in humidity and decrease in baroclinicity resulting from future warming, conditions under which future winter cyclones may resemble present-day summer cyclones. A better understanding of these systems is therefore important for assessing both how they themselves may evolve in a warmer climate and how they can inform our understanding of future winter cyclones.*

2) L. 62: WCBs here I think Madonna et al. 2014 and Heitmann et al. 2024 should be referenced.
   And at L. 73 about the link between WCBs and cyclones: Binder et al. 2016.

Thank you for these helpful suggestions. We agree that it makes sense to refer to Madonna et al. (2014) and Heitmann et al. (2024) when introducing the WCB as a rising air stream responsible for precipitation, and to Binder et al. (2016) in the discussion of the link between WCBs and extratropical cyclones. We have added these references to our text, along with an additional reference to Browning (1990). Note: the original sentences have been modified in response to comments 8 and 11 from reviewer 1.

*The rising airflow is often termed the warm conveyor belt (WCB), and it is intrinsically linked to the extratropical cyclone (Eckhardt et al., 2004; Pfahl et al., 2014). As air ascends in the WCB, moisture condenses and rains out, releasing latent heat and thereby enhancing the ascent (Browing, 1990; Heitmann et al., 2024; Madonna et al., 2014). Consequently, the WCB is capable of influencing the dynamics of the cyclones – a role that has been underscored in the context of cyclone intensification by Binder et al. (2016).* (lines 78-85)

3) WCBs: throughout the paper the authors should be much more cautious with their definition of the WCB and clearly define what they mean. Usually, this airstream is defined as ascending by 600 hPa or more in 48 h. It is very likely that in summer the airstreams are ascending less (see also the substantially lower frequency of WCBs in summer over the North Atlantic). Also, the convective parts of the ascent are probably missed in the approach chosen by

the authors (Oertel et al. 2021) calculating the trajectories based on 3 hourly 3D wind fields on the still relatively coarse ERA5 grid.

Thank you for this comment. In response, we have revised the relevant sentences in the introduction to more clearly define both warm conveyor belts (WCBs) and atmospheric rivers (ARs), highlighting that these features are both airstreams within the warm sector of extratropical cyclones that start to ascent. However, rather than applying the strict trajectory-based definition of WCBs – defined as parcels ascending by at least 600 hPa within 48 hours – we have intentionally adopted a looser, more qualitative usage of the term in this study. Our focus is on the airstreams that transport warm, moist air poleward and upward, without explicitly identifying WCBs via trajectory thresholds. This is particularly relevant in summer, when, as you mentioned as well, ascent rates are generally lower. Instead, in Section 4.4, we discuss the presence of weaker ascent in these airstreams during summer, rather than suggesting that WCBs are absent.

Regarding the second part of the comment, we agree that convective components of ascent are likely underrepresented due to the use of 3-hourly 3D wind fields on the relatively coarse ERA5 grid. This could indeed result in an underestimation of rapid convective ascent, particularly during summer. We now explicitly acknowledge this limitation in Section 4.4. Nevertheless, previous studies (e.g., Binder et al., 2020) have shown that ERA5 is still able to capture the large-scale structure of WCB cloud features, including their location and cloud phase. Given that our analysis emphasizes broad structural and thermodynamic characteristics rather than detailed convective dynamics, we consider ERA5 adequate for the goals of this study.

4) L. 76-87: I don't understand the use of discussing atmospheric rivers in such great detail since they are not identified or discussed further in the results of this paper. I would suggest shortening and shifting the discussion on the potential role and link with ARs to the conclusion.

Thank you for your comment. We agree that the initial discussion of ARs in the introduction was a bit too detailed, especially given that ARs are not directly identified or analysed in the results. In response, we have revised this section to be more concise and focused, emphasizing the potential role of ARs as moisture sources for warm-sector precipitation, and their links to the WCB. The potential link between ARs and the studied precipitation patterns is also acknowledged in the conclusion.

5) Q2 is not addressed in a cyclone-relative way. I do think that the authors would have the necessary data and tools to address this question, with a bit more coding work and gridding the uptakes for different times relative to cyclone maximum depth.

Thank you for raising this point. Unfortunately, this analysis is more difficult than it seems, because the tool that we use only provides moisture uptake as an accumulated footprint over the eight-day period rather than at individual time steps. Please see our reply to the second general comment of reviewer 1, in which we discuss this in more detail.

6) L. 117: Here the Lagrangian method used should also be referenced (Sodemann et al. 2008).

Thank you for the suggestion. At this point in the text, our intention is to introduce the Lagrangian approach in general terms, in order to contrast it with Eulerian methods. We do not yet refer to our specific method. In light of comment 7, we have revised the corresponding sentences to make this comparison clearer, and we hope this now better justifies why the method by Sodemann et al. (2008) is not referenced here. The specific methodology we use is introduced and referenced in detail later in the manuscript.

7) L. 120: these are not adequate references for the use of Lagrangian moisture source identification to distinguish between different air streams. For example Pfahl et al. 2014 could be a good option.

Thank you for this comment. We agree that Pfahl et al. (2014) is a more appropriate reference when referring to the use of Lagrangian moisture source identification to distinguish between different air streams. We have now included this reference specifically in that context. The original references (Gimeno et al., 2012; Pérez-Alarcón et al., 2022) are more relevant for discussion on Eulerian versus Lagrangian approaches, and we have moved them earlier in the sentence accordingly. We have also revised the sentence structure to make the comparison between the two approaches clearer and to ensure that all references are cited in the most appropriate places.

*Lagrangian methods follow individual air parcels as they move in space and time, while Eulerian methods analyse moisture budgets at fixed locations (Gimeno et al., 2012). Although an Eulerian approach is also suitable for constructing a moisture budget (e.g. van der Ent and Tuinenburg, 2017), the advantage of the Lagrangian approach is that it allows for the quantification of moisture uptakes along air parcel trajectories and provides a high spatial resolution of moisture source diagnostics (Gimeno et al., 2012; Pérez-Alarcón et al., 2022). Furthermore, it enables the distinction between different air streams, allowing for the analysis of their associated moisture sources separately (e.g. Pfahl et al., 2014).* (lines 146-152)

8) L. 123: if the North Atlantic was studied several times before, then mention several studies. Here maybe e.g Gimeno et al. 2012, Perez-Alarcon et al. 2022 could be good options.

Thank you for this comment. We have included one of your suggestions (Perez-Alarcon et al. 2022) and added another one (Coll-Hidalgo et al., 2025).

*… as the North Atlantic has been studied several times before (e.g. Chang and Song, 2006; Coll-Hidalgo et al., 2025; Pérez-Alarcón et al., 2022), …*

9) L. 160: "Thereafter a new time axis … is defined", I think it's not a new axis, just a new time of reference.

Thank you for this suggestion. We have revised the sentence as follows:

*Thereafter, a new time axis of reference relative to the cyclone life cycle is defined.*

10) L. 168-170: Why did you exclude these cyclones? Some of them might also have made extratropical transition and actually be quite interesting to study in more detail.

We exclude these cases, as these cyclones already begin to exhibit wintertime characteristics due to increased baroclinicity later in the season. Our goal is to analyse summer cyclones only under summertime characteristics, which are characterized by weaker baroclinicity and generally weaker vertical motion. We fully agree that cyclones undergoing extratropical transition are scientifically interesting – particularly with regard to how their characteristics change throughout the year, especially during the transition seasons. However, we believe that these systems should to be studied separately, rather than be included as an essential part of this study.

11) L. 181: "exhibit strong movement" what does this mean exactly?

We argue that the cyclones in our subset exhibit strong movement because they can cross the Atlantic basin and make landfall in Europe. We quantify this further using the definition of a moving cyclone. To better link the statement to the definition, we updated the text (also incorporating comment 15 of reviewer 1).

12) Fig. 1: add contours of track density to help interpret Fig. 8.

Thank you for this suggestion. In the figure below, we have added the contours of the track density to the figure (here for all subregions taken together). However, we believe adding these contours make the figure less legible as there are already many different lines in this figure. Therefore, we prefer to keep the original version in the manuscript.

[Figure]

(a) Gulf Stream    Nr of tracks: 86 / 515
(b) Labrador Sea    Nr of tracks: 105 / 567
(c) East Atlantic    Nr of tracks: 231 / 1211
(d) Nordic Seas    Nr of tracks: 266 / 1517

Good to note is that we have added the mean cyclone track and the mean location of the cyclone center at the time of the precipitation to Fig. 8, which makes it easier to interpret.

13)L. 202: add Wernli and Davies 1997 for LAGRANTO to reference the original publication as well as.

Thank you for pointing this out. We have added the reference.

14)L. 212: justify the 8-days based on studies about the moisture residence time in this region and season.

We have justified the 8-days using the study by Läderach and Sodemann (2016).

*The latter is dependent on the residence time of moisture in the atmosphere, which in the North Atlantic region is estimated to be approximately four days on average, with values up to one day longer in summer than in winter (Läderach and Sodemann, 2016).* (lines 256-258)

15)L. 214: "LAGRANTO's ability to allocate a significant portion of precipitation to the right moisture sources" not sure I understand what you mean here. Do you mean Watersip instead of LAGRANTO? And how do you know what the "right sources" are?

Thank you for pointing this out. It is indeed not LAGRANTO itself that attributes precipitation to moisture sources, but rather the WaterSip diagnostic applied to the

LAGRANTO trajectories. As the WaterSip tool will be introduced a few sentences later, we chose to stick to a general term here.

We also acknowledge that the term *"right"* may be misleading, as we do not have observational evidence to confirm the correctness of the identified sources. We have therefore revised the sentence to clarify that the focus is on identifying *specific* moisture source regions, without implying absolute correctness.

*The number of days is chosen based on three factors: (1) computational cost and computer power, (2) numerical accuracy, and (3) the ability to allocate a significant proportion of precipitation to specific moisture sources.* (lines 254-256)

> 16)L. 219: Did you use the official Watersip code? Or an own implementation in which case it would be clearer to simply reference the original publication of the algorithm with Sodemann et al. 2008 and not call the algorithm Watersip. For reproducibility it would be easiest if you provided a link to the code used.

We used the official WaterSip code (version 3) for our analysis. To clarify this, we have updated the reference accordingly and included a better description and the version number when the tool is properly introduced in the manuscript.

*Line 261: Using these variables, the moisture budget of the air parcel can be constructed by applying the moisture source diagnostic WaterSip (Sodemann, 2025).*
*Line 264: In this study, we employ WaterSip (version 3), a software tool that implements the widely used Lagrangian moisture source and transport diagnostic developed by Sodemann et al. (2008), to identify the moisture sources (Sodemann, 2025).*

> 17)L. 239: in the 48 h around maximum cyclone depth.

We have revised the sentence to incorporate the 48-hour window.

*Once all the moisture uptake locations are identified, the absolute uptake amount is translated into a moisture source footprint for every 3 hours within the 48-hour window around maximum cyclone (from -24 h to 24 h).*

> 18)L. 255: at arrival in the cyclone (trajectory start is confusing).

We revised the sentence. See also comment 20 of reviewer 1 for similar changes in the manuscript.

> 19)L. 370: I would not call this the upper troposphere, there are very few trajectories coming from above 500 hPa.

We have rephrased the sentence as 'descending air parcels from higher up in the atmosphere' .

> 20)L. 376: implying that local moisture recycling is becoming important: what does that mean exactly?

In this context, we mean that the moisture that precipitates within the cyclone center is re-evaporated from the surface and reused by the same cyclone.

21)L. 415-416: here Aemisegger and Papritz, 2018 would be more fitting.

We have replaced the reference *Aemisegger and Sjolte, 2018* by *Aemisegger and Papritz, 2018*.

22)L. 453: It's not clear what the initial moisture is: the diagnosed precipitation?

With 'initial moisture' we mean the specific humidity carried by each air parcel at the start of the 8-day backward trajectory, i.e. in the cyclone center. We have revised the sentence as follows:

*Overall, WaterSip effectively attributes the majority of precipitation to its source region. For approximately 90% of all precipitating air parcels, over 50% of the water vapour arriving at the cyclone center could be accounted for within the 8-day period (Fig. A3).* (lines 529-531)

23)L . 465: I think this is really an important point of this paper. It should be emphasised more. This contradicts the usual assumption that in cyclones from the tropics making extratropical transition, subtropical or even tropical moisture gets exported into the midlatitudes.

Thank you for pointing this out. We agree that this is an important finding that deserves to be emphasized more clearly in the manuscript. Our results contradicts the common assumption that extratropical cyclones, especially those undergoing extratropical transition, are predominantly fuelled by subtropical or tropical moisture. We have revise Lines 543–544 to formulate this finding more strongly:

*However, this southeastern moisture source remains secondary to midlatitude sources, and we do not see any sources deep in the tropics.*

In the discussion section (lines 751-754 and lines 763-772), we go into more details:

*In addition, subtropical moisture sources appear more frequently than in winter, especially during the intensification phase, and contribute to more long-range moisture transport into the storm system. Yet they remain secondary to midlatitude sources, and we have not found any sources deep in the tropics. […] Despite the larger role of remote sources in summer, (sub)tropical moisture contributes only marginally to cyclone precipitation. This contradicts the common assumption that (sub)tropical moisture is an important contributor to precipitation associated with extratropical cyclones. The assumption likely arises from two factors: first, moisture availability is highest over subtropical oceans, where evaporation in summer exceeds precipitation by up to 4 mm day$^{-1}$ (Kållberg et al., 2005; Knippertz and Wernli, 2010). Secondly, ARs (or moisture transport axes in general) typically run parallel to the cold front, directed from the subtropics toward heavy precipitation regions in the extratropics (Dacre and Clark, 2025). Taken together, these factors would suggest a pathway for direct transport, that is especially evident in late summer when tropical cyclones undergoing extratropical transition are expected to*

*follow this pathway and enhance (sub)tropical moisture import. Yet our results indicate that midlatitude precipitation is predominantly caused by moisture originating from the same latitude band.*

24) L. 469-472: this raises the question of cyclone-relative sources, which is not addressed quantitatively in this paper.

Thank you for raising this point. Please see our reply to the second general comment of reviewer 1, in which we discuss this in more detail.

25) L. 474: Here "local" needs to be defined more quantitatively. Local relative to the cyclone?

We have revised this line in the manuscript to clarify what we mean. For a more detailed discussion, please see our reply to the first general comment from reviewer 1.

26) Fig. 9b: I think panel b does not make much sense given the relatively large North-South temperature gradient.

Thank you for raising this. We have decided to remove the panel and the associated interpretation.

27) Fig. 9d: make clear that panel d is based on the explained fraction.

In figure 9, all panels are based on the fraction of precipitation that was attributed to a source, and therefore 'explained'.

28) L. 526: "…hinting at cold-air advection" I think this is speculative.

There are parts of the manuscript where it seems that we have been too speculative (see also comment 30 and 28 and 33 by reviewer 1). We have therefore removed this statement.

29) L. 545: then I would say they are not WCB trajectories. What is the role of convection and the relatively coarse spatial and temporal resolution of the ERA5 data?

As noted in our response to comment 3, we prefer not to state that WCBs are absent simply because the ascent does not meet the typical threshold of 600 hPa in 48 hours. This threshold is primarily based on wintertime WCBs, and applying it to summertime cyclones may exclude relevant airstreams, as ascent tends to be weaker in summer due to reduced baroclinicity. Instead of quantifying how many trajectories meet that threshold, we prefer to focus on the presence of an ascending airstream consistent with the conceptual WCB. It seems that even in the original conveyor belt model described by Browning (1990), such airstreams are still considered part of the WCB – albeit with weaker vertical motion.

Regarding the role of ERA5, we agree this is important to acknowledge. The relatively coarse spatial and temporal resolution of ERA5, along with its

parameterization of convective processes, may underestimate the strength and vertical extent of ascent, especially in cases of deep convection like in the WCB. We have clarified this point in the ERA5 description in the manuscript.

30) Section 4.5: the discussion in this section is a bit speculative without cyclone-relative analysis of the moisture sources.

Thank you for raising this point. In this section, we analyse the spatial distribution of surface evaporation within and around the cyclone center, rather than the actual moisture sources. To make this distinction clearer, we have changed the section title to *"Cyclone-relative perspective on evaporation"* . In addition, we have added cyclone-relative winds to Figure 11, providing additional evidence on the direction of moisture transport. Therefore, the lines 664–666 have been removed, and the lines 675–677 have been revised, as both contained hypotheses regarding the cyclone-relative moisture sources. Instead, the lines have been replaced with the actual interpretation of Figure 11:

*The cyclone-relative winds, however, indicate that the flow in this region is directed away from the cyclone center toward the south and southwest. The converging motion expected during the intensification phase is instead observed to the north of the cyclone center, and will promote the ascending motion. As the cyclones mature, the cyclone-relative winds become more cyclonic, linked to the cyclones propagating more slowly and becoming deeper. At this stage, evaporation to the south of the cyclones still cannot contribute to the cyclones' precipitation, because the winds still have no northerly component.*

*However, the moisture evaporating behind the cold front is not transported to the warm sector or into the WCB, as the presence of strong horizontal divergence in the cyclone-relative winds, together with subsidence in the cold sector, acts as a barrier to vertical ascent and transport across the front.*

31) L. 570: "… facilitating strong upward latent heat fluxes": yes and what matters even more for your study, is that given that the subsiding air is dry, it's efficiency in taking up humidity is large (Aemisegger and Papritz 2018).

Agreed. We have revised the sentence, please see our response to comment 51 of reviewer 1 for the changes.

32) L. 571: here maybe Illotovitz et al. 2021 about the impact of the dry intrusion on boundary layer dynamics and surface fluxes would be a good reference.

Agreed. We have revised the sentence, please see our response to comment 51 of reviewer 1 for the changes.

33) L. 657: The fact that you do not find a feeder airstream is due to the fact that you do not look at the moisture sources in a cyclone-relative perspective. Without such a cyclone-relative analysis (which I do think is feasible and which I would strongly recommend), you cannot really make this statement. If you really do not find a "feeder airstream" in your summer cyclone based on such an analysis this would reveal an interesting contrast and rise new

questions about the cyclone-relative moisture cycling in summer vs. winter cyclones.

Indeed, without the cyclone-relative moisture sources it is difficult to identify a feeder airstream. Performing the cyclone-relative analysis is more difficult than it seems, because the tool that we use only provides moisture uptake as an accumulated footprint over the eight-day period rather than at individual time steps. We have therefore added the cyclone relative winds, as this gives a first-order estimate of the direction of moisture transport.

Instead of discussing the feeder airstream concept, which is difficult without a cyclone-relative analysis, we have chosen to revise the sentence (Line 789) to focus more on what is (not) observed in Figure 11.

*In the composites of all 688 cyclones, however, we could not identify a primary cyclone in the MSLP contours, nor a region of enhanced surface evaporation to the northeast.*

> 34) L. 688: "… or from the developing cyclone's own cold sector (which appears to be more important in summer)", this sounds contradictory with the explanation at L. 573ff, where the authors discuss that this moisture recycling pathway is unlikely due to the necessary crosscold front motion of the air parcels that would rain out in the same cyclone. I recommend to rephrase this according to the statements made earlier in the paper.

Thank you for pointing this out. You are absolutely right, we do not expect moisture that evaporates in a cyclone's own cold sector to contribute directly to its own precipitation. We have revised this sentence as follows:

*The second involves cold-air advection, in which relatively colder air from the north acquires moisture within the cyclone's cold sector. In our case study, we observed that this mechanism can serve as a moisture source for precipitation in a subsequent cyclone. From a climatological perspective, however, this behaviour is more common in winter.* (lines 818-821)

**Community comment**
Although the authors would not necessarily be aware of this work, a recent article with similar aims and methods has been published (Coll-Hidalgo et al. 2025) https://www.sciencedirect.com/science/article/pii/S0169809525002972?via%3Dihub#f0025
I believe that the manuscript would benefit from a comparison with the methods, results and conclusions of this article.

Thank you for pointing us to this paper, which we were not aware of. While the overall aim is closely aligned with ours, namely to investigate moisture sources for precipitation in North Atlantic extratropical cyclones, there are also important methodological differences. Notably, their study focuses on the winter season, employs downscaled ERA5 data using the WRF mesoscale model, and determines the moisture sources for three specific cyclone features: the cyclone radius (which aligns with our approach), the warm conveyor belt (WCB) footprint, and the squareroot spiral area. These differences make a direct, one-to-one comparison difficult, as differences in seasonality, model resolution, and cyclone feature definitions can all influence the results. That said, we agree that comparing findings based on the cyclone radius framework is particularly relevant, and we will reflect on the sensitivity of the different precipitation targeting approaches in our discussion section (lines 775-778).

*At the same time, it is important to acknowledge that the estimated contribution of (sub)tropical sources depends on how the precipitation target is defined. In this study, we use a radius approach. Coll-Hidalgo et al. (2025) found that using alternative precipitation-targeting methods can extend the moisture source footprint farther south into the subtropics.*

**References**

Benedict, I., Weijenborg, C., van der Ent, R., Keune, J., Koren, G., and Kalverla, P.: A moisture tracking intercomparison study - Addressing the uncertainty in modelling the origins of precipitation, EMS Annual Meeting 2024, Barcelona, Spain, 1–6 Sep 2024, EMS2024-1040, https://doi.org/10.5194/ems2024-1040, 2024.

Binder, H., Boettcher, M., Joos, H., Sprenger, M., and Wernli, H.: Vertical cloud structure of warm conveyor belts – a comparison and evaluation of ERA5 reanalysis, CloudSat and CALIPSO data, Weather Clim. Dynam., 1, 577–595, https://doi.org/10.5194/wcd-1-577-2020, 2020.

Browning, K.: Organization of clouds and precipitation in extratropical cyclones, in: Extratropical cyclones, 129–153, Springer, https://doi.org/10.1007/978-1-944970-33-8_8, 1990.

Browning, K. A.,: The dry intrusion perspective of extra-tropical cyclone development. Meteor. Appl., 4, 317–324, https://doi.org/10.1017/S1350482797000613, 1997.

Chang, E. K. M., Lee, S., and Swanson, K. L.: Storm track dynamics, J. Climate, 15, 2163–2183, https://doi.org/10.1175/1520-0442(2002)015<02163:STD>2.0.CO;2, 2002.

Coll-Hidalgo, P., Nieto, R., Fernández-Alvarez, J. C., and Gimeno, L.: Assessment of the origin of moisture for the precipitation of North Atlantic extratropical cyclones: Insights from downscaled ERA5, Atmos. Res., 324, 108205, https://doi.org/10.1016/j.atmosres.2025.108205, 2025.

Dacre, H. F. and Clark, P. A.: A kinematic analysis of extratropical cyclones, warm conveyor belts and atmospheric rivers, npj Clim. Atmos. Sci., 8, 97, https://doi.org/10.1038/s41612-025-00942-z, 2025.

Dacre, H. F., Martinez-Alvarado, O., and Mbengue, C. O.: Linking atmospheric rivers and warm conveyor belt airflows, J. Hydrometeorol., 20, 1183–1196, https://doi.org/10.1175/JHM-D-18-0175.1,  2019.

Gimeno, L., Stohl, A., Trigo, R. M., Dominguez, F., Yoshimura, K., Yu, L., Drumond, A., Durán-Quesada, A. M., and Nieto, R.: Oceanic and terrestrial sources of continental precipitation, Rev. Geophys., 50, RG4003, 10.1029/2012RG000389, 2012.

Hartmann, D. L.: Global Physical Climatology, Academic Press, San Diego, CA, 408 pp., ISBN: 9780123285300, 1994.

Läderach, A., and Sodemann, H.: A revised picture of the atmospheric moisture residence time, Geophys. Res. Lett., 43, 924–933, doi:10.1002/2015GL067449, 2016.

Oertel, A., Boettcher, M., Joos, H., Sprenger, M., Konow, H., Hagen, M., and Wernli, H.: Convective activity in an extratropical cyclone and its warm conveyor belt – a case-study combining observations and a convection-permitting model simulation, Q. J. Roy. Meteorol. Soc., 145, 1406–1426, 2019.

Papritz, L., Aemisegger, F., and Wernli, H.: Sources and transport pathways of precipitating waters in cold-season deep North Atlantic cyclones, J. Atmos. Sci., 78, 3349–3368, https://doi.org/10.1175/JAS-D-21-0105.1, 2021.

Peixoto, J. P. and Oort, A. H.: Physics of Climate, American Institute of Physics, New York, 520 pp., ISBN: 9780883187128, 1992.

Raveh-Rubin, S.: Dry Intrusions: Lagrangian Climatology and Dynamical Impact on the Planetary Boundary Layer. J. Climate, 30, 6661–6682, https://doi.org/10.1175/JCLI-D-16-0782.1, 2017.

Sodemann, H.: The Lagrangian moisture source and transport diagnostic WaterSip V3.2, EGUsphere [preprint], https://doi.org/10.5194/egusphere-2025-574, 2025.

Sodemann, H., and Stohl, A.: Asymmetries in the moisture origin of Antarctic precipitation, Geophys. Res. Lett., 36, L22803, doi:10.1029/2009GL040242, 2009.

---

## Referee Report (RR1)

Precipitation, Moisture Sources and Transport Pathways associated with Summertime North Atlantic Deep Cyclones
egusphere-2025-1752

This paper aims to evaluate the moisture sources for summertime extratropical cyclones. It employs a Lagrangian back trajectory method to determine the sources and sinks of moisture for parcels that result in precipitation at the centre of cyclones. The authors have done a good job of addressing the comments I made on the original submission. There are, however, a couple of remaining points that should be addressed before publication.

1. Line 112: The authors state that 'ARs are capable of feeding multiple WCBs from individual cyclones. This is not correct. As shown nicely by the authors in figure 11, and described on lines 662, the cyclone-relative wind vectors point away from the cyclone centre indicating that the moisture within the AR is travelling slower than the cyclone itself. Therefore, the AR cannot feed moisture into the WCB 'because the winds still have no northerly component' (line 666)

2. Line 383-7, figure 8, lines 555-565: The weighted mean source distance is proportional to the ETC propagation speed. I.e. Early in the ETC lifecycle, when they are moving fastest, the weighted mean source distance is large, and later in the ETC lifecycle, when they slow down, the weighted mean source distance is small. Thus the 'moisture sources move closer to the cyclone centre' as the ETCs reach maturity (fig 8) because the ETCs transport moisture polewards at a slower speed over the same time period, so it moves a shorter distance from its origin. This is also why 'despite greater source distances observed during the intensification phase, the residence time remains relatively constant' (line 610). It would be good for the authors to link up these ideas from different parts of the paper to illustrate nicely that the weighted mean source distance is proportional to the speed at which the ETC is moving.

---

## Author Response (AR2)

**Precipitation, Moisture Sources and Transport Pathways associated with Summertime North Atlantic Deep Cyclones**
egusphere-2025-1752
**Authors' response**

This paper aims to evaluate the moisture sources for summertime extratropical cyclones. It employs a Lagrangian back trajectory method to determine the sources and sinks of moisture for parcels that result in precipitation at the centre of cyclones. The authors have done a good job of addressing the comments I made on the original submission. There are, however, a couple of remaining points that should be addressed before publication.

Thank you for reviewing the paper again. We are glad to hear that we properly addressed your comments on the original submission. We have addressed your final two points below, with changes to the manuscript in italics.

1. Line 112: The authors state that 'ARs are capable of feeding multiple WCBs from individual cyclones. This is not correct. As shown nicely by the authors in figure 11, and described on lines 662, the cyclone-relative wind vectors point away from the cyclone centre indicating that the moisture within the AR is travelling slower than the cyclone itself. Therefore, the AR cannot feed moisture into the WCB 'because the winds still have no northerly component' (line 666)

Thank you for pointing this out. This statement was included to summarize previous studies, particularly that of Sodemann and Stohl (2013). They found that the WCBs of several cyclones in December 2006 were fed by atmospheric rivers. Indeed, our results (Figure 11) show no evidence of similar behaviour, and the recent study by Dacre and Clark (2025) also stresses that the moisture within the AR is travelling slower than the cyclone itself, implying that continual local moisture replenishment acts as a source rather than the AR structure. Including the statement would therefore generalize too much, and to avoid any confusion we have chosen to remove the full sentence from the manuscript.

2. Line 383-7, figure 8, lines 555-565: The weighted mean source distance is proportional to the ETC propagation speed. I.e. Early in the ETC lifecycle, when they are moving fastest, the weighted mean source distance is large, and later in the ETC lifecycle, when they slow down, the weighted mean source distance is small. Thus the 'moisture sources move closer to the cyclone centre' as the ETCs reach maturity (fig 8) because the ETCs transport moisture polewards at a slower speed over the same time period, so it moves a shorter distance from its origin. This is also why 'despite greater source distances observed during the intensification phase, the residence time remains relatively constant' (line 610). It would be good for the authors to link up these ideas from different parts of the paper to illustrate nicely that the weighted mean source distance is proportional to the speed at which the ETC is moving.

Thank you for pointing this out. We did not realize that we did not discuss the values of the weighted mean source distance provided in Figure 8. Therefore, we agree that it makes sense to discuss this in Section 4.2 and link it to the propagation speed of the cyclones. Furthermore, we believe it would make sense to revisit this relationship in section 4.3. The changes we made are as follows:

*Another important observation of the moisture source footprint is its large spatial extent, especially during the intensification phase. Consequently, the weighted mean source distance is also large during this phase.* (Line 551-552)

*This north-eastward shift of the moisture footprint exceeds the movement of the mean cyclone center in all subregions except in the Gulf Stream region (Fig. A4e), as the cyclones slow down once they mature. This implies that, over time, moisture sources move closer to the cyclone center. The weighted mean source distance, which scales with the cyclones' propagation speeds, thus decreases throughout the cyclone life cycle (Fig. 8). At a later stage, local moisture recycling becomes more important, although overall moisture uptake is lower than during the intensification phase.* (Lines 562-567)

*Despite the greater source distances observed during the intensification phase (Fig. 9a), the residence time remains relatively constant, suggesting that moisture must be transported more rapidly during this phase. This is consistent with the presence of stronger winds and a faster-moving cyclone, both of which facilitate faster long-range transport.* (Line 612-615)

One more note for the editor: after reading the manuscript, some extra spaces have been removed, "coloured" has been changed to "coloured" for consistency (in the caption of Fig. 5), just as 'lifecycle' has changed to 'life cycle' (line 387).